



# Study the impact of three Asian industrial regions on PM$_{2.5}$ in Taiwan and the process analysis during transport

Ming-Tung Chuang[1], Maggie Chel Gee Ooi[2], Neng-Huei Lin[2], Joshua S. Fu[3], Chung-Te Lee[4], Sheng-Hsiang Wang[2], Ming-Cheng Yen[2], Steven Soon-Kai Kong[2], Wei-Syun, Huang[2]

[1]Research Center for Environmental Changes, Academia Sinica, Taipei 11529, Taiwan
[2]Department of Atmospheric Science, National Central University, Taoyuan, 32001, Taiwan
[3]Department of Civil and Environmental Engineering, University of Tennessee, Knoxville, TN 37996, USA
[4]Graduate Institute of Environmental Engineering, National Central University, Taoyuan, 32001, Taiwan

*Correspondence to: mtchuang100@gmail.com*

**Abstract.** The outflow of East Asian haze (EAH) has gathered much attention in recent years. For downstream areas, it is meaningful to understand the impact of crucial upstream sources and the process analysis during transport. This study evaluated the impact of PM$_{2.5}$ from the three biggest industrial regions in Asian continent: Bohai Rim industrial region (BRIR), Yangtze River Delta industrial region (YRDIR), and Pearl River Delta industrial region (PRDIR) on Taiwan and discussed the processes during transport with the help of air quality modeling. The simulation results revealed the contributions of monthly average

PM$_{2.5}$ from BRIR and YRDIR were 0.7~1.1 µg m$^{-3}$ and 1.2~1.9 µg m$^{-3}$ (~5 % and 7.5% of total concentration) on Taiwan, respectively in January 2017. When the Asian anticyclone moved from Asian continent to the West Pacific, e.g. on Jan 9th 2017, the contributions from BRIR and YRDIR to northern Taiwan could reach 6~8 and 9~12 µg m$^{-3}$. The transport of EAH from BRIR and YRDIR to low latitude regions was horizontal advection (HADV), vertical advection (ZADV), and vertical diffusion (VDIF) over Bohai Sea and East China Sea. Over Taiwan Strait and northern South China Sea, cloud processes

(CLDS) was the major production process of PM$_{2.5}$ due to high relative humidity environment. Along the transport from high latitude regions to low latitude regions, Aerosol chemistry (AERO) and Dry deposition (DDEP) were the major removal processes. When the EAH intruded northern Taiwan, the major production processes of PM$_{2.5}$ at northen Taiwan were HADV and AERO. The stronger the EAH was the easier the EAH could influence central and southern Taiwan. Although PRDIR was located at the downstream of Taiwan under northeast wind, the PM$_{2.5}$ from PRDIR could transport upward above boundary

layer and moved eastwards. When the PM$_{2.5}$ plume moved overhead Taiwan, PM$_{2.5}$ could transport downward via boundary layer mixing (VDIF) and further enhanced by the passing cold surge. In contrast, for the simulation of July 2017, the influence from three industrial regions was almost negligible unless there was special weather system like thermal lows, which may carried pollutants from PRDIR to Taiwan, but the occurrence was rare.

## 1. Introduction

The damage of PM$_{2.5}$ (aerodynamic diameter is equal or less than 2.5 µm) on respiratory system has been proved (Kagawa, 1985; Schwartz et al., 1996;Zhu et al., 2011). The short-term human exposure to PM$_{2.5}$ could inflict cardiovascular and respiratory diseases, reducing lung functions, and increasing respiratory symptoms such as rapid breath, cough, and asthma. While the long-term influences include the mortality from heart or lung disease, cardiovascular illness (Pope et al., 2004;Brook et al., 2004;Ohura et al., 2005), and overuse of medical resources (Atkinson et al., 2001). Environmentally, the PM$_{2.5}$

not only absorbs and scatters solar radiation but also impairs visibility (Na et al., 2004), influences the balance of radiation and global climate (Hu et al., 2017), and the heterogeneous reactions of oxidants in the troposphere (Tie et al., 2005).
The East Asian haze (EAH) usually occurs in spring and winter around the East Asia due to the rapid development of Asian countries over the last few decades (Fu et al., 2014; Yang et al., 2016). When the Asian anticyclones was formed at the Siberia moved southeastwards, the peripheral circulation usually transported EAH to downwind regions including Korea, Japan, and





Taiwan (Zhang et al., 2015). Most literatures discussing the transport of EAH in recent years generally applied two methods: the trajectories statistics (TS) and the chemical transport modeling (CTM). The TS method calculated the frequency of the backward trajectories passing through specific surrounding regions. The frequency of the trajectories passing through a specific region implied the impact level of this region. The trajectories could be calculated from the archived meteorological data from NOAA ARL (www.ready.noaa.gov/archives.php) or the model outputs of MM5 (Mesoscale Model version 5, Dudhia, 1993) or WRF (Weather Research and Forecasting, Skamarock and Klemp, 2008). Pawar et al. (2015) utilized the TS method to assess the impacts of short-range and long-range transport (LRT) $PM_{2.5}$ on Mohali in north-west Indo-Gangetic plain. Similar method was applied to evaluate the contribution of LRT of $PM_{2.5}$ to south-western Germany (Garg and Sinha, 2017) and eastern Germany (van Pinxteren et al., 2019). Yang et al. (2018) also used this method to evaluate the influence of $PM_{2.5}$ from the Bohai Sea, Yangtze River Delta, and Pearl River Delta regions on Beijing. Although the TS method has been used widely, the passing frequency over some specific regions can only approximate statistics of the contributions from those regions. Using trajectory to express the moving of a polluted plume would contain substantial uncertainty.

The application of CTM on the study of transport usually comprises two methods: the Brute Force Method (BFM) and the Apportionment Method (AM). The principle of BFM is to run two simulations: one control run and another one without certain emission source. The difference of these two simulations is the contribution of that specific source. BFM method has been widely used for estimating the contribution of a specific source or the effect of a control strategy (Marmur et al., 2005; Burr and Zhang, 2011; Chen et al., 2014; Li et al., 2017) because this method is easy and straightforward. Nevertheless, this method is not perfect for potentially under-represented chemical reaction between the specific source with the remaining sources. Therefore, the BFM method is more reliable if the effect of the chemical reaction is minor. The AM method is more complex and applied the idea of apportionment technique into CTM model. The simulation consumes much computing resources, but it could estimate the contributions of different emission sources in a run. Skyllakou et al. (2014) applied the particulate matter source apportionment technique (PSAT, Wagstrom et al., 2008) in PMCAMx model (Fountoukis et al., 2011) to assess the impact of local pollution (LP), short distance transport (50-500 km), and LRT (>500 km) on Paris in France. Kwok et al. (2013) also developed a similar technique called Integrated Source Apportionment Method (ISAM) in CMAQ model (Byun and Schere, 2006). The AM method can be used to evaluate the contributions of different emission sources simultaneously; however, it does not comprehensively account for the non-linear chemical reactions between sources. BFM and AM methods both have their edge over the other. The CTM modeling requires large computer resources and contains many uncertainties like emissions, meteorology, chemical mechanisms, and numerical methods. However, the CTM is able to give clearer contributions from a specific source compared to the TS method.

The LRT of EAH has tremendous impact on the air quality in Taiwan. The following is a brief of such modelling studies. Chang et al. (2000) applied the CTM to simulate the influence of LRT acid pollutants from East Asian to Taiwan. In the six events of 1993, the average contribution accounted for 9－45% and 6－33% of total sulfur and nitrogen deposition on Taiwan, highest when the northeast monsoon prevailed. Lin et al. (2004) examined the meteorological and air quality data from November 1999 to May 2000, and from November 2000 to May 2001 in Taiwan. They classified the LRT in winter into dust transport, frontal transport with pollutants, and LRT of background air mases which contributed an average $PM_{10}$ level of 127.6 µg m$^{-3}$, 85.0 µg m$^{-3}$, and 32.8 µg m$^{-3}$ respectively. Furthermore, the frequencies of LRT events and LP events were 25.2% and 71.7% (missing data accounts 3.1%). Chuang et al. (2008a) classified the pollution weather patterns for Taipei $PM_{2.5}$ events. They coined the weather system during LRT events as the "high-pressure pushing" in which the high-pressure systems advected the pollutants from Asian continent to Taiwan. Subsequently, Chuang et al. (2008b) utilized CMAQ to simulate the chemical evolution of $PM_{2.5}$ compositions in the moving plume. They found that the proportion of nitrate and sulfate would decrease and increase respectively along the path. Chen et al. (2013, 2014) also applied the CMAQ to assess the $PM_{2.5}$ distribution in East Asia and subsequently estimated the impact of $PM_{2.5}$ from Asian continent on Taiwan. They suggested the direct and indirect LRT accounted for 27% and 10% of $PM_{2.5}$ in Taiwan in 2007. For the autumn and winter of 2007, the LRT



contributed 39% and 41% of total PM$_{2.5}$ in Taiwan. Wang et al. (2016) combined backward trajectories and AOD distribution to estimate the impact of EAH on Taiwan. Their results suggested the PM$_{2.5}$ level was 57.1±13.6 µg m$^{-3}$ for haze event, which

is four folds of the background events (13.7±7.4 µg m$^{-3}$) from 2005 to 2013. They also estimated pollution transport time from the Yangtze River Delta (YRD) to the northern tip of Taiwan was about 28 hours. Chuang et al. (2017) discussed three types of PM$_{2.5}$ episodes into the long-range transport (LRT), the local pollution (LP)and the LRT/LP mix. Both the simulation and observation showed the proportion of NO$_3^-$ in PM$_{2.5}$ was very small in the EAH and strong north to northeast wind increased the proportion of sea salt. Chuang et al. (2018) developed an efficient method to estimate the LRT-PM$_{2.5}$ and LP-PM$_{2.5}$ at any

place in Taiwan. They classified the daily PM$_{2.5}$ into LRT-Event, LRT-Ordinary, and LRT/LP&Pure LP events, which were 31-39 µg m$^{-3}$, 12-16 µg m$^{-3}$, 4-13 µg m$^{-3}$ at the northern tip of Taiwan from 2006 to 2015 for northeast monsoon period. On average, the ratio of LRT-PM$_{2.5}$ and LP-PM$_{2.5}$ for LRT-Event was 70:30 for northern Taiwan, 50:50 for central Taiwan, and 30:70 for southern Taiwan; for LRT-Ordinary was 60:40 for northern Taiwan and 40:60 for central and southern Taiwan; for LRT/LP&Pure LP was 30:70 for northern Taiwan and 25:75 for central and southern Taiwan. Their results also showed the

annual LRT-PM$_{2.5}$ decreased since 2013, which implied the emissions in Asian continent decreased since then.
The above studies all showed the East Asian continent was the dominant source of LRT PM$_{2.5}$ for Taiwan in winter period. Therefore, if we can realize the sources contribute the most to LRT PM$_{2.5}$ and the transport pathway, then we can enhance the ability to predict the LRT PM$_{2.5}$, i.e. the EAH. From the emission map of Asia (Zheng et al., 2018), the largest emission source was power and industry sector. The three biggest industrial regions in mainland China are the Bohai Rim industrial region

(BRIR), the Yangtze River Delta industrial region (YRDIR), and the Pearl River Delta industrial region (PRDIR), as illustrated in Fig. 1. The present study attempts to assess the impact of these three industrial regions on the PM$_{2.5}$ in Taiwan. It applied the CTM with BFM method to simulate four scenarios: the *Base* (control case with integrated emissions), *Brir* (all emissions except BRIR), *Yrdir* (all emissions except YRDIR), and *Prdir* (all emissions except PRDIR) scenarios and thus resulted in the contributions of each industrial region. When estimating the contribution of BRIR, YRDIR, and PRDIR to PM$_{2.5}$ in Taiwan,

we used the difference between the *Base* case and the *Brir*, *Yrdir*, and *Prdir* cases. In addition, this study applied the Integrated Process Rate (IPR) technique (Byun and Schere, 2006; Liu and Zhang, 2013; Zhu et al., 2015) in CMAQ to discuss the process analysis during transport from the industrial regions to Taiwan. The bottom 20 layers (below 1.7 km) were selected for IPR analysis since they have covered the boundary layer where the physical and chemical processes take place. The climate in East Asia basically is divided into the northeast monsoon season in winter and southwest monsoon season in summer. In order to

understand the LRT in different seasons, the simulation periods for this study were January and July 2017. We also selected representative events to discuss in detail.

## 2. Methods

It is known that the EAH events mainly occur in winter (Chuang et al., 2008a; Wang et al., 2016). Although the high PM$_{2.5}$ events in Taiwan caused by the EAH during spring period sometimes was enhanced by the Southeast Asian biomass burning

aerosol (Yen et al., 2013; Chuang et al., 2016; Lin et al., 2017), the latter would implicitly complicate the transport of EAH and their co-occurrence has left to be a study in the near future. Therefore, this study chose January and July 2017 to represent the LRT in winter and summer period and the contrast between them. In addition, year 2017 was selected for this study is that it can reflect the impact of EAH lately because the anthropogenic emission in China has been decreasing obviously in recent years (Zheng et al., 2018; Chuang et al., 2018).

### 120   2.1 Geographical location of meteorological and air quality observation sites

Taiwan is an island located in the West Pacific and separated from mainland China on the west by the Taiwan Strait. The north is the China East Sea and the south sits the Philippines across the Bashi Strait. For meteorology evaluation, we chose eight





representative stations: the PJY (#1 in Fig. 1), TPE (#2 in Fig. 1), CP (#3 in Fig. 1), TC (#4 in Fig. 1), CY (#5 in Fig. 1), TN (#6 in Fig. 1), KH (#7 in Fig. 1), and HC (#8 in Fig. 1) stations to evaluate the modeling performance of temperature, wind speed, and wind direction. Since most residents lived at the relatively flat western Taiwan, the observations at the BQ (#9 in Fig. 1), PZ (#10 in Fig. 1), ML(#11 in Fig. 1), ZM (#12 in Fig. 1), CY (#13in Fig. 1), TN (#14 in Fig. 1), ZY (#15 in Fig. 1), and HC (#16 in Fig. 1) stations were chosen for PM$_{2.5}$ evaluation.

## 2.2 Models and modeling configuration

This study applied the WRF v3.9.1 (Skamarock and Klemp, 2008) and CMAQ v5.2.1 (Byun and Schere, 2006) for scenario simulations. The initial meteorological condition was from NCEP diagnostic fields. Horizontal resolutions of four domains from outer to inner were 81, 27, 9, and 3 km, respectively. The first domain covered the East Asia and Southeast Asia and the fourth domain contained only the Taiwan island. The vertical layers were 46, about 20 layers below 1.7 km, in which the boundary layer was well resolved. The anthropogenic emissions for East Asia and Taiwan island were obtained from MIX (Multi-resolution Emission Inventory for China, Li et al., 2017) and TEDS 10.0 (Taiwan Emission Data System, TEPA, 2017), which are based on the years of 2010 and 2016, respectively. The MIX emissions of SO$_2$, NO$_X$, NMHC, NH$_3$, CO, PM$_{10}$, and PM$_{2.5}$ were adjusted with change of -62%, -17%, 11%, 1%, -27%, -38%, and -35%, respectively, according to the change of annual emission between 2010 and 2017 (Zheng et al., 2018). This study assumes the emission of 2017 in Taiwan is the same as that of 2016. The biogenic emissions were prepared by the Biogenic Emission Inventory System version 3.09 (BEIS3, Vukovich and Pierce, 2002) for Taiwan island and Model of Emissions of Gases and Aerosols from Nature v2.1 (MEGAN, Guenther et al., 2012) for regions outside Taiwan. While the biomass burning emissions imported the data of FINN v1.5 inventory (Wiedinmyer et al., 2011). All the remaining modeling configuration for this study is the same as that in Chuang et al. (2017).

## 2.3 Model evaluation

This study used statistical indexes such as MB (Mean Bias), MAGE (Mean Average Gross Error), and IOA (Index of Agreement) to evaluate temperature and wind speed, and used WNMB (Wind Normalized Mean Bias) and WNME (Wind Normalized Mean Error) for wind direction. For PM$_{2.5}$ performance, we applied MB (Mean Bias), MFB (Mean Fractional Bias), and MFE (Mean Fractional Error), R (Correlation coefficient), and IOA indexes. All the formulas for above indexes are from Emery (2001) and TEPA (2016), illustrated in Supplement S1.

### 2.3.1 Evaluation of WRF meteorological modeling

The MB performance shows that the temperature is slightly overestimated for PJY which is located in the outer sea of northern Taiwan (Table 1). The MAGE appeals simulated temperature at all stations is reasonable in both months. While the IOA indicates the simulated temperature at PJY and KH was not well enough. The deviation of simulated temperature for PJY and KH could be influenced by the sea surface temperature since these stations are nearer the sea than other stations. The performance of MB indicates the simulated wind speed was underestimated at TN, which led to the low IOA. In contrast, the simulated wind speed was overestimated at HC, which could be due to the smoother terrain in the simulation than the actual situation. The performance of wind direction at most stations are within the range of acceptance but not so well for TC and CY. The deviation could potentially due to the influences of nearby buildings. In summary, the simulated temperature, wind speed, and wind direction performed reasonably acceptable since most indices at many stations complied with the benchmark.

### 2.3.2 Evaluation of CMAQ chemical modeling

For the *Base* case, the simulated PM$_{2.5}$ was overestimated in all stations except CY and HC in January 2017 (Table 2). The performance of trend (correlation coefficient, R) is acceptable or good for all stations except HC. It is rather difficult to simulate



the wind speed at HC well which is located at the downwind south tip of Taiwan (Chuang et al., 2016). It is therefore reasonable that overestimated wind speed in HC led to poor underestimation of PM$_{2.5}$. Because the performance of PM$_{2.5}$ in HC is very poor, the following discussion will exclude this station and leave it to future improvement.

## 3. Results and Discussion

### 3.1 The impact of PM$_{2.5}$ from the Chinese three major industrial regions in January 2017

For the impact of three industrial regions on PM$_{2.5}$ in Taiwan in January 2017, the monthly mean impact from BRIR was about 0.7-1.1 µg m$^{-3}$ as illustrated in Fig. 2(a). The impact was higher in the northern Taiwan, about 5% of total PM$_{2.5}$. The proportion of influence gradually decreased from north to south (Fig. 2(b)). From the view of daily average, Fig. 3(a-1)-(a-7) show that the trend is similar for seven air quality stations and the impact on northern Taiwan was higher than central and southern Taiwan. In January 2017, the proportion of influence was higher on the 8th to 14th and the 20th to 23rd. It is found that the influence of EAH was closely related to the intrusion of Asian anticyclones. This study selected Jan 9th and Jan 13th for discussion of PM$_{2.5}$ events in section 3.5.

Comparing Fig. 2(a)/(b) with Fig. 2(c)/(d), it is apparent that the monthly mean influence from YRDIR was higher than BRIR. The reason is that YRDIR was nearer to Taiwan than BRIR. The monthly mean impact from YRDIR was about 1.2-1.9 µg m$^{-3}$, highest in northern Taiwan with the proportion of about 7.5% of total monthly average PM$_{2.5}$ concentration. The the spatial influence from BRIR was similar to YRDIR since these two industrial regions are both located off the north of Taiwan, i.e., the upstream of Taiwan under prevailing northeast wind. For the daily mean influence, the impact of YRDIR was also higher than BRIR and the influencing period were almost the same for both regions (Fig. 3(a-1)-3(a-7), Fig. 3(b-1)-3(b-7)). In particulary, the contributions from BRIR and YRDIR to northern Taiwan could reach 6~8 and 9~12 µg m$^{-3}$ on Jan 9th 2017. The spatial distribution of influence from PRDIR was totally different from BRIR and YRDIR as shown in Fig. 2(e) and Fig. 2(f). Interestingly, the impact from PRDIR was higher on the mountains than on the ground. For the ground stations, there was minor influence on 8th to 12th January 2017 (Fig. 3(c-1)-3(c-7)). It is found that there is a stationary front from the sea north of Taiwan extended southwest to Fujian and Guangdong provinces on January 7th (Fig. S2.1(a)). The front passed Taiwan on January 8th (Fig. S2.1(b)). Fig. 3(c-1)-3(c-7)) show that the influence on the southern Taiwan was higher than that on the northern Taiwan. Similar fronts passed Taiwan on January 10th (Fig. S2.1(c)) and 12th (Fig. S2.1(d)). From Fig. 4, it is found that the PM$_{2.5}$ from PRDIR would transport pollutants upward above the top of boundary and then moved eastwards (Fig. 4(a-1), Fig. 4(b-1)). When pollutants ran into the mountains in Taiwan, most part was blocked and transported to the ground through vertical mixing (Fig. 4(a-2)-4(a-3), Fig. 4(b-2)-4(b-3)). This transport mechanism is quite similar to the biomass burning aerosols from Indochina to Taiwan (Chuang et al., 2016, Yen et al., 2013). In addition to boundary layer mixing, the subsidence of cold surge enhanced the downward transport and increased PM$_{2.5}$ on the ground.

### 3.2 The physical and chemical processes of LRT from the Chinese three major industrial regions to Taiwan in January 2017

This study applied the process analysis technique in the CMAQ model, in which the terms of Horizontal advection (HADV), Vertical advection (ZADV), Horizontal diffusion (HDIF), Vertical diffusion (VDIF), Emissions (EMIS)、Dry deposition (DDEP)、Cloud process and aqueous chemistry (CLDS), Gas chemistry (CHEM), and Aerosol chemistry (AERO) in the diffusion equation can be resolved (Byun and Schere, 2006). Each term contributes to the rate of change of PM$_{2.5}$ level at the locations chosen in this study: the position #17 (Fig. 1) located between Bohai Sea and China Ease Sea, #18 (Fig. 1) located between China East Sea and Taiwan, #19 (Fig. 1) located in the middle of Taiwan Strait, #20 located in the northern South China Sea, BQ (#9 in Fig. 1) in northern Taiwan, ZM (#12 in Fig. 1) in central Taiwan, and CY (#13 in Fig. 1) in southwestern Taiwan. Those positions were chosen because they are on the path of northeast wind. Through the value of each term in the





process analysis, we can understand whether each term can produce or remove PM$_{2.5}$ at these positions and therefore realize the physical and chemical processes during LRT.

The Fig. 5(a-1) is quite similar to Fig. 5(a-2), which implies the contribution of PM$_{2.5}$ to #17 was mainly from BRIR. The main production process was HADV, followed by ZADV and VDIF and the removal process was mainly AERO. The removal process is likely caused by the evaporation of ammonium nitrate in PM$_{2.5}$ plume moving from high latitude regions to low latitude regions (Stelson and Seinfeld, 1982; Chuang et al., 2008b). In contrast, there was less PM$_{2.5}$ occasionally from YRDIR (Fig. 5(a-3)) and nearly none from PRDIR (Fig. 5(a-4)). It is expected because northeast wind prevails in winter, the BRIR and YRDIR/PRDIR are located at the upstream and downstream of #17, respectively. From Fig. 5(b-1)-(b-4), it is apparent that #18 was influenced by both the BRIR and YRDIR, mainly produced through non-uniform HADV, VDIF, ZADV, and CLDS; and removed through AERO and occasional HADV and DDEP processes, and almost unaffected by PRDIR. For #19, PM$_{2.5}$ was influenced mainly by BRIR and YRDIR but it was also influenced by PRDIR from 8th to 12nd (Fig. 5(c-1)-(c-4)), which has been verified to be related to the intrusion of cold surge and transboundary transport in last section (Fig. 4). The production from BRIR and YRDIR were mainly attributed to CLDS; and removal process was mainly AERO and secondly DDEP. The production and removal processes of PM$_{2.5}$ for #20 were very similar to #19 but slightly lower (Fig. 5(d-1)-5(d-4)) because it is farther from BRIR and YRDIR than #19. Although #20 is very near PRDIR, it was influenced more by YRDIR (Fig. 5(d-3)-5(d-4)) since the prevailing wind was mainly northeast wind in January. From above, it is found that the PM$_{2.5}$ plume transported southwards from BRIR or YRDIR in a three-dimensional path, i.e., horizontal and vertical advection, and vertical diffusion over Bohai Sea and China East Sea. During the southward transport, AERO was always the major removal process, i.e., evaporation of volatile species. When the plume transported to subtropical regions, cloud process became the major production process of PM$_{2.5}$. The reason was possibly the condensation in the mix of cold PM$_{2.5}$ plume from high latitude regions to warm air/sea at low latitude regions.

The production processes of PM$_{2.5}$ at BQ were mainly HADV with minor CLDS, and the removal processes were mainly ZADV with minor AERO (Fig. 5(e-1)). It suggests that the PM$_{2.5}$ plume transported in a mainly horizontal when it was close to and reached northern Taiwan. Moreover, each industrial region contributed PM$_{2.5}$ to BQ in very similar processes (Fig. 5(e-2)-(e-4)). The LRT from three industrial regions or nearby local sources transport to BQ through mainly HADV followed by CLDS. Minor PM$_{2.5}$ was formed in northern Taiwan probably due to the high relative humidity, which was probably induced by the cloud or fog produced by terrain uplifting. The PM$_{2.5}$ at BQ then transport up- and then southwards. Comparing Fig. (f-1) with Fig (f-2)-Fig (f-3), it is obvious that the PM$_{2.5}$ of ZM was produced more by local from vertical transport than BRIR or YRDIR, which only exerted less PM$_{2.5}$ along with the cold surge, and removed by horizontal transport. In other words, the PM$_{2.5}$ in upstream northern Taiwan was vertically advected and diffused southwards to central Taiwan and then horizontally advected to downwind areas. On the other hand, the influence from PRDIR was much less when the prevailing wind was northeast monsoon (Fig. 5(f-4)). However, when the cold surge passed Taiwan (Jan 8th and 10th), the influence from PRDIR could not be ignored, which has been illustrated in Fig 2(f), Fig. 4 and Fig. 5(f-4). The reason is that the downward motion of the transboundary transported PM$_{2.5}$ plume was enhanced by the subsidence of the cold surge anticyclone (Yen et al., 2013; Chuang et al., 2016). For CY located in southwestern Taiwan, the production processes of PM$_{2.5}$ were mainly VDIF and HADV, and the removal processes were mainly ZADV and AERO; however, occasionally when the production processes were ZADV and VDIF, the removal processes were HADV and AERO (Fig. 5(f-1)). Compared Fig. 5(f-2)-(f-4) and Fig. 5(g-2)-(g-4), it is obvious the production and removal processes for CY were very similar to for ZM. The impact from BRIR and YRDIR was less and mainly from local. When the cold surge passed Taiwan, PRDIR influenced PM$_{2.5}$ at CY as well.

### 3.3 The impact of PM$_{2.5}$ from the Chinese three major industrial regions in July 2017

The Fig. 6(a) and Fig. 6(b) reveals that the impact of BRIR on PM$_{2.5}$ in Taiwan was negligible. The monthly contribution was less than 0.01 μg m$^{-3}$ or less than 0.04% of total PM$_{2.5}$ on the western Taiwan. The influence from YRDIR and PRDIR on





Taiwan was equally small with BRIR (Fig. 6(c)-Fig. 6(f)). The daily contribution from BRIR was also small, highest on 25th

to 28th July with less than 0.1 µg m$^{-3}$ (Fig. 7(a-1)-(a-7)). The daily contribution of YRDIR on western Taiwan was only visible on 27th to 29th July (Fig. 7(b-1)-(b-7)), about 0.1-0.3µg m$^{-3}$. The influence of BRIR and YRDIR was less because they are located at the downstream of Taiwan when the southwest wind prevailed during most of July. As for PRDIR, its daily impact was detectable on 28th July but rose to 0.2-0.5 µg m$^{-3}$ on 30th to 31st July (Fig. 7(c-1)-(c-7)).

**3.4 The physical and chemical processes of LRT from the Chinese three major industrial regions to Taiwan in July**
**2017**

From Fig. 8(a-1) to Fig. 8(a-4), it is obvious that #17 was more influenced by YRDIR than BRIR and PRDIR in July 2017. On most days, the production process was AERO, followed by ZADV. In contrast, the removal process was mainly HADV. #17 was less influenced by the downstream BRIR except six days at the end of July 2017. The weather map appeared the circulation around #17 in the last few days was not like prevailing southwest wind but changeable (unshown). In addition to #17, all other
remaining specific locations were less influenced by three industrial regions under prevailing southwest wind, as illustrated in Fig. 8(b)-Fig. 8(f). However, the #19 and #20 were obviously influenced by YRDIR and PRDIR on the last two days of July. The production process from YRDIR and PRDIR on #19 was mainly CLDS and HADV, followed by VDIF. While the removal process was mainly AERO (Fig. 8(c)). The production and removal processes on #20 was opposite for those days, which implies the wind field has changed (Fig. 8(d)). The weather map revealed that there was a thermal low near Taiwan at the end
of July (Fig. S2.2). Obviously, this thermal low could transport less pollutants from three industrial regions and have a smaller influence on PM$_{2.5}$ at BQ (Fig. 8(e-2)-(e-4)), ZM (Fig. 8(f-2)-(f-4)), and CY (Fig. 8(g-2)-(g-4)) depending on the distance . From Fig. 8(e-1)-(g-1), it is found that the production process of PM$_{2.5}$ at BQ and CY was mainly HADV and the removal processes was mainly ZADV. While for city ZM in central Taiwan, the major production and removal processes were ZADV and HADV before July 13. For the second half of July, the major production process was HADV and the removal processes
were ZADV and AERO. It seems that there was transport between each other and caused different production and removal processes in cities. To assess the transport between cities was a very difficult issue and was beyond the present study. In short, during the period of prevailing southwest wind, the influence of BRIR, YRDIR, or PRDIR could be ignored unless there was special weather system like the aforementioned thermal low which could transport less PM$_{2.5}$ from distant sources.

**3.5 Analysis of the episodes occurring on 13th and 9th January 2017**

On January 13th 2017, the Asian anticyclone transported pollutants from Asian continent to Taiwan and caused high PM$_{2.5}$ episodes. Such LRT events occurred at a weather pattern as illustrated in Fig. 9. Although the impact of LRT on Jan 13th was less than Jan 8th, 9th, 20th or 22nd (Fig. 3), the physical and chemical processes during transport were similar for these days since the weather patterns were quite analogous to each other. The Asian anticyclone was moving from East Asian to the West Pacific. The peripheral circulation of the Asian anticyclone was the strong northeast wind on coastal areas and the sea. It was
found the northeast wind formed lee wakes in southern Taiwan where PM$_{2.5}$ accumulated (Fig. 10(a)-(b)). When the leading edge of Asian anticyclone arrived, the wind speed increased and therefore enhanced the dispersion of PM$_{2.5}$ in southern Taiwan (Fig. 10(c)-(e)). Subsequently, the LRT haze arrived (Fig. 10(f)) and split to the east and west side of Taiwan due to the blocking of mountains, more on the west side. (Fig. 10(g)-(i)).

Fig. 11(a-1)-(a-4) shows that the influence of BRIR on #17 was more than YRDIR and PRDIR on Jan 13th, since BRIR is
located at the upstream of #17 under northeast wind. The major production process was VDIF below 760 m (layer4) and AERO above 760 m. It implies the transport path from BRIR to #17 is vertical. The removal process was AERO below 760 m and VDIF above. It suggests the ascent and subsidence of air parcels might enhance the formation and removal of aerosol in upper and lower level, respectively. Although #17 was slightly influenced by YRDIR, the contribution of different processes YRDIR on #17 was less and non-uniform (Fig. 11(a-3)). The contribution of different processes from PRDIR to #17 was also non-





uniform and even less (Fig. 11(a-4)). From Fig. 11(b-1)-(b-4), it was found that #18 was mainly influenced by YRDIR on Jan 13th. The major production processes below layer 9 (~310 m) were HADV, VDIF, and ZADV and removal processes were DDEP and AERO (Fig. 11(b-3)). #18 was slightly influenced by BRIR with major production process were VDIF and ZADV and removal process was AERO (Fig. 11(b-2)). On Jan 13th, #19 and #20 were less influenced by all industrial regions (Fig. 11(c-2)-(c-4), Fig. 11(d-2)-(d-4)). It implied that #19 was possibly influenced by nearby Fujian province on the north and west

side of Taiwan Strait. On Jan 13th, #20 was also less influenced by three industrial regions probably due to BRIR and YRDIR was distant and PRDIR was located at the downstream of #20. Comparing Fig. 11(e-1) and Fig. (e-2)-(e-4), it was found the BQ was much influenced by YRDIR. The major production process below 200 m (layer 7) was HADV, followed by AERO and above 200 m was either of VDIF, ZADV, CLDS and mixed of them. The major removal process was ZADV followed by VDIF below 200 m but HADV and AERO above. BQ was less influenced by BRIR due to long distance, deviation of wind

direction and by PRDIR due to BQ is located at upstream of PRDIR. In this event, ZM and CY were less influenced not only by BRIR and PRDIR but also YRDIR (Fig. 11(f-1) - Fig. (g-4)). It explains the haze plume passed BQ and then transported to the west coast of Taiwan instead the inland ZM and CY.

The PM$_{2.5}$ event occurring in western Taiwan on Jan 9th was similar to that on Jan 13th, which were both LRT of EAH. However, there were still slightly differences between these two events. First, the impact of three industrial regions on PM$_{2.5}$

in western Taiwan was much higher on Jan 9th than Jan 13th. Second, for the haze from BRIR and YRDIR, the production and removal processes on BQ were mainly HADV/AERO and ZADV/VDIF below 200 m (layer 7, Fig. 11(e-3)) on Jan 13th and less different processes at different layers above 200 m. While on Jan 9th, the major production processes at BQ was mainly HADV below 380 m (layer 10), AERO between 120 to 900 m (layer 5 to 15), and ZADV/CLDS between 650 to 1500 m (layer 13 to 19), as illustrated in Fig. S2.3(e-2)-(e-3). The removal process was mainly ZADV below 460 m (layer 11),

HADV between 550 to 900 m (layer 12 to layer 15), and HADV/AERO between 1000 to 1300 m (layer 16 to 18). Third, the stronger event occurring on Jan 9th has more obvious impact on ZM and CY than that on Jan 13th. The higher production of HADV near surface on Jan 9th explains the rapid moving EAH. In contrast, the higher production of AERO near surface occurring on Jan 13th explains slow moving EAH had time to react with the local pollutants, e.g. HNO$_3$ in Asian plume reacted with local NH$_3$ to form NH$_4$NO$_3$, which has been discussed in Chen et al. (2014).

**3.6 Analysis of the episodes occurring on 18th and 30th July 2017**

Around the East Asia and West Pacific regions, it usually prevails southwest wind in summer (from June to August). The dominating weather system is often the southwest monsoon, Pacific high and occasional thermal lows/typhoons. For example, on 18th July 2017, the surface weather map Fig. 12 illustrates that Taiwan was located on the west edge of the Pacific high. Meanwhile, the prevailing wind was the south wind around Taiwan. Although it prevailed south wind at this period, the local

thermal circulation also influenced the local transport. The PM$_{2.5}$ distributed over the western Taiwan and Taiwan Strait (Fig. 13). The major sources were from the urban and industrial sources in western Taiwan and the coastal area of Fujian province of Mainland China (Fig. 13).

From Fig. 14(a-1) to Fig. 14(a-4), it was found that #17 was mainly influenced by YRDIR other than BRIR or PRDIR on July 18th 2017. The production and removal processes were non-uniform below 80 m (layer 4). But from 120 m to 460 m (layer 5

to layer 11), the major production processes were AERO and ZADV and the removal process was mainly HADV. The Fig. 14 shows that the influence of three industrial regions on #18, #19, #20, BQ, ZM, or CY was almost ignorable. It suggested the PM$_{2.5}$ was mainly from local pollution on July 18th. On the other hand, the #19, #20, BQ, ZM, and CY was influenced by PRDIR at the end of July 30th (Fig. 8). As mentioned earlier, the thermal low over Taiwan Strait (Fig. S2.2) caused unstable wind field and transported pollutants from coastal areas of Asian continent to northern South China Sea and Taiwan strait

(Fig.S2.4). In this case, there is hardly amount of PM$_{2.5}$ transported from three industrial regions to those specific locations on July 30th except from PRDIR to BQ, as illustrated in Fig. S2.5.





### 3.7 Discussion of the chemical compositions and emissions

Lee et al. (2017) conducted PM$_{2.5}$ sampling at BQ, ZM, and CY every six days in 2017. The sampling of Jan 13th was used to compare with simulated PM$_{2.5}$ compositions, as indicated in Fig. 15 The previous studies (Chuang et al., 2008b;Wang et al.,

2016) suggested it took about 28 hours for the PM$_{2.5}$ haze transported from Yangtze River estuary to the northern tip of Taiwan island. Therefore, the simulated PM$_{2.5}$ compositions at #17 and #18 on Jan 12th were also illustrated. According to the main content, BRIR and YRDIR were the major sources of #17 - #20. As illustrated in Fig. 15, no matter on Jan 12nd or Jan 13th, the major compositions were sulfate and OC for #17 - #20. However, the proportion of nitrate in PM$_{2.5}$ at #17 on Jan 12th was higher than those at #18, #19, and #20 on Jan 13th. It explains the nitrate would evaporate from aerosol phase to gas phase for

PM$_{2.5}$ plume transported from high to low latitude regions (Chuang et al., 2008b). The proportions of Na$^+$ and Cl$^-$ in PM$_{2.5}$ at # 19 and #20 were higher than those at #17 and #18. The higher sea salt due to stronger wind speed is expected because the Taiwan Strait was a wind tunnel between Central Mountain Range in Taiwan and Wuyi Mountain Range in Fujian province (Lin et al., 2012). In addition, the proportions of nitrate in PM$_{2.5}$ at BQ, ZM, and CY were higher than those over #17 - #20. That should be caused by the local pollution. The comparison between simulation and observation indicated that the

performance of simulation was not bad. The simulated proportion of nitrate and ammonium in PM$_{2.5}$ was slightly higher than the observations. While the simulated proportion of K$^+$, Ca$^{2+}$, Mg$^{2+}$, Na$^+$ was slightly underestimated. This suggested the emission of biomass burning and wind-blown dust over Taiwan island and the influence of sea salt still have room for improvement.

We also compared the simulated PM$_{2.5}$ compositions with observations on July 18th 2017 (Fig. 16). As mentioned in main

content, #17 was influenced by upstream YRDIR, the proportion of nitrate in PM$_{2.5}$ #17 was higher than further upstream #18, #19, and #20. The proportion of nitrate in PM$_{2.5}$ at #19 and #20 was higher than #18, it implies #19 and #20 were influenced more by PRDIR than #18. For BQ, ZM, and CY, the proportion of simulated OC in PM$_{2.5}$ was slightly overestimated as compared with observation but nitrate, sulfate and others were underestimated. Since BQ, ZM, CY were less influenced by PRDIR on July 18th, the overestimation of OC and underestimation of nitrate should be related to the bias of local emission

inventory. In addition to local emission inventory, the underestimation of sulfate could possibly be related to underestimation of emission from ships around Taiwan since the local emission of SO$_2$ is quite low. Moreover, the uncertainty of emission in the Southeast Asia is also another issue that needs to be improved.

On July 30th 2017, there was a thermal low which influence the circulation near Taiwan. The Fig. S2.6 illustrates that BQ, ZM, and CY were influenced by local pollution and therefore the proportions of EC and NH$_4^+$ in PM$_{2.5}$ at these three cities

were higher than #17 - #20. It was not easy to form nitrate at BQ, ZM, and CY since the circulation was strong and cloud cover was intense (no PM$_{2.5}$ sampling on July 30th due to bad weather condition).

### 4. Conclusions

This study evaluated the impact of the three biggest industrial regions in Asian continent on PM$_{2.5}$ in Taiwan and discussed the process analysis during transport. It applied the CMAQ model with BFM method and process analysis technique. The

simulation period was January and July 2017.

In January 2017, the LRT from Asian continent to Taiwan was substantial over northern Taiwan and gradually minor in central and southern Taiwan. The impact of monthly PM$_{2.5}$ from BRIR and YRDIR on Taiwan was 0.7-1.1 µg m$^{-3}$ and 1.2-1.9 µg m$^{-3}$, about 5% and 7.5% of total concentration, respectively. The daily impact was the most on January 9. The production of BRIR and YRDIR was 6-8 and 9-12 µg m$^{-3}$, respectively. In contrast, the influence of PRDIR to Taiwan was ignorable.

However, when the cold surge passed Taiwan, the PM$_{2.5}$ from PRDIR can influence Taiwan with monthly average impact of about 0.5 µg m$^{-3}$ via transboundary transport and boundary layer mixing (VDIF). When the cold surge induced-events occurred, the impact from BRIR and YRDIR was substantial on BQ. The transport mechanism of EAH from BRIR and YRDIR was



horizontal (HADV) and vertical (ZADV and VDIF) at Bohai Sea and China East Sea. When the EAH moved to Taiwan Strait and northern South China Sea, CLDS became the major production of PM$_{2.5}$ under high relative humidity environment. Along the transport, AERO and DDEP were always the removal process for the EAH transporting from high latitude regions to low latitude regions. When the EAH moved to northern Taiwan, HADV and AERO were the major production processes of PM$_{2.5}$ at BQ. The transport mechanism from northern Taiwan to central Taiwan and southern Taiwan was changeable due to complex terrain and complex land canopy. In addition, the intensity of EAH would have different production and removal processes in different height. The stronger the intensity of EAH, the impact on central and southern Taiwan was more obvious, the proportion of HADV in PM$_{2.5}$ production was more obvious near surface.

In July 2017, the influence from three industrial regions on the PM$_{2.5}$ was ignorable in Taiwan, i.e. PM$_{2.5}$ was mainly come from upwind adjacent local sources unless if there was special weather system, e.g. a thermal low nearby which may carry pollutants from PRDIR to Taiwan but it is a minor.

In regards of performance of MIX emission inventory, this study compared the simulated and observed PM$_{2.5}$ compositions on Jan 13th, July 18th, and July 30th. The simulated proportion of nitrate and ammonium in PM$_{2.5}$ during the winter time was slightly overestimated but the simulated K$^+$, Ca$^{2+}$, Mg$^{2+}$, Na$^+$ was underestimated at BQ, ZM, and CY. It suggested the bias in the local emission inventory has lacked the correct information of local biomass burning. During the summertime, the simulated proportion of OC in PM$_{2.5}$ was overestimated but underestimated for nitrate, sulfate, and others. In addition to the bias of local emission inventory, the LRT emission of sulfate is another reason that caused the difference.

**Author contribution**

Ming-Tung Chuang designed the experiment, carried out most part of the study and wrote the original draft.

Maggie Chel Gee Ooi helped produce half of figures and revised the manuscript.

Neng-Huei Lin is the project leader, provided consultation and acquired the financial support for this study.

Joshua S. Fu submitted valuable questions and helped enhancement of the writing.

Chung-Te Lee provided the PM$_{2.5}$ compositions data and provided related consultation.

Sheng-Hsiang Wang provided beneficial consultation according to his previous publications.

Ming-Cheng Yen provided beneficial consultation according to his previous publications.

Steven Soon-Kai Kong helped part of the post processing of the simulation results.

Wei-Syun, Huang helped maintenance of the computing machine.

**Acknowledgements**

We express deep gratitude for the support from the Taiwan Environmental Protection Agency (EPA-105-FA18-03-A215, EPA-106-FA18-03-A215, and EPA-107-FA18-03-A215). The researchers also acknowledge contributions from the U.S. National Center for Environmental Prediction and the Data Bank for Atmospheric & Hydrologic Research (managed by the Department of Atmospheric Science of Chinese Cultural University and sponsored by Ministry of Science and Technology) for input data used in the meteorological modeling and for monitoring data in the evaluation of meteorological modeling.

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






**Table 1 The performance of meteorological modeling results for the present study**

| | | Temperature | | | Wind speed | | | Wind direction | |
|---|---|---|---|---|---|---|---|---|---|
| | | MB (°C) | MAGE (°C) | IOA | MB (m s$^{-1}$) | MAGE (m s$^{-1}$) | IOA | WNMB | WNME |
| Standard | | ±1.5 | <3 | >0.7 | ±1.5 | <3 | >0.6 | ±10% | <30% |
| PJY | Jan | 1.54 | 1.63 | 0.90 | -0.01 | 1.16 | 0.91 | -2.09 | 5.91 |
| | July | 0.43 | 1.18 | 0.69 | 0.05 | 1.29 | 0.93 | 0.00 | 4.27 |
| TPE | Jan | 0.00 | 0.60 | 0.99 | -0.75 | 1.10 | 0.74 | 8.91 | 13.16 |
| | July | -0.31 | 0.98 | 0.91 | -0.06 | 0.92 | 0.81 | 5.71 | 22.04 |
| CP | Jan | 0.12 | 0.61 | 0.98 | 0.52 | 0.86 | 0.84 | 2.70 | 13.85 |
| | July | -0.02 | 0.73 | 0.95 | 0.16 | 0.68 | 0.80 | 4.50 | 19.01 |
| TC | Jan | 0.17 | 1.02 | 0.96 | 0.06 | 0.47 | 0.87 | 3.16 | 41.33 |
| | July | 0.61 | 1.19 | 0.92 | 0.05 | 0.56 | 0.80 | 6.84 | 25.30 |
| CY | Jan | 0.05 | 0.83 | 0.98 | -0.21 | 0.61 | 0.83 | 12.34 | 32.40 |
| | July | 0.02 | 1.06 | 0.93 | -0.35 | 0.83 | 0.78 | 5.61 | 21.18 |
| TN | Jan | 0.18 | 0.83 | 0.97 | -1.82 | 1.84 | 0.52 | 9.42 | 20.26 |
| | July | -0.14 | 0.85 | 0.93 | -0.97 | 1.12 | 0.69 | -1.33 | 20.76 |
| KH | Jan | -0.07 | 0.94 | 0.93 | 1.15 | 1.26 | 0.60 | 4.22 | 23.40 |
| | July | -1.27 | 1.47 | 0.66 | 1.19 | 1.56 | 0.73 | 4.84 | 12.81 |
| HC | Jan | -1.29 | 1.39 | 0.88 | 2.17 | 2.31 | 0.80 | -0.60 | 7.39 |
| | July | -0.79 | 1.13 | 0.90 | 1.88 | 1.96 | 0.66 | 1.01 | 8.58 |

Note: The standard of statistical evaluation is based on Emery (2001) and TEPA (2016).





**Table 2 Simulated PM$_{2.5}$ at eight air quality stations in western Taiwan**

|  |  | MB | MFB (%) | MFE (%) | R | IOA |
|---|---|---|---|---|---|---|
|  |  |  | <±65 | <85 | >0.5 | >0.6 |
| BQ | Jan | 5.0 | 10% | 38% | 0.85 | 0.82 |
|  | July | 5.3 | 40% | 49% | 0.46 | 0.55 |
| PZ | Jan | 5.1 | 9% | 38% | 0.71 | 0.68 |
|  | July | 3.2 | 17% | 29% | 0.63 | 0.67 |
| ML | Jan | 0.2 | -17% | 42% | 0.73 | 0.77 |
|  | July | 4.8 | 22% | 40% | 0.76 | 0.65 |
| ZM | Jan | 5.5 | 12% | 29% | 0.82 | 0.83 |
|  | July | 3.3 | 16% | 33% | 0.68 | 0.76 |
| CY | Jan | -2.6 | -10% | 23% | 0.69 | 0.80 |
|  | July | 0.3 | 5% | 30% | 0.52 | 0.70 |
| TN | Jan | 0.5 | -2% | 22% | 0.64 | 0.77 |
|  | July | 7.4 | 46% | 46% | 0.69 | 0.68 |
| ZY | Jan | 1.1 | 1% | 17% | 0.67 | 0.79 |
|  | July | 1.7 | 12% | 35% | 0.52 | 0.72 |
| HC | Jan | -4.1 | -62% | 77% | 0.14 | 0.43 |
|  | July | 0.4 | -18% | 53% | 0.19 | 0.26 |

Note: the standard of statistical evaluation is based on Emery (2001) and TEPA (2016).



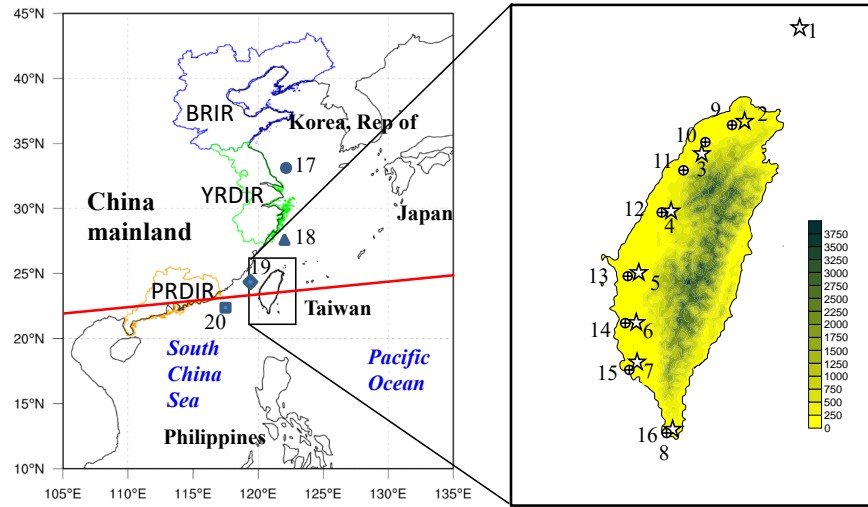

**Figure 1: Geographic location of three major industrial regions (BRIR (blue line enclosed region), YRDIR (green) and PRDIR (orange)) in East Asia and meteorological and air quality stations in Taiwan. Meteorological stations: #1: PJY, #2: TPE, #3: CP, #4: TC, #5: CY, #6: TN, #7: KH, and #8: HC; air quality stations: #9: BQ, #10: PZ, #11: ML, #12: ZM, #13: CY, #14: TN, #15: ZY, and #16: HC. The circular, triangle, diamond, and rectangular symbols are #17, #18, #19, and #20, respectively. The red line is the cross-section plot for Figure 4**



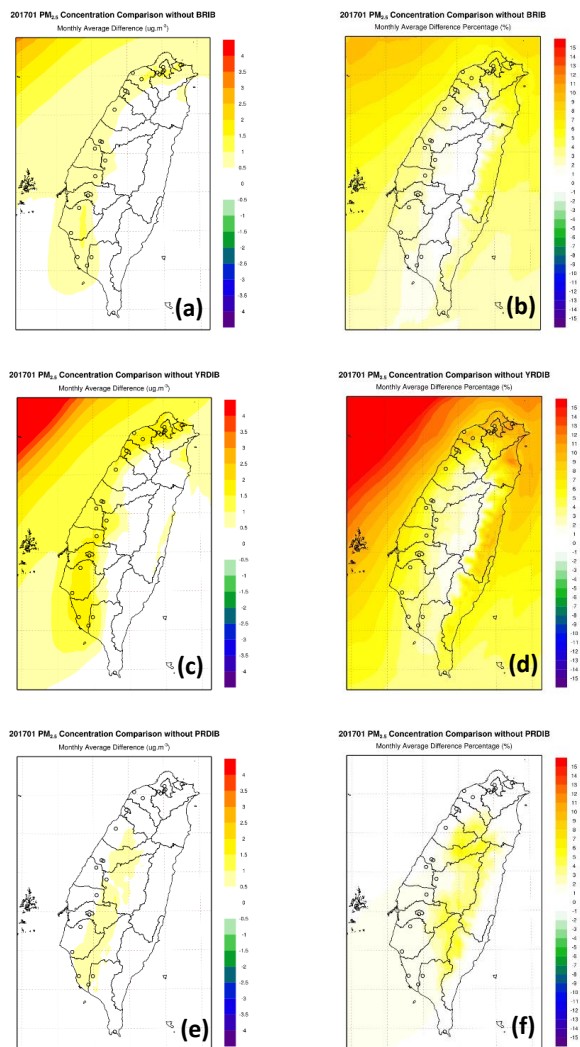


**Figure 2: The monthly average impact of PM₂.₅ from BRIR: concentration (a) and percentage (b);YRDIR: concentration (c) and percentage (d);PRDIR: concentration (e) and percentage (f) on Taiwan in January 2017**





**Figure 3: The daily average impact of PM$_{2.5}$ from BRIR, YRDIR, PRDIR on air quality stations in Taiwan in January 2017**





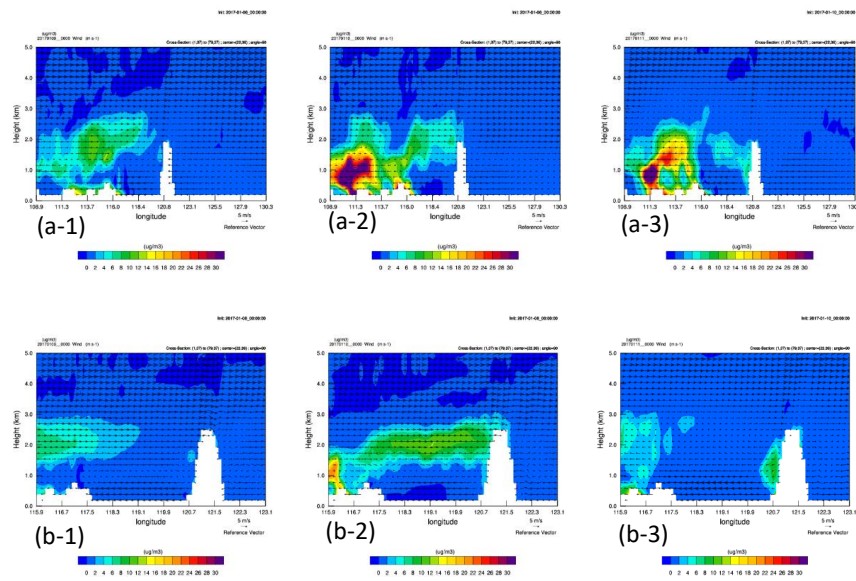


**Figure 4: Cross-section plot of PM$_{2.5}$ along the red line of Fig. 1 at 08:00 LT (Local Time) on Jan 9th (a-1), 08:00 LT on Jan 10th (a-2), 08:00 LT on Jan 11th (a-3) of domain 2 for Base case minus Prdir case. Synchronized plots for domain 3 are (b-1) to (b-3)**





Figure 5: The daily contributions of individual processes to the concentrations of PM₂.₅ in January 2017, a,b,c,d,e,f, and g represent #17, #18, #19, #20, BQ, ZM, and CY, respectively ; 1, 2, 3, and 4 represent influence of total emissions, BRIR, YRDIR, and PRDIR, respectively




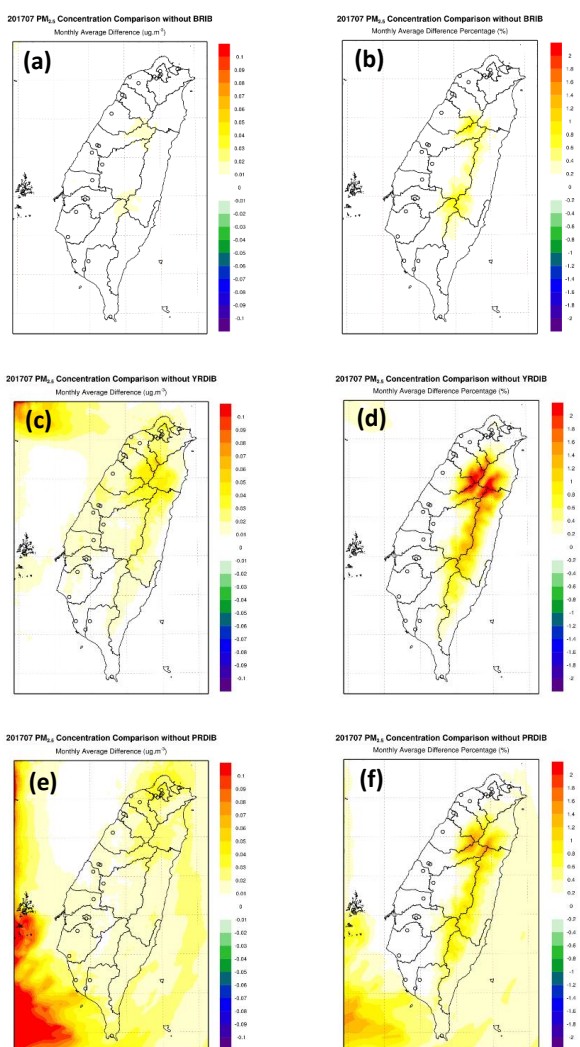

**Figure 6: The monthly average impact of PM$_{2.5}$ from BRIR: concentration (a) and percentage (b);YRDIR: concentration (c) and percentage (d);PRDIR: concentration (e) and percentage (f) on Taiwan in July 2017**





**Figure 7:** The daily average impact of PM$_{2.5}$ from BRIR, YRDIR, PRDIR on air quality stations in Taiwan in July 2017



**Figure 8: The daily contributions of individual processes to the concentrations of PM$_{2.5}$ in July 2017, a,b,c,d,e,f, and g represent #17, #18, #19, #20, BQ, ZM, and CY, respectively;1, 2, 3, and 4 represent influence of total emissions, BRIR, YRDIR, and PRDIR, respectively**



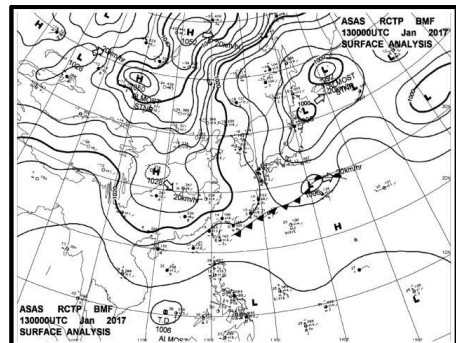

**Figure 9: The surface weather map on 08:00 LT Jan 13th 2017**


**Figure 10:** The every 3 hour simulated wind vector and PM$_{2.5}$ distribution on the event at 00:00 LT (a) 03:00 LT (b) 06:00 LT (c) 09:00 LT (d) 12:00 LT (e) 15:00 LT (f) 18:00 LT (g) 21:00 (h) Jan 13th and 00:00 LT (i) Jan 14th 2017








**Figure 11: The hourly average contribution of physical process at each layer on Jan 13th 2017, a,b,c,d,e,f, and g represent #17, #18, #19, #20, BQ, ZM, and CY, respectively;1, 2, 3, and 4 represent influence of total emissions, BRIR, YRDIR, and PRDIR, respectively**



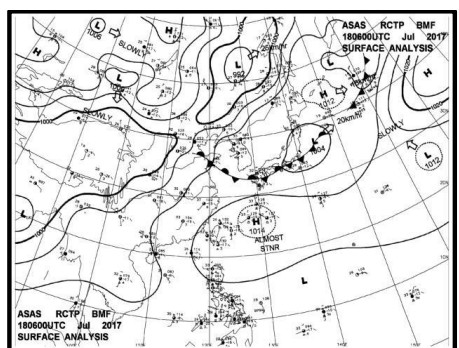


**Figure 12: The surface weather map on July 18th 2017**





**Figure 13: The every 3 hour simulated wind vector and PM$_{2.5}$ distribution at 00:00 LT (a) 03:00 LT (b) 06:00 LT (c) 09:00 LT (d) 12:00 LT (e) 15:00 LT (f) 18:00 LT (g) 21:00 (h) July 18th and 00:00 LT (i) July 19th 2017**







**Figure 14: The hourly average contribution of physical process at each layer on July 18th 2017, a,b,c,d,e,f, and g represent #17, #18,**
**#19, #20, BQ, ZM, and CY, respectively;1, 2, 3, and 4 represent influence of total emissions, BRIR, YRDIR, and PRDIR,**
**respectively**





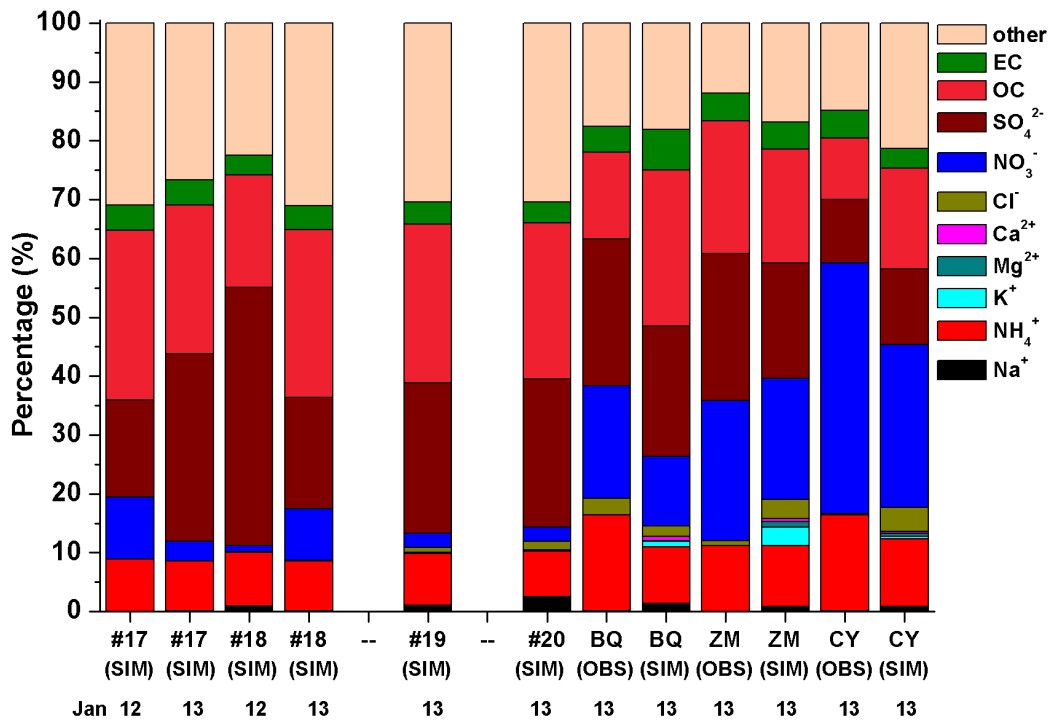

**Figure 15: The comparison of simulation (SIM) and observation (OBS) of PM$_{2.5}$ compositions at #17-#20 and BQ, ZM, and CY on Jan 12th and 13th 2017**






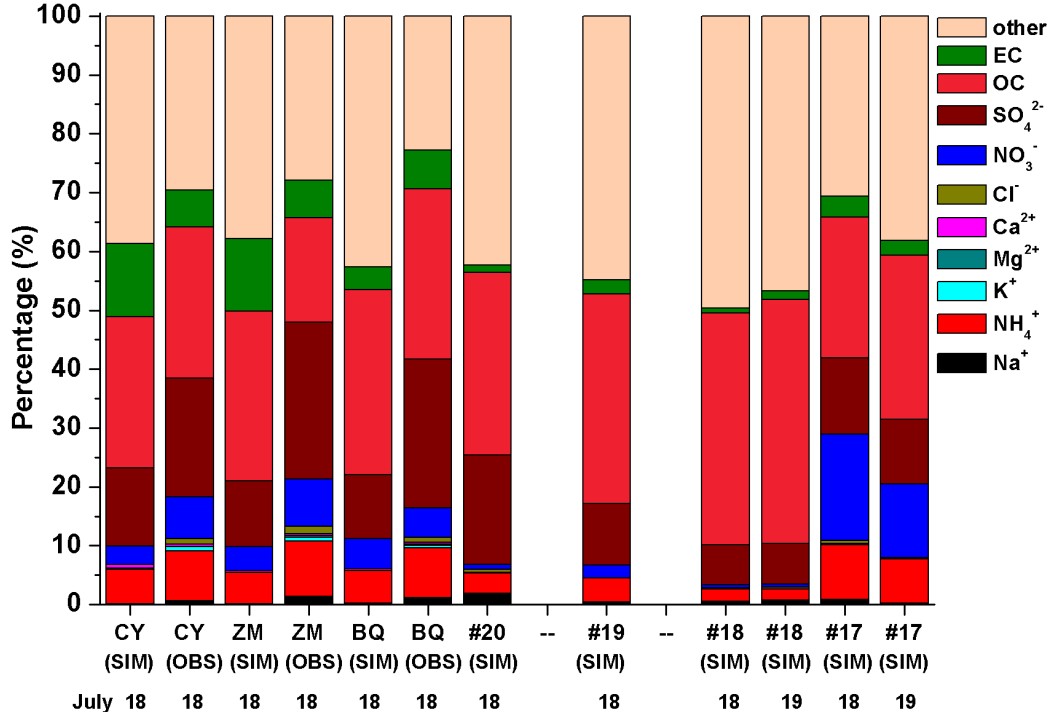

**Figure 16: The comparison of simulation (SIM) and observation (OBS) of PM$_{2.5}$ compositions at #17-#20 and BQ, ZM, and CY on July 18th and 19th 2017**
