# Peer review of "Study on the impact of three Asian industrial regions on PM2.5 in Taiwan and the process analysis during transport"

_Atmospheric Chemistry and Physics, 2019_

## Referee Comment (RC1) · Anonymous Referee #1 · 17 Dec 2019

General Comment

The manuscript focuses on the impact of three industrial regions in China on the PM2.5 pollution in Taiwan for the winter and summer in 2017 by using the WRF/CMAQ modeling system. The authors demonstrated the net impact of each industrial region as well as the different roles of different physical and chemical processes in detail. Some evaluation of model output through comparison with observations were also provided. The topic of the manuscript is suitable for ACP, it presents original data, and might give a useful information for East Asian Haze studies in Taiwan. However, it requires some major revisions (see my major concerns below) to be accepted. The authors

should also address some other specific remarks (listed below). Moreover, several grammatical issues can be found, so English proofreading is strongly recommended before revised submission.

Major Concerns:

1) The manuscript shows the analysis both for January and July. However, the impacts of three industrial regions on Taiwan in summer (July) is quite small, almost negligible even in the last few days when the impacts were relatively large. I don't think it is worthwhile spending much space for the July analysis, rather focusing on winter case would make the paper more concise and scientifically focused.

2) The results of process analysis was described and discussed in 3.2, 3.4, 3.5, and 3.6, which formed a main part of this paper. However, the descriptions in these sections were not firmly reasoned. In these sections, the author argued "dominant" contribution of three industrial regions at some locations. For example, in 3.2, the author pointed out that PM2.5 was influenced "mainly" by BRIR and YRDIR at the place #19. However, these arguments were not convincing. For the abovementioned example, Fig5 (c-2) and (c-3) which was regarded as representing the contributions by process of BRIR and PRDIR, respectively, showed similar variations to those of total contributions shown as Fig5 (c-1). However, the range of values largely differed each other, so I cannot understand why the author can conclude that the BRIR and YRDIR were "main" contributors to the variation of PM2.5 at #19. Similar arguments to this case can be found in these sections, and they considerably deteriorate the persuasiveness of the manuscript. I strongly recommend the author to revise such arguments in these sections and provide how to read and understand the main figures (Fig 5, 8, 11, and 14).

Specific Comments:

- L37: Seasonality of EAH is not "due to" rapid economic grows in Asian countries.

- L43-45: Why did you specify these data and models for trajectory analysis?

-L50-51: Could you state more clearly why TS method would contain substantial uncertainty?

- L54: The difference between those two runs does not directly mean the contribution of specific source but impact of the reduction of that specific source. To distinguish these two concepts is quite important.

- L56-58: What do you mean "under-represented chemical reaction" here? Could you explain more specific?

- L67: CTM? This should be AM method?

- L87: These abbreviations (LRT, LP) have already been defined.

- L90: Meaning of these terms (LRT-Event and so on) should be explained.

- L98-99: Are power and industrial sectors the largest for entire Asia or any specific region in Asia?

- L103-104: This should be "the impact of reduction in source emission in each industrial region", because BFM can estimate "impact" not "contribution". Or you can define the wording that you will use the word "contribution" for the deference between control runs and sensitivity run.

- L123-127: For Figure1, the formal, not abbreviated, names for each monitoring station should be appeared here.

- L130-131: Why you don't show the model domains in Figure 1 but just describe horizontal resolution?

- L146: "MB" has already been defined in the previous sentences.

- For the evaluation of WRF and CMAQ shown in Table 1 and 2, the results from which domain were used? And in addition to the summary of statistical indices in Table 1,

figures of comparisons of temperature and wind between observation and simulation are quite informative. Could you put them together at least as supplement?

- You should explain how you draw Fig3. Are the values in Fig3 difference between Base case and sensibility case? If so, it's better to note it in the manuscript or in figure caption. Fig3 is a bit busy, so it seems better to select fewer locations out of seven to avoid redundancy.

- L176: Remove unnecessary "the".

- Could you check the wording "China East Sea"? "East China Sea" has been also used for the same area in many literatures.

- For Figure5, you should explain how to deduce the values shown in the figure, in particular the values in Fig5(*-2,3,4). Are they the difference between Base case and sensitivity case? If so, you should instruct briefly how to interpret these Figures. Is the title of y-axis correct? This should be "$\Delta$concentration" or "daily concentration change"?

- L204: Fig5(a-1) and (a-2) do not seem quite similar to each other. Could you specify more about which features of both figures look similar?

- L204: You concluded that main contributor to #17 PM2.5 is BRIR, but I cannot understand why you can conclude like this. The values in Fig5(a-1) and (a-2) are quite different. You should give an instruction how to read and understand the Fig5.

- L205: Can HADV process "produce" PM2.5? The term "production" here is not appropriate.

- L211: What process considered in AERO can reduce PM2.5?

- L213: If the intrusion of PM2.5 from PRDIR is like that depicted in Fig4, why the contribution of ZADV is not so large in Fig5(c-4)? Since #19 is located between PRDIR and Taiwan island and the transport of PM2.5 between them occurs about 1-2 km high

above the surface as in Fig4, any kind of vertical (downward) motion should transport PM2.5 from that layer to the location of #19 which must be at the surface.

- L227: What does "minor PM2.5" means here?

- L228: Why can you describe "The PM2.5 at BQ then transport up- and then south-wards"? Which figure show this transport of PM2.5?

- L228-229: Fig.(f-1) -> Fig5 (f-1)

- L234-235: If this is true, why ZADV in Fig5 (f-4) is largely negative from Jan 8 to 10?

- L256: Why did you exclude Fig.8(a)?

- L267: Could you put the prevailing wind vector in Figures 2 and 6, otherwise I can not verify what you described here and similar descriptions in the manuscript explaining the impact of wind patterns.

- L280: Layer4? Is this Layer14?

- L281: It is apparent that only vertical motion can not transport PM2.5 from BRIR to #17. What do you mean here?

- L282-283: Why does ascent (descent) motion enhance (decrease) aerosol formation? What processes are involved ?

- L291: Fig. (e-2)-(e-4) -> Fig11. (e-2)-(e-4).

- L293: mixed -> mixture

- L340: higher -> lower?

- L341: underestimated -> overestimated?

- L353: There is not Fig.S2.6 in the supplement.

- L380: There is no comparison for July 30th (no Fig. S2.6).

---

## Author Comment (AC1) · 18 Feb 2020

Dear reviewer#1 and editor:

On behave of all co-authors, we are really grateful to reviewer#1 who spent much time reviewing the original manuscript. I know we have made some misleading narratives and have modified those in the revised manuscript. After all reviewers' comments are replied, the authors will an English language editing company to revise the manuscript. We know it is impossible but we hope the final manuscript could be an article written by a native English writer. In this response, we have attached three files: the manuscript of the main context, the supplement, and the one-to-one reply. We welcome any further

comments from reviewer#1. Thanks.

Best regards.

Ming-Tung Chuang

---

## Author Comment (AC2) · 18 Feb 2020

Dear reviewer#1 and editor:

On behave of all co-authors, we are really grateful to reviewer#1 who spent much time reviewing the original manuscript. I know we have made some misleading narratives and have modified those in the revised manuscript. After all reviewers' comments are replied, the authors will an English language editing company to revise the manuscript. We know it is impossible but we hope the final manuscript could be an article written by a native English writer. In this response, we have attached three files: the manuscript of the main context, the supplement, and the one-to-one reply. We welcome any further

comments from reviewer#1. Thanks.

Best regards.

Ming-Tung Chuang

———————————————————
**Response to Reviewers**

**Manuscript *acp-2019-762***

*We greatly appreciate the insightful comments and suggestions of the reviewers. Below please find a list of the Reviewers' remarks in contrast to our responses to them:*

**Review #1**

| Major Concerns | Responses |
|---|---|
| 1) The manuscript shows the analysis both for January and July. However, the impacts of three industrial regions on Taiwan in summer (July) is quite small, almost negligible even in the last few days when the impacts were relatively large. I don't think it is worthwhile spending much space for the July analysis, rather focusing on winter case would make the paper more concise and scientifically focused. | First, the authors really appreciate the reviewer spend much time and efforts reviewing this manuscript carefully and giving valuable opinions. They are truly grateful for the reviewer's comments which are very helpful to make this manuscript better. The authors admitted that they accidentally used non-precise and inappropriate words and so as to make misleading narratives. They promised that they will ask an English language editing company to revise the manuscript after all reviewers' comments are responded.

 Yes, the authors agree with the reviewer's suggestions and have cut down the contents of July analysis. They only keep the original Fig. 6 (Fig. 10 in the revised manuscript) in the main content and moved original Fig. 7 and Fig. 8 to Fig. S2.6 and Fig. S2.8 in the supplement of revised manuscript), and delete original Fig. 12, Fig. 13, Fig. 14, and Fig. 16. |
| 2) The results of process analysis was described and discussed in 3.2, 3.4, 3.5, and 3.6, which formed a main part of this paper. However, the descriptions in these sections were not firmly reasoned. In these sections, the author argued "dominant" contribution of three industrial regions at some locations. For example, in | Yes, the authors have written several misleading narratives in the original manuscript. After careful checking, first, they have modified the arrangement of the manuscript such that they combined the section 3.3, 3.4 and 3.6 of the original manuscript to section 3.5 of the revised manuscript in order to cut down the content of July analysis, and separate section 3.5 of the original manuscript into section 3.4 and 3.5 in the revised manuscript. Therefore, Fig. 11 of the original manuscript was changed to Fig. 9 and Fig. 14 was deleted. Now the main part is section 3.2, 3.4, and 3.5 for the revised manuscript.

 Second, they revised misleading narratives in order to avoid the |

**Fig. 1.** one-to-one reply
**Study the impact of three Asian industrial regions on PM$_{2.5}$ in Taiwan and the process analysis during transport**

Ming-Tung Chuang[1], Maggie Chel Gee Ooi[2], Neng-Huei Lin[2], Joshua S. Fu[3], Chung-Te Lee[4], Sheng-Hsiang Wang[2], Ming-Cheng Yen[2], Steven Soon-Kai Kong[2], Wei-Syun, Huang[2]

5  [1]Research Center for Environmental Changes, Academia Sinica, Taipei 11529, Taiwan
[2]Department of Atmospheric Science, National Central University, Taoyuan, 32001, Taiwan
[3]Department of Civil and Environmental Engineering, University of Tennessee, Knoxville, TN 37996, USA
[4]Graduate Institute of Environmental Engineering, National Central University, Taoyuan, 32001, Taiwan

*Correspondence to: mtchuang100@gmail.com*

10  **Abstract.** The outflow of East Asian haze (EAH) has gathered much attention in recent years. For downstream areas, it is meaningful to understand the impact of crucial upstream sources and the process analysis during transport. This study evaluated the impact of PM$_{2.5}$ from the three biggest industrial regions in Asian continent: Bohai Rim industrial region (BRIR), Yangtze River Delta industrial region (YRDIR), and Pearl River Delta industrial region (PRDIR) on Taiwan and discussed the processes during transport with the help of air quality modeling. The simulation results revealed the contributions of monthly average

15  PM$_{2.5}$ from BRIR and YRDIR were 0.7〜1.1 µg m$^{-3}$ and 1.2〜1.9 µg m$^{-3}$ (〜5 % and 7.5% of total concentration) on Taiwan, respectively in January 2017. When the Asian anticyclone moved from Asian continent to the West Pacific, e.g. on Jan 9th 2017, the contributions from BRIR and YRDIR to northern Taiwan could reach 6〜8 and 9〜12 µg m$^{-3}$. The transport of EAH from BRIR and YRDIR to low latitude regions was horizontal advection (HADV), vertical advection (ZADV), and vertical diffusion (VDIF) over Bohai Sea and East China Sea. Over Taiwan Strait and northern South China Sea, cloud processes

20  (CLDS) was the major contribution to PM$_{2.5}$ due to high relative humidity environment. Along the transport from high latitude regions to low latitude regions, Aerosol chemistry (AERO) and Dry deposition (DDEP) were the major removal processes. When the EAH intruded northern Taiwan, the major processes to the gains of PM$_{2.5}$ at northen Taiwan were HADV and AERO. The stronger the EAH was the easier the EAH could influence central and southern Taiwan. Although PRDIR was located at the downstream of Taiwan under northeast wind, the PM$_{2.5}$ from PRDIR could transport upward above boundary layer and

25  moved eastwards. When the PM$_{2.5}$ plume moved overhead Taiwan blocked by mountains, PM$_{2.5}$ could transport downward via boundary layer mixing (VDIF) and further enhanced by the passing cold surge. In contrast, for the simulation of July 2017, the influence from three industrial regions was almost negligible unless there was special weather system like thermal lows, which may carried pollutants from PRDIR to Taiwan, but the occurrence was rare.

**1. Introduction**

30  The damage of PM$_{2.5}$ (aerodynamic diameter is equal or less than 2.5 µm) on respiratory system has been proved (Kagawa, 1985; Schwartz et al., 1996;Zhu et al., 2011). The short-term human exposure to PM$_{2.5}$ could inflict cardiovascular and respiratory diseases, reducing lung functions, and increasing respiratory symptoms such as rapid breath, cough, and asthma. While the long-term influences include the mortality from heart or lung disease, cardiovascular illness (Pope et al., 2004;Brook et al., 2004;Ohura et al., 2005), and overuse of medical resources (Atkinson et al., 2001). Environmentally, the PM$_{2.5}$

35  not only absorbs and scatters solar radiation but also impairs visibility (Na et al., 2004), influences the balance of radiation and global climate (Hu et al., 2017), and the heterogeneous reactions of oxidants in the troposphere (Tie et al., 2005).

[Figure]

**Fig. 2.** revised manuscript

Supplement S1. Formulas for statistical evaluation indexes

1. Mean Bias

$$MB = \frac{1}{N}\sum_{1}^{N}(Sim - Obs)$$

5   2. Mean Average Gross Error

$$MAGE = \frac{1}{N}\sum_{1}^{N}|Sim - Obs|$$

3. Root Mean Square Error

$$RMSE = \sqrt{\frac{1}{N}\sum_{1}^{N}(Sim - Obs)^2}$$

4. Index of agreement, IOA

10

$$IOA = 1 - \frac{\sum_{i=1}^{N}(Sim - Obs)^2}{\sum_{i=1}^{N}(|Sim - \overline{Obs}| + |Obs - \overline{Obs}|)^2}$$

5. Wind Normalized Mean Bias

$$WNMB = \frac{\sum_{i=1}^{N}(Sim - Obs)}{N \times 360^o} \times 100\%$$

3. Wind Normalized Mean Error

$$WNME = \frac{\sum_{i=1}^{N}|Sim - Obs|}{N \times 360^o} \times 100\%$$

15   6. Mean Fractional Bias

$$MFB = \frac{2}{N}\sum_{1}^{N}\left(\frac{Sim - Obs}{Sim + Obs}\right)$$

7. Mean Fractional Error

$$MFE = \frac{2}{N}\sum_{1}^{N}\left|\frac{Sim - Obs}{Sim + Obs}\right|$$

8.Correlation coefficient (R)

20

$$R = \frac{1}{N}\sum_{i=1}^{N}\left[\frac{(Sim - \overline{Sim})(Obs - \overline{Obs})}{S_P S_o}\right]$$

$$S_P = \left[\frac{1}{N}\sum_{i=1}^{N}(Sim - \overline{Sim})^2\right]^{\frac{1}{2}}$$

$$S_o = \left[\frac{1}{N}\sum_{i=1}^{N}(Obs - \overline{Obs})^2\right]^{\frac{1}{2}}$$

**Fig. 3.** revised supplement file

---

## Author Comment (AC3) · 22 Feb 2020

Dear reviewer#1 and editor:

On behave of all co-authors, we are really grateful to reviewer#1 who spent much time reviewing the original manuscript. I know we have made some misleading narratives and have modified those in the revised manuscript. After all reviewers' comments are replied, the authors will an English language editing company to revise the manuscript. We know it is impossible but we hope the final manuscript could be an article written by a native English writer. In this response, we have attached three files: the manuscript of the main context, the supplement, and the one-to-one reply. We welcome any further

comments from reviewer#1. Thanks. Sorry that it was the first time the first author submitted a manuscript to ACP, so he did not upload the reply file at the correct place on the webpage. Please find the reply file called "corrected.version.zip" in which the authors have made one-to-one reply to the referee's comments. Best regards.

Ming-Tung Chuang

Please also note the supplement to this comment:
https://www.atmos-chem-phys-discuss.net/acp-2019-762/acp-2019-762-AC3-supplement.zip

---

## Author Comment (AC4) · 28 Feb 2020

**Response to Reviewers**

**Manuscript *acp-2019-762***

*We greatly appreciate the insightful comments and suggestions of the reviewers. Below please find a list of the Reviewers' remarks in contrast to our responses to them:*

**Review #1**

| Major Concerns | Responses |
|---|---|
| 1) The manuscript shows the analysis both for January and July. However, the impacts of three industrial regions on Taiwan in summer (July) is quite small, almost negligible even in the last few days when the impacts were relatively large. I don't think it is worthwhile spending much space for the July analysis, rather focusing on winter case would make the paper more concise and scientifically focused. | First, the authors really appreciate the reviewer spend much time and efforts reviewing this manuscript carefully and giving valuable opinions. They are truly grateful for the reviewer's comments which are very helpful to make this manuscript better. The authors admitted that they accidentally used non-precise and inappropriate words and so as to make misleading narratives. They promised that they will ask an English language editing company to revise the manuscript after all reviewers' comments are responded.

 Yes, the authors agree with the reviewer's suggestions and have cut down the contents of July analysis. They only keep the original Fig. 6 (Fig. 10 in the revised manuscript) in the main content and moved original Fig. 7 and Fig. 8 to Fig. S2.6 and Fig. S2.8 in the supplement of revised manuscript), and delete original Fig. 12, Fig. 13, Fig. 14, and Fig. 16. |
| 2) The results of process analysis was described and discussed in 3.2, 3.4, 3.5, and 3.6, which formed a main part of this paper. However, the descriptions in these sections were not firmly reasoned. In these sections, the author argued "dominant" contribution of three industrial regions at some locations. For example, in | Yes, the authors have written several misleading narratives in the original manuscript. After careful checking, first, they have modified the arrangement of the manuscript such that they combined the section 3.3, 3.4 and 3.6 of the original manuscript to section 3.5 of the revised manuscript in order to cut down the content of July analysis, and separate section 3.5 of the original manuscript into section 3.4 and 3.5 in the revised manuscript. Therefore, Fig. 11 of the original manuscript was changed to Fig. 9 and Fig. 14 was deleted. Now the main part is section 3.2, 3.4, and 3.5 for the revised manuscript.

 Second, they revised misleading narratives in order to avoid the |

| | |
|---|---|
| 3.2, the author pointed out that PM2.5 was influenced "mainly" by BRIR and YRDIR at the place #19. However, these arguments were not convincing. For the abovementioned example, Fig5 (c-2) and (c-3) which was regarded as representing the contributions by process of BRIR and PRDIR, respectively, showed similar variations to those of total contributions shown as Fig5 (c-1). However, the range of values largely differed each other, so I cannot understand why the author can conclude that the BRIR and YRDIR were "main" contributors to the variation of PM2.5 at #19. Similar arguments to this case can be found in these sections, and they considerably deteriorate the persuasiveness of the manuscript. I strongly recommend the author to revise such arguments in these sections and provide how to read and understand the main figures (Fig 5, 8, 11, and 14). | argued dominant contribution of three industrial regions. For example,

One line 229-230

From Fig. 5(b-1)-(b-4), among the three industrial regions it is apparent that #18 was influenced by both the BRIR and YRDIR,…….

On line 232-233

For #19, PM$_{2.5}$ was influenced mainly by YRDIR (Fig. 5(c-2)) and occasionally by BRIR (Fig. 5(c-3)) for those three industrial regions,………

On line 237-238

Although #20 is very near PRDIR, it was influenced more by YRDIR (Fig. 5(d-3)-5(d-4)) and other sources in the north other than three industrial regions since the prevailing wind was mainly northeast wind in January.

On line 324-325

From Fig. S2.8(a-1) to Fig. S2.8(a-4), it was found that #17 was influenced more by YRDIR than BRIR or PRDIR on July 18th 2017.

On line 337-338

According to the main content, among those three industrial regions BRIR and YRDIR were the major sources of #17 and #19 - #20, respectively. |
| Specific comments: | |
| L37: Seasonality of EAH is not "due to" rapid economic grows in Asian countries. | Yes, the authors thank the reviewer pointing out this error. In order to avoid the Chinese English writing, the authors promise that they will ask an English language editing company to help revise the revised manuscript after responding all reviewer's comments. In this temporary revised manuscript, that narrative was revised on line 37-38 as follows.

The East Asian haze (EAH) has been disturbing in spring and winter around the East Asia due to the spread of anticyclones over |

| | Asian continent. (Fu et al., 2014; Yang et al., 2016). |
|---|---|
| L43-45: Why did you specify these data and models for trajectory analysis? | The authors tried to make examples by mentioning the NOAA's data and models MM5 or WRF. They didn't mean to specify these data and models. In order to avoid misleading, the authors have revised the narratives on line 43-45 as

The trajectories could be calculated from, for example the archived meteorological data of NOAA ARL (www.ready.noaa.gov/archives.php) or the model outputs of MM5 (Mesoscale Model version 5, Dudhia, 1993) or WRF (Weather Research and Forecasting, Skamarock and Klemp, 2008).. |
| L50-51: Could you state more clearly why TS method would contain substantial uncertainty? | In the original manuscript, the authors intended to express that TS methods estimated the contribution of some upstream place on a receptor is to get the product of weighting of frequency passing that upstream place and concentration at that receptor. The authors have removed that narrative "Using trajectory to express the moving of a polluted plume would contain substantial uncertainty." in the original manuscript but added the following narratives on line 51-54 in the revised manuscript.

The plume transport from an upstream place to the receptor would exchange and react with air and pollutants along the path of transport. It suggests the plume arriving the receptor is no longer the plume emitted from the initial upstream place. The farther the upstream place is away from the receptor; the more uncertainty will be in the TS method. Therefore, the TS method would contain substantial uncertainty. |
| L54: The difference between those two runs does not directly mean the contribution of specific source but impact of the reduction of that specific source. To distinguish these two concepts is quite important. | The authors agreed with the reviewer's opinion regarding to the BFM methods and have modified narratives in the revised manuscript. On line 57-61:

The difference of the base case and the zero-out case is the reduction of the zero-out source. The reduction is approximate the contribution of that zero-out source only under the assumption when the contributions of each sources are additive. However, there is indirect contribution not considered in BFM method, i.e., the chemical reactions between the specific zero-out source and surrounding sources is neglected. The indirect contribution could be large if the zero-out sources and surrounding sources are both huge and have enough time to react.

The following description is not included in the revised manuscript but provide to the reviewer for communication.

If pollutants from BRIR or YRDIR moved to the sea and |

| | transported southward or pollutants from PRDIR moved to the free atmosphere and transported eastward, it is expected the pollutants emitted from those aforementioned three industrial regions should not have enough time to react with pollutants other than the industrial regions including areas other than three industrial regions in mainland China, along the transport and arriving at Taiwan. In other words, the contribution from the chemical reactions between the pollutants from industrial regions and pollutants from surrounding area is insignificant. In that case, we can roughly consider the reduction of the BRIR/YRDIR/PRDIR sources as the contribution of these industrial sources. When the pollutants from those three industrial regions arrived at Taiwan, it may react with pollutants from the local when they meet in the first place. In Chen et al. (2014), they estimated the indirect reactions between pollutants from mainland China and pollutants in Taiwan accounted about 10% of $PM_{2.5}$ in Taiwan. It is expected that the chemical reactions between pollutants from areas other than three industrial regions and pollutants from three industrial regions is not important because those two masses of pollutants did not mixed well during the transport. |
|---|---|
| L56-58: What do you mean "under-represented chemical reaction" here? Could you explain more specific? | The authors have change the word "under-represented" to "ignoring" on line 64 in the revised manuscript. |
| L67: CTM? This should be AM method? | Yes, the authors have modified that sentence to "However, the CTM especially the AM method is able to give clearer contributions from a specific source compared to the TS method." on line 74 in the revised manuscript. |
| L87: These abbreviations (LRT, LP) have already been defined | Thanks the reviewer's reminder. The authors have removed the repeated words. |
| L90: Meaning of these terms (LRT-Event and so on) should be explained | Yes, the authors should explain these terms and have already done on line 97-98 in the revised manuscript. They rewrite the sentence as "…LRT-Event (high concentration events caused nearly by pure LRT), LRT-Ordinary (non-events caused nearly by pure LRT), and LRT/LP&Pure LP (other days influenced by mix of LRT and LP & pure LP),…." |
| L98-99: Are power and industrial sectors the largest for entire Asia or any specific region in Asia? | Unlike developed countries, power and industrial sectors are the largest for most countries in Asia. According to the MIX Asian emission inventory, China and India dominate the emission of Asia for most of the species (Li et al. 2017). In the statistics of emissions |

from five anthropogenic sectors in Asia, the point source like power/Industry has the largest emission for $SO_2$, NMHC, TSP/$PM_{10}$/$PM_{2.5}$, OC, and $CO_2$, and is comparable to transportation for $NO_X$. The transportation is the largest emission for CO and BC. According to Zheng et al. (2018), the emissions from power and industrial sectors are the largest among all anthropogenic emissions except $NH_3$ that are mainly from agriculture in China in recent years. For NMHC, the emission from industry, residential, transportation, and solvent use are comparable to each other. Another famous Asian emission inventory REAS (latest version 3.1, Kurokawa and Ohara, 2020) also show similar results. However, there are occasional exception, for example, the domestic sector in South Asia other than India in 2015 has the largest emission for $SO_2$, NOx, $CO_2$, and $PM_{10}$/$PM_{2.5}$ than other sectors. and BC. While in Taiwan, $SO_2$ and CO are mainly from point source like power and industry; however, TSP/$PM_{10}$/$PM_{2.5}$/VOCs are mainly from area sources. $NO_X$ are mainly from point and mobile sources (TEPA, 2017).

As for Zheng et al. (2018) mainly discussed the anthropogenic emission in China, the authors understand the reviewer's comments and changed the citation to Li et al. (2017) and Kurokawa and Ohara (2020) on line 106 in the revised manuscript.

Kurokawa, J., and Ohara, T.: Long-term historical trends in air pollutant emissions in Asia: Regional Emission inventory in Asia (REAS) version 3.1, Atmos. Chem. Phys. Discuss., https://doi.org/10.5194/acp-2019-1122, in review, 2020.

Li, M., Zhang, Q., Kurokawa, J.-I., Woo, J.-H., He, K., Lu, Z., Ohara, T., Song, Y., Streets, D. G., Carmichael, G. R., Cheng, Y., Hong, C., Huo, H., Jiang, X., Kang, S., Liu, F., Su, H., and Zheng, B.: MIX: a mosaic Asian anthropogenic emission inventory under the international collaboration framework of the MICS-Asia and HTAP, Atmos. Chem. Phys., 17, 935–963, https://doi.org/10.5194/acp-17-935-2017, 2017.

TEPA: Building of the Taiwan emission data system. Taiwan EPA report, EPA-106-FA18-03-A263, in Chinese, 2017.

Zheng, B., Tong, D., Li, M., Liu, Fei, Hong, C., Geng, G., Li, H., Li, X., Peng, L., Qi, J., Yan, L., Zhang, Y., Zhao, H., Zheng, Y., He, K., and Zhang, Q.: Trends in China's anthropogenic emission since 2010 as the consequence of clear air actions. Atmos. Chem. Phys., 18, 14095–14111, https://doi.org/10.5194/acp-18-14095-2018,

| | 2018. |
|---|---|
| L103-104: This should be "the impact of reduction in source emission in each industrial region", because BFM can estimate "impact" not "contribution". Or you can define the wording that you will use the word "contribution" for the deference between control runs and sensitivity run. | Thanks the reviewer's suggestion. The authors have revised the narrative to "As mentioned above, the difference of Base and sensitivity scenarios is the reduction of the specific source. Only when the chemical reactions are not important then the reduction can be approximate the contribution of that specific source. In this study, the pollutants from those three industrial regions transport directly to Taiwan instead of meandering movement. Therefore, we can roughly estimate the contribution of BRIR, YRDIR, and PRDIR to PM$_{2.5}$ as the difference between the *Base* case and the *Brir*, *Yrdir*, and *Prdir* cases." on line 112-115 in the revised manuscript. |
| L123-127: For Figure1, the formal, not abbreviated, names for each monitoring station should be appeared here. | Thanks the reviewer's reminder. The authors have rewritten the names for each monitoring stations on line 133-139 in the revised manuscript.

For meteorology evaluation, we chose eight representative stations: Peng Jiayu (PJY, #1 in Fig. 1), Taipei (TPE, #2 in Fig. 1), Chupei (CP, #3 in Fig. 1), Taichung (TC, #4 in Fig. 1), Chiayi (CY, #5 in Fig. 1), Tainan (TN, #6 in Fig. 1), Kaohsiung (KH, #7 in Fig. 1), and Hengchun (HC, #8 in Fig. 1) stations to evaluate the modeling performance of temperature, wind speed, and wind direction. Since most residents lived at the relatively flat western Taiwan, the observations at the Banqiao (BQ, #9 in Fig. 1), Pingzhen (PZ, #10 in Fig. 1), Miaoli (ML, #11 in Fig. 1), Zhongming (ZM, #12 in Fig. 1), Chiayi (CY, #13in Fig. 1), Tainan (TN, #14 in Fig. 1), Zuoying (ZY, #15 in Fig. 1), and Hengchun (HC, #16 in Fig. 1) stations were chosen for PM$_{2.5}$ evaluation. |
| L130-131: Why you don't show the model domains in Figure 1 but just describe horizontal resolution? | Yes, the authors have redrawn the Figure 1 which shows the model domains in the revised manuscript. |
| L146: "MB" has already been defined in the previous sentences | Thanks the reviewer for carefully pointing out this extra. The authors have already removed the repeat one. |
| For the evaluation of WRF and CMAQ shown in Table 1 and 2, the results from which domain were used? And in addition to the summary of statistical indices in Table 1, | The authors have explained the simulated results from the fourth domain was evaluated for Table 1 and 2 one line 158 in the revised manuscript.

The authors have added figures of comparisons of observed and simulated temperature (Fig. S2.1), wind speed (Fig. S2.2), and wind direction (Fig. S2.3) in the supplement of the revised manuscript. |

| | |
|---|---|
| figures of comparisons of temperature and wind between observation and simulation are quite informative. Could you put them together at least as supplement? | In addition, the authors also added Fig. S2.4 which show the comparisons of observed and simulated $PM_{2.5}$ in the supplement of the revised manuscript. |
| You should explain how you draw Fig3. Are the values in Fig3 difference between Base case and sensibility case? If so, it's better to note it in the manuscript or in figure caption. Fig3 is a bit busy, so it seems better to select fewer locations out of seven to avoid redundancy. | Yes, the authors have already explained how to get the values in Fig. 3 both on line 223 in the revised manuscript and the caption of figure 3. In addition, the authors have removed few locations but only remained BQ, ZM, and CY to representative northern, central, and southern Taiwan.

On line 182-183 in the revised manuscript:

As mentioned, the impact was considered as the reduction of specific source removed or roughly the contribution of that specific source, i.e. the difference between the base and sensitivity scenarios. |
| L176: Remove unnecessary "the". | Thanks the reviewer for pointing out this typo. The authors have already removed the extra "the". |
| Could you check the wording "China East Sea"? "East China Sea" has been also used for the same area in many literatures. | Thanks the reviewer's careful checking for this manuscript. The authors have already unified the word to "East China Sea" in the revised manuscript. |
| For Figure5, you should explain how to deduce the values shown in the figure, in particular the values in Fig5(*-2,3,4). Are they the difference between Base case and sensitivity case? If so, you should instruct briefly how to interpret these Figures. Is the title of y-axis correct? This should be "_concentration" or "daily concentration change"? | Yes, the authors have followed the reviewer's suggestion to explained Fig. 5 are deduced by the difference between Base case and sensitivity cases on line 220-222 in the revised manuscript as follows.

Similar to Fig. 2, we deduced the difference of base and sensitivity scenarios for IPR analysis. This study considered the reduction as the approximate contribution for each industrial region. Therefore, the reader should keep in mind that the following discussion is satisfied on when the chemical reaction between each industrial region and surrounding was ignored.

Thanks the reviewer's reminder that the title of y-axis should be "daily concentration change" or "change in daily concentration". The authors have already corrected this error in Fig 5 And Fig S2.8 in the revised manuscript. |
| L204: Fig5(a-1) and (a-2) do not seem quite similar to each other. Could you specify more about which features of both figures | Yes, the authors agree that they did not use precise vocabulary and have removed the word "similar" to avoid misleading. The revised narratives on line 223-224 in the revised manuscript is "The positive and negative contribution terms in Fig 5 (a-1) and Fig. (a- |

| | |
|---|---|
| look similar? | 2) appealed synchronously although their magnitudes were not in equal proportions." |
| L204: You concluded that main contributor to #17 PM2.5 is BRIR, but I cannot understand why you can conclude like this. The values in Fig5(a-1) and (a-2) are quite different. You should give an instruction how to read and understand the Fig5 | The authors have modified the narratives on line 223-225 in the revised manuscript as follows.

The positive and negative contribution terms in Fig 5 (a-1) and Fig. (a-2) appealed synchronously although their magnitudes were not in equal proportions. It implies #17 was influenced by both BRIR and other nearby sources. |
| L205: Can HADV process "produce" PM2.5? The term "production" here is not appropriate. | The authors understand what the reviewer meant and have already modified that narrative on line 224-225 to "The increase of $PM_{2.5}$ was caused mainly by the process HADV, followed by ZADV and VDIF….".

In addition, the authors have examined the whole manuscript and modified all such narratives.

On line 244

The build-up of $PM_{2.5}$ at BQ was mainly HADV with minor CLDS……

On line 259-261

For CY located in southwestern Taiwan, VDIF and HADV mainly contributed to the gains of $PM_{2.5}$, and the removal processes were mainly ZADV and AERO; however, occasionally when the positive contribution to $PM_{2.5}$ were ZADV and VDIF, the removal processes were HADV and AERO (Fig. 5(f-1)).

On line 261-262

Compared Fig. 5(f-2)-(f-4) and Fig. 5(g-2)-(g-4), it is obvious the positive and negative contribution to $PM_{2.5}$ for CY were very similar to for ZM.

On line 286-287

The major processes below layer 9 (~310 m) contributing to the increase of $PM_{2.5}$ were HADV, VDIF, and ZADV and removal processes were DDEP and AERO (Fig. 8(b-3)).

On line 292-293

Although #18 and BQ were most affected by YRDIR, the major contribution process at BQ below 200 m (layer 7) was HADV…..

On line 302-303

Second, for the haze from BRIR and YRDIR, the positive and negative contribution processes on BQ were mainly HADV/AERO….. |

| | On line 304-305 |
|---|---|
| | "While on Jan 9th, the major processes leading to the increase of PM$_{2.5}$ at BQ ……."

On line 325

The positive and negative contribution processes were non-uniform below 80 m (layer 4).

On 325-327

But from 120 m to 460 m (layer 5 to layer 11), the major processes to build-up of PM$_{2.5}$ were AERO and ZADV and the removal process was mainly HADV.

On line 368-369

When the EAH moved to northern Taiwan, HADV and AERO were the major contribution processes of PM$_{2.5}$ at BQ.

On line 371-372

The stronger the intensity of EAH, the impact on central and southern Taiwan was more obvious, the proportion of HADV contributed to PM$_{2.5}$ budget was more obvious near surface. |
| L211: What process considered in AERO can reduce PM2.5? | Since the ambient environment was cold in high latitude regions and warm in low latitude regions, the evaporation process of PM2.5 occurred in the haze during transporting moved southward. In the simulation study of Chuang et al. (2008), the evaporation of NH$_3$NO$_3$ occurred for the PM2.5 plume transported from Shanghai to Taipei and formed ammonia and nitric acid. The ammonia reacted with sulfur dioxides and form ammonium sulfate and the nitric acid remained in the plume and reacted with ammonia emitted in Taipei and formed ammonium nitrate again. It is expected the evaporation of organic carbon also occurred if ambient temperature increased. Another very minor process which could be ignored compared with abovementioned evaporation process is the coagulation of PM$_{2.5}$ particles converted to coarse particles.

Chuang, M. T., Fu, J. S., Jang, C. J., Chan, C. C., Ni, P. C., and Lee, C. T.: Simulation of long-range transport aerosols from the Asian Continent to Taiwan by a Southward Asian high-pressure system. Sci. total. Enviro., 406, 168–179. |
| L213: If the intrusion of PM2.5 from PRDIR is like that depicted in Fig4, why the contribution of ZADV is not so large in Fig5(c-4)? Since #19 is located between PRDIR and Taiwan island and | Fig. 4 is the cross section of red line in domain 2 and domain3. The ZADV is not so large in Fig. 5(c-4) is probably # 19 is not on the red line (the cross section) in Fig. 1. In addition, the less influence of PM$_{2.5}$ from PRDIR was mainly on the mountains, as shown in Fig. 2(e) and Fig. 2(f), i.e. at high altitude about 1-3 km. The downward motion is not obvious unless the plume was blocked |

| | |
|---|---|
| the transport of PM2.5 between them occurs about 1-2 km high above the surface as in Fig4, any kind of vertical (downward) motion should transport PM2.5 from that layer to the location of #19 which must be at the surface | by the mountains in Taiwan (Fig. 4). |
| L227: What does "minor PM2.5" means here? | The authors have replaced the word "minor" with "certain" on line 247 in the revised manuscript. |
| L228: Why can you describe "The PM2.5 at BQ then transport up- and then southwards"? Which figure show this transport of PM2.5? | Thanks the reviewer for pointing the error. The removal process of $PM_{2.5}$ at BG was mainly ZADV. In order to explain clearly, the authors have modified the narrative as "The removal process of $PM_{2.5}$ at BQ was mainly ZADV, which implies PM2.5 at BQ then transport up and reflects BQ is located in a basin." on line 248-249 in the revised manuscript. |
| L228-229: Fig.(f-1) -> Fig5 (f-1) | Thanks the reviewer for pointing the error. The authors have already corrected the type. |
| L234-235: If this is true, why ZADV in Fig5 (f-4) is largely negative from Jan 8 to 10? | Because of the reviewer's comment, the authors found the ZADV has to be treated in an opposite way since the concentration gradient is positive for $PM_{2.5}$ from PRDIR, which is different from the usual cases that $PM_{2.5}$ concentration was usually higher near surface. The authors have modified some narratives in the revised manuscript. On line 206-207 The boundary layer mixing was enhanced by the pass of cold surge and increased $PM_{2.5}$ on the ground. On line 258-259 On Jan 8th to 10th, the negative ZADV indicated the concentration was decreasing at the lower 20 averaged layers but the concentration gradient was positive ($\frac{\partial PM_{2.5}}{\partial z} > 0$, the concentration of $PM_{2.5}$ from PRDIR was higher at high altitude than that at low altitude over Taiwan) implies the vertical velocity had to be negative, i.e. downward motion. Therefore, the boundary layer mixing of the aloft $PM_{2.5}$ plume was enhanced by the pass of the cold surge. (Yen et al., 2013; Chuang et al., 2016). The following is a brief review that was not in the revised manuscript but provide to the reviewer for communication. Yen et al. (2013) suggested the downward motion could bring Southeast Asian biomass burning pollutants aloft to surface through the subsidence of cold surge through the wind field derived from NCEP Global Forecast System analyzed data. Chuang et al. (2016) applied |

the WRF/CMAQ and found the Southeast biomass burning aerosols could be blocked by the mountains in Taiwan and then the boundary layer mixing assisted the subsidence of aloft aerosols to the surface. Huang et al. (2020) suggested the 700-hPa LLJ (Low Level Jet) may have carried the biomass burning plumes aloft located south of the frontal system (cold surge) and accompanied the upward/downward motion south/north of the frontal system. The downward motion occurred at the north of the front or subsidence of cold air region. While in the simulation of present study, the ZADV was negative which also implied the downward advection occurred when the cold surge passed. Yes, it is a pity that there is no observation for the pollutants profile during the pass of cold surge. Otherwise, it would be more persuasive.

Chuang, M. T., Fu, J. S., Lee, C. T., Lin, N. H., Gao, Y., Wang, S. H., Sheu, G. R., Hsiao, T. C., Wang, J. L., Yen, M. C., Lin, T. H., and Thongboonchoo, N.: The Simulation of Long-Range Transport of Biomass Burning Plume and Short-Range Transport of Anthropogenic Pollutants to a Mountain Observatory in East Asia during the 7-SEAS/2010 Dongsha Experiment. Aerosol. Air. Qual. Res., 16, 2933–2949, https://doi.org/10.4209/aaqr.2015.07.0440, 2016.

Huang, H.-Y., Wang, S.-H., Huang, W.-X., Lin, N.-H., Chuang, M.-T., da Silva, A. M., & Peng, C.-M. (2020). Influence of synoptic-dynamic meteorology on the long-range transport of Indochina biomass burning aerosols. Journal of Geophysical Research: Atmospheres, 125, e2019JD031260. https://doi.org/10.1029/2019JD031260.

Yen, M. C., Peng, C. M., Chen, T. C., Chen, C. S., Lin, N. H., Tzeng, R. W., Lee, Y. A., and Lin, C. C.: Climate and weather characteristics in association with the active fires in northern Southeast Asia and spring air pollution in Taiwan during 2010 7-SEAS/Dongsha Experiment, Atmos. Envoron., 78, 35-50, http://dx.doi.org/10.1016/j.atmosenv.2012.11.015, 2013.

| | |
|---|---|
| L256: Why did you exclude Fig.8(a)? | Fig. 8(a) is for #17. That sentence begins with "In addition to #17". It is obvious that # 17 was influenced by BRIR at the end of July 2017 and by YRDIR on most days in the same month. The Fig. 8 in the original manuscript has been moved to Fig. S2.8. |
| L267: Could you put the prevailing wind vector in Figures 2 and 6, otherwise I can | The authors have added monthly average wind field in Fig. 2 and Fig. 6 already. It is obviously the prevailing wind in winter was northeast wind (Fig. 2) but south wind in summer (Fig. 6). |

| not verify what you described here and similar descriptions in the manuscript explaining the impact of wind patterns. | |
|---|---|
| L280: Layer4? Is this Layer14? | Thanks the reviewer for pointing out this typo. The authors have corrected 4 to 14 in the revised manuscript. |
| L281: It is apparent that only vertical motion can not transport PM2.5 from BRIR to #17. What do you mean here? | The authors would like express the transport from BRIR to #17 was not just horizontal but also vertical even the distance is not long between them. The authors have modified the narratives on line 278-279 as "It implies the transport path from BRIR to #17 could be horizontal between BRIR and #17 and then vertical at the location of #17." in the revised manuscript. |
| L282-283: Why does ascent (descent) motion enhance (decrease) aerosol formation? What processes are involved ? | The authors have added above narratives on line 277-280 in the revised manuscript.

It is possibly that the ascent motion of air parcel near the warm surface moved to a cold environment in higher altitude. This may cause condensation and triggered heterogeneous reactions of aerosols. On the contrary, the descent motion of air parcel may cause the evaporation of aerosols. |
| L291: Fig. (e-2)-(e-4) -> Fig11. (e-2)-(e-4). | Thanks the reviewer for pointing out this typo. The authors have corrected the typo in the revised manuscript. The Fig. 11 in the original manuscript have been changed to Fig. 8 in the revised manuscript. |
| L293: mixed -> mixture | Thanks the reviewer for pointing out the inappropriate word. The authors have corrected the word on line 294 in the revised manuscript. |
| L340: higher -> lower? | Thanks the reviewer for pointing out this typo. The authors have corrected the typo on line 347 in the revised manuscript. |
| L341: underestimated -> overestimated? | Thanks the reviewer for pointing out this typo. The authors have corrected the typo on line 348 in the revised manuscript. |
| L353: There is not Fig.S2.6 in the supplement | The Fig. S2.6 is on the last page of supplement, now as Fig. 2.11 in the revised supplement. |
| L380: There is no comparison for July 30th (no Fig. S2.6). | It is really a pity that there is no observation on July 30th due to bad weather (the influence of the thermal low). The authors have modified the caption which does not include observation. |

**Review #2**

| | |
|---|---|

**Review #3**

| | |
|---|---|

---

## Referee Comment (RC2) · Anonymous Referee #2 · 14 Apr 2020

General Description

This paper describes the contribution of three major Asian industrial regions on PM2.5 concentrations in Taiwan in January and June 2017. WRF and CMAQ models were used to simulate the transport of pollutions from the Asian industrial regions and also the chemical reactions in these plumes. The performance of the model in capturing temperature, wind speed, and direction, and PM2.5 was evaluated in multiple stations located in Taiwan covering north to south of the island. The authors used the process analysis technique in CMAQ to identify the dominant physical and chemical processes for the production and removal of PM2.5 in different locations in the domain. In general, the topic is suitable for ACP journal and the paper makes interesting conclusions about the contribution of long-range transport under different transport patterns to the air quality of Taiwan. However, the authors need to address some scientific issues discussed in the comments section below. The paper needs major English proofreading, major technical corrections, better quality for figures. I would not recommend this paper for publications unless these issues are addressed.

Please note that I reviewed the updated version of the paper after the comments from reviewer 1 were addressed.

General Comments

1) The contribution of local emissions was discussed very briefly in the last section of the paper. I believe adding a discussion about the contribution of local emission to the measured PM2.5 can be beneficial for drawing fair conclusions.

2) I recommend adding backtrajectory analysis using HYSPLIT when discussing transport patterns on specific days. I added more details in the specific comments section.

3) The paper misses a lot of important information such as the main configurations of the model, details on the emission inventory used, and information about the location of measurement sites and equipment. I highly recommend adding these to the paper for the purpose of reliability and reproducibility of the work.

4) Were there any seasonal or diurnal cycle in the emissions? Are January and July emissions different?

5) Major changes are required for the figures. The texts are too small in many of them, the color bar can be improved. I added more comments about each figure in specific comments.

6) I did not make comments on the grammatical mistakes, incomplete sentences, and inconsistencies as there were too many.

Specific Comments Introduction: 1. The first two paragraphs in the Introduction section need to be re-written with better English.

2. L69. The reference at the end of the sentence (Byuan and Schere, 2006) does not match the reference at the beginning of the sentence (Kwok et al. (2013)).

3. L65. Consider starting a new paragraph when describing the AM method.

4. L65-75. After reading this section I was under the assumption that the AM method performs better and was used in this study. At the end of this paragraph please mention that you did not use the AM method and used the BFM method instead.

5. L86...nitrate and sulfate... Please be consistent and either use the chemical formula or the name in the paper or both.

6. L99. When is the northeast monsoon period? Which season/months?

7. L111. Change Brir to BRIR ...same for other emission regions.

8. L115. What do you mean by "meandering movement"? You can here refer to previous studies that showed this.

9. L120. I suggest moving the discussion of monsoon seasons earlier in the introduction section.

Methods:

10. L128. In addition, year 2017 .... I don't understand this sentence.

11. 2.1. Geographical location of meteorological ... Are stations with the same names (for example #5 and #13) in the same locations? In the text, you use the station names but in Fig 1, you used the numbers. To find the location of each station in Fig 1 readers must go back and forth between section 2.1, fig 1 and the text. Please be consistent and either use numbers or names in figures, tables, and text.

12. 2.1. Geographical location of meteorological ... Please provide more information

about the measuring equipment, the temporal resolution of data and reference to the measurement data used.

13. L142. ...NCEP diagnostic fields. Please use a reference for this data set. There is doi available for this data set.

14. L142. Which nesting method did you use? One or two-way?

15. L144. What is the model's top?

16. L145. What is the temporal and special resolution of the emission inventories used? Is there a diurnal or seasonal variability?

17. L150. Why different biogenic inventories were used for different domains?

18. 2.2 Models and modeling configuration Please add a table (can be in SI) with all main WRF and CMAQ configurations and schemes such as PBL scheme, LSM, cumulus scheme, ... How long was the spin-up? What did you use for chemical initial and boundary conditions?

19. 2.2 Models and modeling configuration Did you do any nudging or re-initialization of the model? Please add details to this section.

20. L161. Is there any RH data available? If yes then adding discussion on model performance in capturing RH can be very beneficial for the paper.

21. L167...which is due to the smoother terrains... In Fig 1, HC is located very close to the sea. Is there a complex terrain in that region? It is not very clear in the figure. Can smoother terrain in the model impact other stations as well?

22. L169. Are other stations influenced by buildings?

23. L173. Please use better quality plots for figure S2.3. Also, be consistent in the title of subplots.

24. Table 1 and table 2. Please add mean model and observed values to these tables.

This can help better compare January and June values and values in different stations.

25. L173. 2.3.2. Evaluation of CMAQ chemical modeling… Please add an emission map. Are any of the stations close to major emission sources?

26. L173. 2.3.2. Evaluation of CMAQ chemical modeling… Please mention that PM2.5 values are very low in HC compared to other stations (Fig S2.4)

27. L173. 2.3.2. Evaluation of CMAQ chemical modeling… Is there a significant difference between PM2.5 values in January compared to June?

Results and Discussion 28. L185. How did you calculate 5%? Is this for the whole island or 5% is the maximum value?

29. Fig 2. Please consider using a better color bar. Why negative values for the color bar? Use more colors for 0-2ug/m3 (right column) and 0-5% (left column).

30. L186. Fig 3 only shows three stations, not seven. Why did you use only these stations? How far are they from major local emission sources?

31. L187. This is not true for PRDIB contribution which is higher in central and southern Taiwan (C-2 and C-3) compared to northern Taiwan (C-1).

32. L189. January 8th or 9th? 14th or 13th? In Fig 3 column a 9th, 14th, 20th, in column b 9th, 13th, 20th had the highest PM2.5 concentrations and contribution from BRIB and YRDIB. Why did you pick 9th and 13th? Throughout the text, different days were mentioned which can be confusing for the readers. Please be consistent and clearly justify your choice of Jan 9th and 13th.

33. L 195. What do you mean by almost the same? Please be more specific.

34. L196. …could reach … In which stations? 6-8 ug/m3 and 9-12ug/m3, why giving a range?

35. L200. Please show where Fujian and Guangdong are in Fig 2.

36. L202. Fig. 4. There are two red lines in Fig. 1. Did you use both of them? Please clearly mention this in the text.

37. L214. Locations #17-20 are missing from the updated Fig 1.

38. L 214. Please mention that you did not evaluate model performance (transport and chemistry) in these locations.

39. L224. The positive and negative . . . I don't understand this sentence

40. Fig. 5. What does column 1 represent? What do you mean by contribution of total emission? Do you mean the base case?

41. Fig. 5. Please add titles to the subplots. Or at least put titles for each row and column. It is very difficult to interpret this figure.

42. L226. Can you be more specific about the evaporation of ammonium nitrate in PM2.5 when moving from high latitude to low latitude regions?

43. L245. I cannot distinguish between ZADV and CHEM in Fig 5. Use different colors

44. Fig 5e-1. Any comment on why the daily concentration change is much higher in BQ (#10) than others? Does this mean a high contribution of local emissions? Please discuss this.

45. L247. The removal process of . . . . This sentence is unclear.

46. L250. . . . the PM2.5 of ZM. . . I don't understand this sentence.

47. L259. For CY. . . Please mention that CY (#14) and ZM (#13) are closer to each other than BQ (#10).

48. 3.2. The physical . . . Please justify why you chose to only use #10, #14, and #13 in this section. Please provide a more detailed discussion on the contribution of local emissions.

49. L266. The section number is not correct. Why Jan 13th was discussed before Jan

9th? How did you classify Jan 13th as a severe episode and Jan 9th as a moderate episode?

50. L274 Fig. 8. Please add the altitude of each layer to the figure.

51. L275. The arrival of LRT haze on Jan 14-15 can also be seen in Fig 3.

52. Fig 8. Again I don't understand why Jan 13th was chosen for this discussion. The contribution of LRT was small on this day compared to Jan 14th or 15th. Maybe using these days for Fig 8 would be more helpful?

53. L296. Downstream not upstream.

54. L266 Analysis of . . . Adding Hysplit back-trajectories released from locations discussed in this section can be very helpful. It can reveal the trajectory and the origin of the plumes arrived at each of the locations and add confidence to this discussion.

55. L309. What is vv?

56. 3.5 Analysis of the moderate . . . I think it is worth discussing this event further (similar to Jan 13th) especially with the high values in BQ at lower levels.

57. L316 . . .for all cities. Cities or stations

58. L325. Why July 18th? I don't see high PM2.5 concentrations for July 18th in any of the subplots in row a (Fig. S2.8).

59. L325. The positive and negative contribution . . . Does this refer to July 18th? This is not shown in any figure.

60. Fig 2.9 and L330. Please use a better color bar. More colors between 0-20 ug/m3.

61. How much is the local emission contribution in July and how does this compare with January?

62. L225. Where is Fig 15?

63. L338. According to the main content.... Are you referring to Fig 8? If yes then your statement is incorrect, BRIR and YRDIR did not have a contribution to #19 (c-2 and c-3) and #20 (d-2 and d-3). Looks like Jan 13th is not the best day to pick for this discussion. Is this measurement available on Jan 9th or 20th?

64. Fig 11. OC and NH4+ colors are very similar.

---

## Author Response (AR1)

Dear reviewers and editor:

On behave of all co-authors, we are really grateful to reviewers who spent much time reviewing the original manuscript. I know we have made some misleading narratives but have modified those in the revised manuscript. **Please notice that the revision according to reviewer#1's and reviewer#2' comments are written in red words and in yellow background, respectively.** Before the submission of revised manuscript, the authors have asked a professional English editing company to revise the English writing. We hope the revision have avoided grammar mistakes and misleading narratives already. In this response, we have attached three files: the manuscript of the main context, the supplement, and the one-to-one response. We sincerely thank for the editor, reviewers', and ACP staff's effort.

Best regards.

Ming-Tung Chuang

**Response to Reviewers**

**Manuscript *acp-2019-762***

*We greatly appreciate the insightful comments and suggestions of the reviewers. Below please find a list of the Reviewers' remarks in contrast to our responses to them:*

**Review #1**

| Major Concerns | | |
|---|---|---|
| (1) comments from Reviewers | (2) author's response | (3) author's changes in manuscript. |
| 1) The manuscript shows the analysis both for January and July. However, the impacts of three industrial regions on Taiwan in summer (July) is quite small, almost negligible even in the last few days when the impacts were relatively large. I don't think it is worthwhile spending much space for the July analysis, rather focusing on winter case would make the paper | First, the authors really appreciate the reviewer spend much time and efforts reviewing this manuscript carefully and giving valuable opinions. They are truly grateful for the reviewer's comments which are very helpful to make this manuscript better. The authors accidentally used non-precise or inappropriate words and so as to make misleading narratives. Before submission of the revised manuscript, they have asked a professional English editing company to revise the manuscript already. Yes, the authors agree with the reviewer's suggestions and have cut | **Please notice that the revision according to reviewer#1's comments are written in red words.** The discussion of July is concentrated in the section 3.5 and the original Fig. 6 (Fig. 10 in the revised manuscript) was kept in the main content. They also moved original Fig. 7, Fig. 8, Fig. 12, Fig. 13, and Fig. 14 to Fig. S4.8, Fig. S4.9, Fig. |

| | | |
|---|---|---|
| more concise and scientifically focused. | down the contents of July analysis. They concentrated the content of discussion of July in the section 3.5 and kept the original Fig. 6 (Fig. 10 in the revised manuscript) in the main content and moved original Fig. 7, Fig. 8, Fig. 12, Fig. 13, and Fig. 14 in the original main content to Fig. S4.8, Fig. S4.9, Fig. S4.12, Fig. S4.13, and Fig. S4.14 in the revised supplement. | S4.12, Fig. S4.13, and Fig. S4.14 from the main content to the supplement. |
| 2) The results of process analysis was described and discussed in 3.2, 3.4, 3.5, and 3.6, which formed a main part of this paper. However, the descriptions in these sections were not firmly reasoned. In these sections, the author argued "dominant" contribution of three industrial regions at some locations. For example, in 3.2, the author pointed out that PM2.5 was influenced "mainly" by BRIR and YRDIR at the place #19. However, these arguments were not convincing. For the abovementioned example, Fig5 (c-2) and (c-3) which was regarded as representing the contributions by process of BRIR and PRDIR, respectively, showed similar variations to those of total contributions shown as Fig5 (c-1). However, the range of values largely differed each other, | Yes, the authors have written several misleading narratives in the original manuscript. After careful checking, first, they have revised misleading narratives in order to avoid the arguments that described which industrial region was the dominant contribution for the downstream receptors. | One line 272-275

From Fig. 5(b-1)-(b-4), among the three industrial regions it is apparent that #2 was influenced by both the BRIR and YRDIR, mainly produced through nonuniform HADV, VDIF, ZADV, and CLDS; and removed through AERO and occasional HADV and DDEP processes, and almost unaffected by PRDIR.

On line 275-277

For #3, $PM_{2.5}$ was influenced mainly by YRDIR (Fig. 5(c-2)) and occasionally by BRIR (Fig. 5(c-3)), but it was also influenced by PRDIR from the 8th to 12th (Fig. 5 (c-4)), which has been verified to be related to the transboundary transport and intrusion of a cold surge in the last section (Fig. 4).

On line 280-281 |

| | | |
|---|---|---|
| so I cannot understand why the author can conclude that the BRIR and YRDIR were "main" contributors to the variation of PM2.5 at #19. Similar arguments to this case can be found in these sections, and they considerably deteriorate the persuasiveness of the manuscript. I strongly recommend the author to revise such arguments in these sections and provide how to read and understand the main figures (Fig 5, 8, 11, and 14). | | Although #4 is very near PRDIR, it was influenced more by YRDIR (Fig. 5(d-3)-5(d-4)) and other sources in the north rather than three industrial regions since the prevailing wind was mainly northeast wind in January.

On line 379-380
Take July 18, 2017 as an example, in which the $PM_{2.5}$ sampling was implemented, it was found that #1 was influenced more by YRDIR than BRIR among three industrial regions (Fig. S4.11(a-1)-(a-4)). |
| Specific comments: | | |
| (1) comments from Reviewers | (2) author's response | (3) author's changes in manuscript. |
| L37: Seasonality of EAH is not "due to" rapid economic grows in Asian countries. | Yes, the authors thank the reviewer pointing out this error. In order to avoid the misleading writing, the authors have asked a professional English editing company to help revise the revised manuscript already. | On line 44
The EAH has started to spread out from Asia Continent to East Asia in spring and winter due to the movement of anticyclones. (Fu et al., 2014; Yang et al., 2016). |
| L43-45: Why did you specify these data and models for trajectory analysis? | The authors tried to make examples by mentioning the NOAA's data and models MM5 or WRF. They didn't mean to specify these data and models. In order to avoid misleading, the authors have revised the narratives | On line 48-50
The trajectories could be calculated from, for example, the archived meteorological data of NOAA ARL (www.ready.noaa.gov/archives.php), the model outputs of MM5 (Mesoscale Model version 5, Dudhia, 1993), or WRF (Weather Research and Forecasting, Skamarock and Klemp, 2008). |

| | | |
|---|---|---|
| L50-51: Could you state more clearly why TS method would contain substantial uncertainty? | In the original manuscript, the authors intended to express that TS methods estimated the contribution of some upstream place on a receptor is to get the product of weighting of frequency passing that upstream place and concentration at that receptor. The authors have removed that narrative "Using trajectory to express the moving of a polluted plume would contain substantial uncertainty." in the original manuscript but rewritten the narratives. | On line 56-59

 The plume transport from an upstream region to the receptor would mix and react with air and pollutants along the path of transport. This suggests that the plume arriving at the receptor is no longer the plume emitted from the initial upstream region. The farther the upstream place is from the receptor, the more uncertainty there will be in the TS method. Therefore, the TS method would contain substantial uncertainty. |
| L54: The difference between those two runs does not directly mean the contribution of specific source but impact of the reduction of that specific source. To distinguish these two concepts is quite important. | The authors agreed with the reviewer's opinion regarding to the BFM methods and have modified narratives in the revised manuscript.

 The following description is not included in the revised manuscript but provide to the reviewer for communication.
 If pollutants from BRIR or YRDIR moved to the sea and transported southward or pollutants from PRDIR moved to the free atmosphere and transported eastward, it is expected the pollutants emitted from those aforementioned three industrial regions should not have enough time to react with pollutants other than the industrial regions including areas other than three industrial regions in mainland China, along the transport and arriving at Taiwan. In other words, the contribution from the chemical reactions between the pollutants from industrial regions and pollutants from surrounding area is insignificant. In that case, we can roughly consider the reduction of the BRIR/YRDIR/PRDIR sources as the contribution of these industrial sources. It is expected that the chemical | On line 62-66
 The difference between the base case and the zero-out case is the reduction of the zero-out source. The reduction is approximately the contribution of that zero-out source under the assumption when the contributions of each sources are additive. However, there is an indirect contribution not considered in the BFM method, i.e., the chemical reactions between the specific zero-out source and surrounding sources are neglected. The indirect contribution could be large if the zero-out sources and surrounding sources are both huge and have sufficient time to react. |

| | reactions between pollutants from areas other than three industrial regions and pollutants from three industrial regions is not important because those two masses of pollutants did not mix well during the transport. | |
|---|---|---|
| | When the pollutants from those three industrial regions arrived at Taiwan, it may react with pollutants from the local when they meet in the first place. Chen et al. (2014) estimated the indirect reactions between pollutants from mainland China and pollutants in Taiwan accounted for about 10% of $PM_{2.5}$ in Taiwan. Even there exists the controversy that whether the 10% indirect reactions should be for LRT or LP, fortunately the proportion of indirect reactions is not significant. In addition, if the movement of LRT plume is rapid, then it has no sufficient time to react with the local pollutants. While if the movement is slow, although there is sufficient time for the chemical reactions, the pollutants mixing ratios in such plumes are low. It is expected the contribution of chemical reactions is not important. | |
| L56-58: What do you mean "under-represented chemical reaction" here? Could you explain more specific? | The authors have modified that sentence in the revised manuscript. | On line 68-70 Nevertheless, this method is not perfect because it potentially ignores chemical reactions between the specific sources within the remaining sources. |
| L67: CTM? This should be AM method? | Yes, the reviewer#1 is right. The authors have modified that sentence to make it clear. | on line 78-79 The CTM, especially the AM method, is able to give clearer contributions from a specific source compared to the TS method or the BFM method. |
| L87: These abbreviations (LRT, LP) have already been defined | Thanks the reviewer's reminder. The authors have removed the repeated words. | |

| | | |
|---|---|---|
| L90: Meaning of these terms (LRT-Event and so on) should be explained | Yes, the authors should explain these terms and have already done. | on line 105-108

They classified the daily $PM_{2.5}$ into LRT-Events (high concentration events caused nearly by pure LRT), LRT-Ordinary (nonevents caused nearly by pure LRT), and LRT/LP&Pure LP (other days influenced by a mix of LRT and LP & pure LP), which were 31–39 µg m$^{-3}$, 12–16 µg m$^{-3}$, and 4–13 µg m$^{-3}$ at the northern tip of Taiwan from 2006 to 2015 for the northeast monsoon period. |
| L98-99: Are power and industrial sectors the largest for entire Asia or any specific region in Asia? | Unlike developed countries, power and industrial sectors are the largest for most countries in Asia. According to the MIX Asian emission inventory, China and India dominate the emission of Asia for most of the species (Li et al. 2017). In the statistics of emissions from five anthropogenic sectors in Asia, the point source like power/Industry has the largest emission for $SO_2$, NMHC, TSP/$PM_{10}$/$PM_{2.5}$, OC, and $CO_2$, and is comparable to transportation for $NO_X$. The transportation is the largest emission for CO and BC. According to Zheng et al. (2018), the emissions from power and industrial sectors are the largest among all anthropogenic emissions in China except $NH_3$ that are mainly from agriculture in recent years. For NMHC, the emission from industry, residential, transportation, and solvent use are comparable to each other. Another famous Asian emission inventory REAS (latest version 3.1, Kurokawa and Ohara, 2020) also show similar results. However, there are occasional exception, for example, the domestic sector in South Asia other than India in 2015 has the largest emission for $SO_2$, NOx, $CO_2$, and $PM_{10}$/$PM_{2.5}$ than other | On line 116-117

From the emission map of Asia (Li et al., 2017; Kurokawa and Ohara, 2020), the largest emission sources were the power and industry sectors. |

sectors. While in Taiwan, $SO_2$ and CO are mainly from point source like power and industry; however, TSP/$PM_{10}$/$PM_{2.5}$/VOCs are mainly from area sources. $NO_X$ are mainly from point and mobile sources (TEPA, 2017).

Because Zheng et al. (2018) mainly discussed the anthropogenic emission in China, the authors have changed the citation to Li et al. (2017) and Kurokawa and Ohara (2020).

Kurokawa, J., and Ohara, T.: Long-term historical trends in air pollutant emissions in Asia: Regional Emission inventory in Asia (REAS) version 3.1, Atmos. Chem. Phys. Discuss., https://doi.org/10.5194/acp-2019-1122, in review, 2020.

Li, M., Zhang, Q., Kurokawa, J.-I., Woo, J.-H., He, K., Lu, Z., Ohara, T., Song, Y., Streets, D. G., Carmichael, G. R., Cheng, Y., Hong, C., Huo, H., Jiang, X., Kang, S., Liu, F., Su, H., and Zheng, B.: MIX: a mosaic Asian anthropogenic emission inventory under the international collaboration framework of the MICS-Asia and HTAP, Atmos. Chem. Phys., 17, 935–963, https://doi.org/10.5194/acp-17-935-2017, 2017.

TEPA: Building of the Taiwan emission data system. Taiwan EPA report, EPA-106-FA18-03-A263, in Chinese, 2017.

Zheng, B., Tong, D., Li, M., Liu, Fei, Hong, C., Geng, G., Li, H., Li, X., Peng, L., Qi, J., Yan, L., Zhang, Y., Zhao, H., Zheng, Y., He, K., and Zhang, Q.: Trends in China's anthropogenic emission since 2010 as the consequence of clear air actions. Atmos. Chem. Phys., 18, 14095–14111, https://doi.org/10.5194/acp-18-14095-2018, 2018.

| | | |
|---|---|---|
| L103-104: This should be "the impact of | Thanks the reviewer's suggestion. The authors have revised the | On line 123-127 |

| | | |
|---|---|---|
| reduction in source emission in each industrial region", because BFM can estimate "impact" not "contribution". Or you can define the wording that you will use the word "contribution" for the deference between control runs and sensitivity run. | narrative. | As mentioned above, the difference between the base and zero-out scenarios is the reduction of the specific source. The reduction can only approximate the contribution of that specific source when the chemical reactions are unimportant. This study shows that the pollutants from those three industrial regions are transported to Taiwan along with the northeast monsoon. Therefore, we can roughly estimate the contributions of BRIR, YRDIR, and PRDIR to $PM_{2.5}$ with the difference between the *Base* case and the ==BRIR==, ==YRDIR==, and ==PRDIR== cases. |
| L123-127: For Figure1, the formal, not abbreviated, names for each monitoring station should be appeared here. | Thanks the reviewer's reminder. The authors have merged the opinions of reviewer#1 and reviewer#2 and rewritten the names for each monitoring stations. | on line 144-148

For meteorology evaluation; we chose eight representative stations operated and maintained by the Taiwan Central Weather Bureau (CWB): Peng Jiayu (==PJY in Fig. 1==), Taipei (==TPE in Fig. 1==), Chupei (==CP in Fig. 1==), Taichung (==TC in Fig. 1==), Chiayi (==$CY_m$ in Fig. 1==), Tainan (==$TN_m$ in Fig. 1==), Kaohsiung (==KH in Fig. 1==), and Hengchun (==$HC_m$ in Fig. 1==) stations to evaluate the modeling performance of temperature, ==relative humidity==, wind speed, and wind direction.

On line 153-156

Since most residents live in the relatively flat western Taiwan, the observations of air quality |

| | | |
|---|---|---|
| | | monitoring stations operated and maintained by the Taiwan Environmental Protection Agency (TEPA) at the Banqiao (BQ in Fig. 1), Pingzhen (PZ in Fig. 1), Miaoli (ML in Fig. 1), Zhongming (ZM in Fig. 1), Chiayi ($CY_a$ in Fig. 1), Tainan ($TN_a$ in Fig. 1), Zuoying (ZY in Fig. 1), and Hengchun ($HC_a$ in Fig. 1) stations were chosen for $PM_{2.5}$ evaluation. |
| L130-131: Why you don't show the model domains in Figure 1 but just describe horizontal resolution? | Yes, the authors have redrawn the Figure 1 which shows the model domains in the revised manuscript. | Figure 1 |
| L146: "MB" has already been defined in the previous sentences | Thanks the reviewer for carefully pointing out this extra. The authors have already removed the repeat one. | |
| For the evaluation of WRF and CMAQ shown in Table 1 and 2, the results from which domain were used? And in addition to the summary of statistical indices in Table 1, figures of comparisons of temperature and wind between observation and simulation are quite informative. Could you put them together at least as supplement? | The authors have explained the simulated results from the fourth domain was evaluated for Table 1 and 2 in the revised manuscript.

The authors have added figures of comparisons of observed and simulated temperature (Fig. S4.1), wind speed (Fig. S4.2), relative humidity (Fig. S4.3), and wind direction (Fig. S4.4) in the supplement of the revised manuscript.

In addition, the authors also added Fig. S4.5 which show the comparisons of observed and simulated $PM_{2.5}$ in the supplement of the revised manuscript. | On line 185-189

This study used statistical indexes such as MB (Mean Bias), MAGE (Mean Average Gross Error), and IOA (Index of Agreement) to evaluate temperature and wind speed, and used WNMB (Wind Normalized Mean Bias) and WNME (Wind Normalized Mean Error) for wind direction in the fourth domain. For $PM_{2.5}$ performance in the same domain, we applied the MB, MFB (Mean Fractional Bias), and MFE (Mean Fractional Error), R (Correlation coefficient), and IOA indexes. All of the formulas for the above indexes are from Emery (2001) and |

| | | TEPA (2016), illustrated in Supplement S3. |
|---|---|---|
| You should explain how you draw Fig3. Are the values in Fig3 difference between Base case and sensibility case? If so, it's better to note it in the manuscript or in figure caption. Fig3 is a bit busy, so it seems better to select fewer locations out of seven to avoid redundancy. | Yes, the authors have already explained how to get the values in Fig. 3 both in the main content and the caption of figure 3. In addition, the authors have removed few locations but only remained BQ, ZM, and CY to representative northern, central, and southern Taiwan. | On line 223-224

As mentioned, the impact was considered as the reduction of a specific source removed or roughly the contribution of that specific source for BFM method, i.e., the difference between the base and zero-out scenarios, is applied in this study.

Caption of Figure 3

**Figure 3: The daily average impact of PM$_{2.5}$ from BRIR, YRDIR, PRDIR on air quality stations in Taiwan in January 2017. a,b, and c denote the impact on BQ, ZM, and CY from 1 (BRIR), 2 (YRDIR), and 3 (PRDIR). The impact was calculated with BFM method, i.e., the difference between the base and zero-out scenarios.** |
| L176: Remove unnecessary "the". | Thanks the reviewer for pointing out this typo. The authors have already removed the extra "the". | |
| Could you check the wording "China East Sea"? "East China Sea" has been also used for the same area in many literatures. | Thanks the reviewer's careful checking for this manuscript. The authors have already unified the nouns to "East China Sea" in the revised manuscript. | |
| For Figure5, you should explain how to deduce the values shown in the figure, in particular the values in Fig5(*-2,3,4). Are they the difference | Yes, the authors have followed the reviewer's suggestion to explained Fig. 5 are deduced by the difference between Base case and zero-out cases

Thanks the reviewer's reminder that the title of y-axis should be "daily | On line 263-265

Similar to Fig. 2, we deduced the differences of base and zero-out scenarios for the IPR analysis. |

| | | |
|---|---|---|
| between Base case and sensitivity case? If so, you should instruct briefly how to interpret these Figures. Is the title of y-axis correct? This should be "_concentration" or "daily concentration change"? | concentration change". The authors have already corrected this error in Fig 5 And Fig S4.9 in the revised manuscript. | This study considered the reduction as the approximate contribution by each industrial region. Therefore, the following discussion is satisfied when the chemical reaction between each industrial region and the surrounding area was ignored. |
| L204: Fig5(a-1) and (a-2) do not seem quite similar to each other. Could you specify more about which features of both figures look similar? | Yes, the authors agree that they did not use precise vocabulary and have removed the word "similar" to avoid misleading and rewritten the narratives. | On line 265-266

The physical or chemical terms in Fig 5 (a-1) and Fig. (a-2) did not always appeal synchronously, and their proportions in total were not equal. |
| L204: You concluded that main contributor to #17 PM2.5 is BRIR, but I cannot understand why you can conclude like this. The values in Fig5(a-1) and (a-2) are quite different. You should give an instruction how to read and understand the Fig5 | The authors have modified the narratives. Furthermore, they also added titles to the Fig. 5, Fig. 8, Fig. 9, Fig. S4.9, Fig. S4.11 and Fig. S4.12 in the revised manuscript such that the readers can understand the figures arranged in four columns are *Base*, *BRIR*, *YRDIR*, and *PRDIR* cases and the figures arranged in seven rows are #1, #2, #3, #4, BQ, ZM, and CY. Note that #1-#4 in the revised manuscript are the #17-#20 in the original manuscript. | On line 265-267

The physical or chemical terms in Fig 5 (a-1) and Fig. (a-2) did not always appeal synchronously, and their proportions in total were not equal. This implies #1 was influenced by both BRIR and other nearby sources. |
| L205: Can HADV process "produce" PM2.5? The term "production" here is not appropriate. | The authors understand what the reviewer meant and have already modified all such narratives. | On line 267-268

The increase of $PM_{2.5}$ was caused mainly by the process of HADV, followed by ZADV and VDIF, and the removal process was mainly AERO.

On line 287-288

The build-up of $PM_{2.5}$ at BQ were mainly |

HADV with minor CLDS, and the removal processes were mainly ZADV with minor AERO (Fig. 5(e-1)).

On line 303-304

For CY located in southwestern Taiwan, VDIF and HADV mainly contributed to the gains of $PM_{2.5}$, and the removal processes were mainly ZADV and AERO; however, occasionally, when the positive contribution to $PM_{2.5}$ were ZADV and VDIF, the removal processes were HADV and AERO (Fig. 5(f-1)).

On line 305-306

Comparing Fig. 5(f-2)-(f-4) and Fig. 5(g-2)-(g-4), it is obvious the positive and negative contribution to $PM_{2.5}$ for CY were very similar to those for ZM.

On line 333-334

The major processes below layer 9 (~310 m) contributing to the increase of $PM_{2.5}$ were HADV, VDIF, and ZADV, and the removal processes were DDEP and AERO (Fig. 8(b-3)).

On line 340-341

Although #2 and BQ were most affected by YRDIR, the major contribution processes at BQ below 200 m (layer 7) was HADV, followed by AERO and above 200 m it were either VDIF,

ZADV, or CLDS, or mixture of them.

On line 353-355

Second, for the haze from BRIR and YRDIR, the positive and negative contribution processes on BQ were mainly HADV/AERO and ZADV/VDIF below 200 m (layer 7, Fig. 8(e-3)) and less different processes at different layers above 200 m on Jan 13th.

On line 355-357

While on Jan 9th, the major processes leading to the increase of $PM_{2.5}$ at BQ were mainly HADV below 380 m (layer 10), AERO between 120 to 900 m (layer 5 to 15), and ZADV/CLDS between 650 to 1500 m (layer 13 to 19), as illustrated in Fig. 9(e-2)-(e-3).

On line 380-381

The positive and negative contribution processes were nonuniform below 80 m (layer 4).

On 381-382

However, from 120 m to 460 m (layer 5 to layer 11), the major processes to build-up of $PM_{2.5}$ were AERO and ZADV, and the removal process was mainly HADV.

On line 433-434

When the EAH moved to northern Taiwan, HADV and AERO were the major contribution

| | | processes of PM$_{2.5}$ at BQ. |
| | | On line 438-439 |
| | | The stronger the intensity of EAH, the more obvious was the impact on central and southern Taiwan, and the proportion of HADV contributed to the PM$_{2.5}$ budget was more obvious near the surface. |
| L211: What process considered in AERO can reduce PM2.5? | Since the ambient environment was cold in high latitude regions and warm in low latitude regions, the evaporation process of PM$_{2.5}$ occurred in the haze during transporting southward. In the simulation study of Chuang et al. (2008), the evaporation of NH$_3$NO$_3$ occurred for the PM$_{2.5}$ plume transported from Shanghai to Taipei and formed ammonia and nitric acid. It is expected the evaporation of organic carbon also occurred if ambient temperature increased. Another very minor process which could be ignored compared with abovementioned evaporation process is that PM$_{2.5}$ particles coagulate to coarse particles.

 Chuang, M. T., Fu, J. S., Jang, C. J., Chan, C. C., Ni, P. C., and Lee, C. T.: Simulation of long-range transport aerosols from the Asian Continent to Taiwan by a Southward Asian high-pressure system. Sci. total. Enviro., 406, 168–179, https://doi.org/10.1016/j.scitotenv.2008.07.003, 2008b. | |
| L213: If the intrusion of PM2.5 from PRDIR is like that depicted in Fig4, why the contribution of ZADV is not so large in Fig5(c-4)? Since #19 is located between PRDIR and Taiwan | Fig. 4 is the cross section of red line in domain 2 and domain3. The ZADV is not so large in Fig. 5(c-4) is probably # 3 (#19 in the original manuscript) is not on the red line (the cross section) in Fig. 1. In addition, the influence of PM$_{2.5}$ from PRDIR was mainly on the mountains, as | |

| | | |
|---|---|---|
| island and the transport of PM2.5 between them occurs about 1-2 km high above the surface as in Fig4, any kind of vertical (downward) motion should transport PM2.5 from that layer to the location of #19 which must be at the surface | shown in Fig. 2(e) and Fig. 2(f), i.e. at high altitude about 1-3 km. The downward motion is not obvious unless the plume was blocked by the mountains in Taiwan (Fig. 4) and enhanced by the passing of cold surge. | |
| L227: What does "minor PM2.5" means here? | The authors have replaced the word "minor" with "certain" in that sentence. | On line 290-291

In addition, certain PM$_{2.5}$ was formed in northern Taiwan, probably due to the high relative humidity, which was probably induced by the cloud or fog produced by terrain uplifting. |
| L228: Why can you describe "The PM2.5 at BQ then transport up- and then southwards"? Which figure show this transport of PM2.5? | Thanks the reviewer for pointing the error. The removal process of PM$_{2.5}$ at BG was mainly ZADV. In order to explain clearly, the authors have modified the narrative. | On line 282-283

The removal process of PM$_{2.5}$ at BQ was mainly ZADV, which can be explained by BQ being located in the Taipei basin and the PM$_{2.5}$ is transported up to leave the basin. |
| L228-229: Fig.(f-1) -> Fig5 (f-1) | Thanks the reviewer for pointing the error. The authors have already corrected the type. | On line 292-295

Comparing Fig. 5(f-1) with Fig 5(f-2)-Fig 5(f-3), it is obvious that the PM$_{2.5}$ of ZM was produced by local pollution, i.e., the downward diffusion of VDIF, which probably came from northern Taiwan and was removed through HADV to further southern Taiwan under the prevailing north wind. |
| L234-235: If this is true, why ZADV in Fig5 (f-4) is largely negative from Jan 8 to 10? | Because of the reviewer's comment, the authors found the ZADV has to be treated in an opposite way since the concentration gradient is | On line 247-248

The boundary layer mixing was enhanced by |

positive for $PM_{2.5}$ from PRDIR, which is different from the usual cases that $PM_{2.5}$ concentration was usually higher near surface. Therefore, the vertical gradient of PM2.5 is positive in this case. The authors have modified some narratives in the revised manuscript.

The following is a brief review that was not in the revised manuscript but provide to the reviewer for communication. Yen et al. (2013) suggested the downward motion could bring Southeast Asian biomass burning pollutants aloft to surface through the subsidence of cold surge through the analysis of wind field derived from NCEP Global Forecast System analyzed data. Chuang et al. (2016) applied the WRF/CMAQ and found the Southeast biomass burning aerosols could be blocked by the mountains in Taiwan and then the boundary layer mixing assisted the subsidence of aloft aerosols to the surface. Huang et al. (2020) suggested the 700-hPa LLJ (Low Level Jet) may have carried the biomass burning plumes aloft located south of the frontal system (cold surge) and accompanied the upward/downward motion south/north of the frontal system. The downward motion occurred at the north of the front or subsidence of cold air region. While in the simulation of present study, the ZADV was negative which also implied the downward advection occurred when the cold surge passed. However, it is a pity that there is no observation for the pollutants profile during the pass of cold surge. Otherwise, it would be more persuasive.

Chuang, M. T., Fu, J. S., Lee, C. T., Lin, N. H., Gao, Y., Wang, S. H., Sheu, G. R., Hsiao, T. C., Wang, J. L., Yen, M. C., Lin, T. H., and Thongboonchoo, N.: The Simulation of Long-Range Transport of Biomass Burning Plume and Short-Range Transport of Anthropogenic

the passing of a cold surge and increased $PM_{2.5}$ on the ground.

On line 298-303

On Jan 8th to 10th, the negative ZADV indicated the concentration was decreasing in the lower 20 averaged layers, but the concentration gradient was positive ( $\frac{\partial PM_{2.5}}{\partial z} > 0$ , the concentration of $PM_{2.5}$ from PRDIR was higher at a high altitude than that at a low altitude over Taiwan), which implies the vertical velocity had to be negative, i.e., a downward motion. Therefore, the boundary layer mixing of the aloft $PM_{2.5}$ plume was enhanced by the passing of the cold surge (Yen et al., 2013; Chuang et al., 2016).

Pollutants to a Mountain Observatory in East Asia during the 7-SEAS/2010 Dongsha Experiment. Aerosol. Air. Qual. Res., 16, 2933–2949, https://doi.org/10.4209/aaqr.2015.07.0440, 2016.

Huang, H.-Y., Wang, S.-H., Huang, W.-X., Lin, N.-H., Chuang, M.-T., da Silva, A. M., & Peng, C.-M. (2020). Influence of synoptic-dynamic meteorology on the long-range transport of Indochina biomass burning aerosols. Journal of Geophysical Research: Atmospheres, 125, e2019JD031260. https://doi.org/10.1029/2019JD031260.

Yen, M. C., Peng, C. M., Chen, T. C., Chen, C. S., Lin, N. H., Tzeng, R. W., Lee, Y. A., and Lin, C. C.: Climate and weather characteristics in association with the active fires in northern Southeast Asia and spring air pollution in Taiwan during 2010 7-SEAS/Dongsha Experiment, Atmos. Envoron., 78, 35-50, http://dx.doi.org/10.1016/j.atmosenv.2012.11.015, 2013.

| | | |
|---|---|---|
| L256: Why did you exclude Fig.8(a)? | The authors have cut down the discussion of July 2017. Therefore, the discussion of Fig. 8(a) has been removed because it is not important. The Fig. 8 in the original manuscript has been moved to Fig. S4.9. | Fig. S4.9 |
| L267: Could you put the prevailing wind vector in Figures 2 and 6, otherwise I can not verify what you described here and similar descriptions in the manuscript explaining the impact of wind patterns. | The authors have added monthly average wind field in Fig. 2 and Fig. 6 already. It is obviously the prevailing wind in winter was northeast wind (Fig. 2) but south wind in summer (Fig. 6). | Fig. 2 and Fig. 6 |
| L280: Layer4? Is this Layer14? | Thanks the reviewer for pointing out this typo. The authors have corrected 4 to 14 in the revised manuscript. | |
| L281: It is apparent that only vertical motion can not transport PM2.5 from BRIR to | Thanks the reviewer's comment. The authors would like express the transport from BRIR to #1 was not just horizontal but also vertical. The | On line 325-326
 This implies the transport path from BRIR to |

| | | |
|---|---|---|
| #17. What do you mean here? | authors have modified the narratives | #1 could be horizontal between BRIR and #1 and then vertical at the location of #1. |
| L282-283: Why does ascent (descent) motion enhance (decrease) aerosol formation? What processes are involved ? | The authors have added above narratives | On line 328-330

It is possible that the ascent motion of the air parcel near the warm surface moved to a cold environment at a higher altitude. This may cause condensation and trigger heterogeneous reactions of aerosols. In contrast, the descent motion of the air parcel may cause the evaporation of aerosols due to a warmer environment near the surface than aloft. |
| L291: Fig. (e-2)-(e-4) -> Fig11. (e-2)-(e-4). | Thanks the reviewer for pointing out this typo. The authors have corrected it in the revised manuscript. The Fig. 11 in the original manuscript have been changed to Fig. 8 in the revised manuscript. | Fig. 8 |
| L293: mixed -> mixture | Thanks the reviewer for pointing out the inappropriate word. The authors have corrected the word in the revised manuscript. | On line 339-341

Comparing Fig. 8(e-1) and Fig. 8(e-2)-8(e-4), it was found the BQ was much influenced by YRDIR. Although #2 and BQ were most affected by YRDIR, the major contribution processes at BQ below 200 m (layer 7) was HADV, followed by AERO and above 200 m it were either VDIF, ZADV, or CLDS, or mixture of them. |
| L340: higher -> lower? | Thanks the reviewer for pointing out this typo. The authors have corrected it in the revised manuscript. | On line 406-407

The simulated proportions of nitrate and ammonium in $PM_{2.5}$ were slightly lower than the observations. |

| | | |
|---|---|---|
| L341: underestimated -> overestimated? | Thanks the reviewer for pointing out this typo. The authors have corrected it in the revised manuscript. | On line 407

While the simulated proportions of $K^+$, $Ca^{2+}$, $Mg^{2+}$, $Na^+$ were slightly overestimated. |
| L353: There is not Fig.S2.6 in the supplement | The authors have removed Fig. S2.6. | |
| L380: There is no comparison for July 30th (no Fig. S2.6). | It is really a pity that there is no observation on July 30th due to bad weather (the influence of the thermal low). The authors have removed this figure already. | |

**Review #2**

| General Description | | |
|---|---|---|
| (1) comments from Reviewers | (2) author's response | (3) author's changes in manuscript. |
| This paper describes the contribution of three major Asian industrial regions on PM2.5 concentrations in Taiwan in January and June 2017. WRF and CMAQ models were used to simulate the transport of pollutions from the Asian industrial regions and also the chemical reactions in these plumes. The performance of the model in capturing temperature, wind speed, and direction, and PM2.5 was evaluated in multiple stations located in Taiwan covering north to south of the island. The authors used the process analysis technique in CMAQ to identify the dominant physical and chemical processes for the production and removal of | The authors really appreciate the reviewer#2 who spent his/her time reading and commenting the manuscript very carefully.
The authors have asked a professional English editing company to revise the English writing before submitting the revised manuscript. Meanwhile, they have tried their best to redraw designated figures and revise the manuscript according to the reviewer's valuable comments, | **Please notice that the revision according to reviewer#2's comments are written in** **yellow background.** |

| | | |
|---|---|---|
| PM2.5 in different locations in the domain. In general, the topic is suitable for ACP journal and the paper makes interesting conclusions about the contribution of long-range transport under different transport patterns to the air quality of Taiwan. However, the authors need to address some scientific issues discussed in the comments section below. The paper needs major English proofreading, major technical corrections, better quality for figures. I would not recommend this

paper for publications unless these issues are addressed.

Please note that I reviewed the updated version of the paper after the comments from reviewer 1 were addressed. | | |
| Specific comments | | |
| (1) comments from Reviewers | (2) author's response | (3) author's changes in manuscript. |
| 1) The contribution of local emissions was discussed very briefly in the last section of the paper. I believe adding a discussion about the contribution of local emission to the measured PM2.5 can be beneficial for drawing fair conclusions. | The authors have tried to discuss the contribution of local pollution to measured PM2.5 and added related narratives in several places in the revised manuscript.

. | On line 209-220

The difference between observed PM$_{2.5}$ in January and that in July is between 1.8 µg m$^{-3}$ to 31.8 µg m$^{-3}$, the largest in southern Taiwan (CY, TN, and ZY) followed by central (ZM and ML) and northern Taiwan (BQ and PZ), and the smallest at HC. Since the LRT in the prevailing northeast wind should have more impact on |

upstream northern Taiwan than downstream southern Taiwan (Chuang et al., 2018), this reveals that the LP has more impact on southern Taiwan than northern Taiwan. Chuang et al. (2018) used to estimate the contribution of LRT and LP under prevailing northeast wind from 2006 to 2015. The contribution of LP to northern, central, and southern Taiwan were 40%, 60%, and 70% for ordinary events.

The $PM_{2.5}$ at HC is lower compared to the other stations because it is located in a small town, unlike the other stations that were in large cities. This suggests HC is influenced by the local mobile and area emissions and background atmosphere. Even if we ignore the LP and assume the background atmosphere is the only $PM_{2.5}$ source for HC, from Table 2, it is estimated that the contributions of local pollution for northern (BQ and PZ), central (ML and ZM), and southern Taiwan (CY, TN, and ZY) were 41–42%, 54–63%, and 75–78% in January, and 22–32%, 33–48%, and 36–39% in July, respectively. However, the $PM_{2.5}$ levels in January were much higher than those in July due to the impact of EAH.

On line 292-295

Comparing Fig. 5(f-1) with Fig 5(f-2)-Fig 5(f-3), it is obvious that the $PM_{2.5}$ of ZM was produced by local pollution, i.e., the downward diffusion of VDIF, which probably came from northern Taiwan and was removed through HADV to further southern Taiwan under the prevailing north wind.

One line 376-378

We can consider the Asian continent has almost no impact on Taiwan in July. In other words, the origin of $PM_{2.5}$ in Taiwan in July is local pollution and the background atmosphere.

On line 385-386

This suggested the $PM_{2.5}$ was mainly from local pollution and background atmosphere in July.

On line 404-405

In addition, the proportions of nitrate in $PM_{2.5}$ at BQ, ZM, and CY were higher than those over #1 - #4. That should be caused by the local pollution.

On line 440-442

In July 2017, the influence from the three industrial regions on the $PM_{2.5}$ was ignorable in Taiwan, i.e., $PM_{2.5}$ mainly came from local or upwind adjacent sources and the background

| | | atmosphere unless there was special weather system, e.g., a thermal low nearby that may carry small amounts of pollutants from PRDIR to Taiwan. |
|---|---|---|
| 2) I recommend adding backtrajectory analysis using HYSPLIT when discussing transport patterns on specific days. I added more details in the specific comments section. | The authors have added backward trajectory figures by using HYSPLIT modeling results on Jan 13th, Jan 9th, July 18th, and July 30th in Fig. S4.7.

Yes, the authors agree that backward trajectory is useful for LRT analysis. However, the users need to be careful when terrain is near the location of origin and when the wind field is chaotic around the origin. | Fig. S4.7 |
| 3) The paper misses a lot of important information such as the main configurations of the model, details on the emission inventory used, and information about the location of measurement sites and equipment. I highly recommend adding these to the paper for the purpose of reliability and reproducibility of the work. | Thanks for the reviewer's suggestions. The authors have added a Table (Supplement 1) describing the model configuration, emission maps (Supplement 2), revised the way of display the location of measurements (Fig. 1), and added narratives of measuring equipment (section 2.1). | Model configuration: please refer to Supplement 1.

Details on the Emission inventory: please refer to Supplement 2.

Information about the location of measurement sites: Fig. 1.

Information about the equipment: on line 148-152

The Propeller Wind Direction Anemometer (Komatsu's Geophysical Instruments), Isuzu Seisakusho 3-3122 Quartz Precision Thermo-Hygrograph (Isuzu Seisakusho Co.,Ltd.), and R.M. Young 05103 Pt-Electrical Resistance Thermometer (R.M. Young Company) were used to monitor the wind speed/direction, relative humidity and air temperature, |

| | | |
|---|---|---|
| | | respectively. The measurement equipment was under routine calibration by the Taiwan CWB (https://www.cwb.gov.tw/Data/knowledge/announce/MIC.pdf).

On line 157-162

The automatic meteorological and air quality data are provided in hourly recordings to the public.

In this study, we also compared the modeling results with the $PM_{2.5}$ composition analyzed by Lee et al. (2017) at BQ, ZM, and $CY_a$ for Jan 13 and July 18, 2017. They used the MetOne SASS $PM_{2.5}$ samplers (Met One Instruments, Inc.) for collection of the $PM_{2.5}$ composition samples at six stations every six days. The quality assurance of the $PM_{2.5}$ monitoring and analysis is referred to chapter 4 of Lee et al. (2017). |
| 4) Were there any seasonal or diurnal cycle in the emissions? Are January and July emissions different? | Yes, there is seasonal/diurnal cycles for anthropogenic and biogenic emissions, only diurnal for aircraft emissions.

While for remaining emissions, there is no seasonal/diurnal variation like shipping emissions.

For biomass burning emissions, it directly depends on the FINN database.

In summary, yes, the emissions for January and July are slightly different. | |

| | | |
|---|---|---|
| 5) Major changes are required for the figures. The texts are too small in many of them, the color bar can be improved. I added more comments about each figure in specific comments. | Thanks for the reviewer's comments. The authors have tried their best to redraw nearly all of the figures accordingly to those specific comments. | |
| 6) I did not make comments on the grammatical mistakes, incomplete sentences, and inconsistencies as there were too many. | Before submitting the revised manuscript, the authors have asked a professional English language editing company to revise the English writing of the manuscript. | |
| Specific Comments | | |
| 1. The first two paragraphs in the Introduction section need to be re-written with better English. | The authors have asked a professional English language editing company to revise the English writing before submitting the revised manuscript. | |
| 2. L69. The reference at the end of the sentence (Byuan and Schere, 2006) does not match the reference at the beginning of the sentence (Kwok et al. (2013)). | The reference "Byuan and Schere, 2006" is for CMAQ model which shows for the first time in the manuscript. | |
| 3. L65. Consider starting a new paragraph when describing the AM method. | Thanks for the reviewer's suggestion. The authors have started a new paragraph for the AM method. | |
| 4. L65-75. After reading this section I was under the assumption that the AM method performs better and was used in this study. At the end of this paragraph please mention that you did not use the AM method and used the BFM method instead. | The authors agree with the reviewer's opinion that AM method could be better than BFM method for this study. At the moment we executed the simulation, we haven't resolved using the AM method yet. Therefore, the authors applied the BFM in this study.

The authors have added the description that they suggest to use AM method for future studies. | On lien 78-82

The CTM, especially the AM method, is able to give clearer contributions from a specific source compared to the TS method or the BFM method. However, the AM method requires large computer resources and complicated preparation of individual emission files. Therefore, the AM method was not used in this |

| | | |
|---|---|---|
| | | study and we selected BFM instead. Despite this, the AM method should be widely used when computer resources are not a problem. |
| 5. L86...nitrate and sulfate: : : Please be consistent and either use the chemical formula or the name in the paper or both. | The authors have rewritten that narrative to avoid misleading. | On line 94-95

They found the proportion of nitrate in $PM_{2.5}$ would decrease but that of sulfate would increase along the transport path. |
| 6. L99. When is the northeast monsoon period? Which season/months? | Chuang et al. (2018) have analyzed the northeast monsoon PM2.5 level from 2006-2015 in Taiwan. It is noted that the northeast monsoon has to be connected to anticyclones originating from the Siberian-Mongolian. The northeast monsoon usually started from Autumn to about one month after Spring, i.e., from September to May of next year.

Chuang, M.T., Chung-Te Lee, Hui-Chun Hsu, 2018. Quantifying PM2.5 from long-range transport and local pollution in Taiwan during winter monsoon: An efficient estimation method. Journal of Environmental Management 227, 10-22. | On line 41-42

The observations of meteorology from the Taiwan Central Weather Bureau showed that the winter monsoon usually extends from September to May (Chuang et al., 2018). |
| 7. L111. Change Brir to BRIR : : :same for other emission regions. | The authors have followed the reviewer's suggestion and have changed *Brir* to *BRIR* and other similar nouns. | On line 120-123

It applied the CTM with the BFM method to simulate four scenarios: *Base* (control case with integrated emissions), *BRIR* (all emissions except BRIR), *YRDIR* (all emissions except YRDIR), and *PRDIR* (all emissions except PRDIR) scenarios and thus resulted in the determining the contributions of each industrial region. |

| | | On line 126-127 |
|---|---|---|
| | | Therefore, we can roughly estimate the contributions of BRIR, YRDIR, and PRDIR to $PM_{2.5}$ with the difference between the *Base* case and the ==*BRIR*==, ==*YRDIR*==, and ==*PRDIR*== cases. |
| 8. L115. What do you mean by "meandering movement"? You can here refer toprevious studies that showed this. | Thanks the reviewer for pointing out the confusion. The authors have rewritten that sentence. | On line 125-126

This study shows that the pollutants from those three industrial regions are transported to Taiwan along with the northeast monsoon. |
| 9. L120. I suggest moving the discussion of monsoon seasons earlier in the introduction section. | Thanks for the reviewer's suggestion. The authors have written a discussion of monsoon seasons in the introduction section. | On line 40-43

==Chang et al. (2011) described the East Asian Winter monsoon is characterized by the cold-core Siberian-Mongolian High at the surface. The observations of meteorology from the Taiwan Central Weather Bureau showed that the winter monsoon usually extends from September to May (Chuang et al., 2018). During the winter monsoon period, northeast wind prevails over East Asia and transports East Asian haze (EAH) to downwind regions,== including Korea, Japan, and Taiwan (Zhang et al., 2015). |
| 10. L128. In addition, year 2017 : : :. I don't understand this sentence. | The authors have rewritten the narrative. | On line 138-141

==In previous studies (Zheng et al., 2018; Chuang et al., 2018), the anthropogenic emissions in China have obviously decreased since 2013; therefore, to show the difference of== |

| | | transport between winter and summer, this study chose January and July 2017 to represent the LRT in the winter and summer period and the contrast between them, with more discussion on the winter transport due to greater impact of EAH. |
|---|---|---|
| 11. 2.1. Geographical location of meteorological : : : Are stations with the same names (for example #5 and #13) in the same locations? In the text, you use the station names but in Fig 1, you used the numbers. To find the location of each station in Fig 1 readers must go back and forth between section 2.1, fig 1 and the text. Please be consistent and either use numbers or names in figures, tables, and text. | Actually the geographical locations of meteorological and air quality stations with the same name is not the same but in the same town or city. That's why they have the same name.

Thanks for the reviewer's opinion. The authors have removed the numbers for meteorological and air quality stations in section 2.1 and the caption of Fig. 1. | The caption of Figure1

**Figure 1: Geographic location of three major industrial regions (BRIR (blue line enclosed region), YRDIR (green) and PRDIR (orange)) in East Asia and meteorological and air quality stations in Taiwan. Meteorological stations: PJY, TPE, CP, TC, $CY_m$, $TN_m$, KH, and $HC_m$; air quality stations: BQ, PZ, ML, ZM, $CY_a$, $TN_a$, ZY, and $HC_a$. The numbers in red along the coast of East China #1, #2, #3, and #4, represent the locations of Bohai sea, East china Sea, Taiwan Strait, and northern South China Sea, respectively. The red line is the cross-section plot for Figure 4.**
On line 144-148
For meteorology evaluation; we chose eight representative stations operated and maintained by the Taiwan Central Weather Bureau (CWB): Peng Jiayu (PJY in Fig. 1), Taipei (TPE in Fig. |

| | | |
|---|---|---|
| | | 1), Chupei (CP in Fig. 1), Taichung (TC in Fig. 1), Chiayi (CY$_m$ in Fig. 1), Tainan (TN$_m$ in Fig. 1), Kaohsiung (KH in Fig. 1), and Hengchun (HC$_m$ in Fig. 1) stations to evaluate the modeling performance of temperature, relative humidity, wind speed, and wind direction. On line 153-156 Since most residents live in the relatively flat western Taiwan, the observations of air quality monitoring stations operated and maintained by the Taiwan Environmental Protection Agency (TEPA) at the Banqiao (BQ in Fig. 1), Pingzhen (PZ in Fig. 1), Miaoli (ML in Fig. 1), Zhongming (ZM in Fig. 1), Chiayi (CY$_a$ in Fig. 1), Tainan (TN$_a$ in Fig. 1), Zuoying (ZY in Fig. 1), and Hengchun (HC$_a$ in Fig. 1) stations were chosen for PM$_{2.5}$ evaluation. |
| 12. 2.1. Geographical location of meteorological : : : Please provide more information about the measuring equipment, the temporal resolution of data and reference to the measurement data used. | The authors have found out the information of measurement equipment of wind, temperature, relative humidity, and PM2.5. The temporal resolution of data is hourly. As for manual sampleing, Lee et al. (2017) used the MetOne SASS PM$_{2.5}$ sampler (Met One Instruments, Inc) to collect PM$_{2.5}$ at six stations every six days. In addition to PM$_{2.5}$ mass, they analyzed the inorganic ions and organic/element carbon for all the PM2.5 samples. | On line 148-152 The Propeller Wind Direction Anemometer (Komatsu's Geophysical Instruments), Isuzu Seisakusho 3-3122 Quartz Precision Thermo-Hygrograph (Isuzu Seisakusho Co.,Ltd.), and R.M. Young 05103 Pt-Electrical Resistance Thermometer (R.M. Young Company) were used to monitor the wind speed/direction, relative humidity and air temperature, |

| | | |
|---|---|---|
| | | respectively. The measurement equipment was under routine calibration by the Taiwan CWB (https://www.cwb.gov.tw/Data/knowledge/announce/MIC.pdf).

On line 156-162

The METONE_BAM1020 particulate monitor (Met One Instruments, Inc.) was used to monitor $PM_{2.5}$. The automatic meteorological and air quality data are provided in hourly recordings to the public.

In this study, we also compared the modeling results with the $PM_{2.5}$ composition analyzed by Lee et al. (2017) at BQ, ZM, and $CY_a$ for Jan 13 and July 18, 2017. They used the MetOne SASS $PM_{2.5}$ samplers (Met One Instruments, Inc.) for collection of the $PM_{2.5}$ composition samples at six stations every six days. The quality assurance of the $PM_{2.5}$ monitoring and analysis is referred to chapter 4 of Lee et al. (2017). |
| 13. L142. : : :NCEP diagnostic fields. Please use a reference for this data set. There is doi available for this data set. | Thanks for the reviewer's remainder. The authors have supplemented the reference for that data set. | On line 165-167

The initial meteorological condition was from ds083.3 NCEP GDAS/FNL 0.25 Degree Global Tropospheric Analyses and Forecast Grids (DOI: 10.5065/D65Q4T4Z, https://rda.ucar.edu/datasets/ds083.3/). |
| 14. L142. Which nesting method did you use? | For WRF modeling, two-way was used; for CMAQ, one-way was | On line 165 |

| | | |
|---|---|---|
| One or two-way? | used. The authors have supplemented that narrative in the revised manuscript. | The WRF and CMAQ modeling used two-way and one-way nesting methods, respectively, in this study. |
| 15. L144. What is the model's top? | The authors have supplemented the information of model's top in the revised manuscript. | On line 170

The model's top is set to 50 hPa. |
| 16. L145. What is the temporal and special resolution of the emission inventories used? Is there a diurnal or seasonal variability? | The temporal resolution of emissions is 1 hour. While the spatial resolution of MIX and TEDS10.0 are 45 km and 1 km, respectively. We regrided the data to fit the design of model resolution.

For anthropogenic (like industry, power plants, residential, and transportation) and biogenic emissions, there are diurnal and seasonal variability. The temporal profile outside Taiwan regions is provided by Li et al. (2017). While the temporal profile in Taiwan is partly from TEPA (2017) and partly from government's publications. | |
| 17. L150. Why different biogenic inventories were used for different domains? | In Taiwan, we can get plant species distribution data from Forestry Bureau, Council of Agriculture. The number of plant species or the accordingly emission factors in database for Taiwan is far more than that in MEGAN v2.1. Therefore, we can apply the BEIS in SMOKE emission processing system to produce biogenic emissions for domain. However, for regions outside Taiwan, we don't find such detailed database; therefore, we can only apply the MEGAN model to produce biogenic emission. | |
| 18. 2.2 Models and modeling configuration Please add a table (can be in SI) with all main WRF and CMAQ configurations and schemes such as PBL scheme, LSM, | Thanks for the reviewer's suggestion. The authors have added the modeling configuration in Supplement 1.

The spin-up was 10 days for the simulations.

For chemical modeling, we used a very clean initial and boundary conditions in which the pollutants concentrations are about the same | On line 179-181

The model configurations of physics and chemistry for this study are listed in Supplement 1; and the emission maps of e.g., NO for four domains are referred to Supplement 2. |

| | | |
|---|---|---|
| cumulus scheme, ... How long was the spin-up? What did you use for chemical initial and boundary conditions? | magnitude as that based on year 2010, provided by MICS_Asia modeling group. In our experience, such low pollutants concentrations has nearly impact on the modeling results after 10 days spin-up. | |
| 19. 2.2 Models and modeling configuration Did you do any nudging or re-initialization of the model? Please add details to this section. | Yes, this study has applied the FDDA in the simulation. The grid nudging was used for domain 1, 2, and 3. While the observation nudging was used for domain 4 with meteorological data from 26 surface meteorological stations and 2 radio sonde stations.

No re-initialization was used. | On line 170-172

In order to get a better meteorological field, the WRF modeling applied four-dimensional data assimilation with grid nudging for domains 1, 2, and 3, and with observation nudging for domain 4. |
| 20. L161. Is there any RH data available? If yes then adding discussion on model performance in capturing RH can be very beneficial for the paper. | The authors have added the modeling performance of RH in Table 1. Furthermore, they also added the comparison of simulated and observed RH in Fig. S4.3.

The discussion of modeling RH performance is supplemented in section 2.3.1. | One line 199-204

Although there is no benchmark for relative humidity in Taiwan, the performance of simulated relative humidity is good. The relative humidity in KH was slightly overestimated compared with the other stations but still acceptable. The comparisons of the observed and simulated temperature, wind speed, relative humidity, and wind direction are illustrated in Fig. S4.1, S4.2, S4.3, and S4.4. |
| 21. L167: : :which is due to the smoother terrains: : : In Fig 1, HC is located very close to the sea. Is there a complex terrain in that region? It is not very clear in the figure. Can smoother terrain in the model impact other stations as well? | The star symbol indicates the location of the HC station. From the google map, it is obviously that the complex terrain is east of HC. Mountains around 500 meters on the east of HC stations reduced to around 100 to 200 meters in high resolution topographic height database of WPS preprocessing (preprocessor of WRF modeling). It the simulation could not totally reflect the effect of complex terrain blocking. Therefore, the wind speed was overestimated at HC. | |

| | | |
|---|---|---|
| |
[Figure]
 Except HC, other stations chosen for performance evaluation is on flat plain far from complex terrain. The impact of smoother terrain should be less for other stations. | |
| 22. L169. Are other stations influenced by buildings? | Although the Central Weather Bureau (CMB) claims that their meteorological stations are not influenced by surrounding building at all. They also claim if the CMB stations are set up on flat ground, there is no building nearby. If not, the stations would be set up on the top of buildings. But, according to Lin et al. (2017), http://photino.cwb.gov.tw/rdcweb/lib/cd/cd03cons/compilation/2017/10 6M03-final.pdf), strictly speaking, it is hard to say whether other meteorological stations was influenced by nearby buildings nowadays. In other words, it is hard to say the micro-scale climate around meteorological stations is not influenced by nearby buildings. After all, the nearby buildings indeed would influence the wind field around the | |

stations even the adjacent building is not right next to stations. Moreover, nowadays the urban heat/cool island effect is getting worse in modern metropolitans which may have exerted impact on the observed temperature at stations. Then it is impossible to say that the stations are 100% not influenced by near buildings. While, Lin et al. (2017) concluded, basically, the meteorological observations at the meteorological stations are still representative for the meteorological conditions at high confidence.

Lin, 2017. Evaluation and countermeasures of the influence of metropolitan environment on the meteorological observation, Taiwan Central Weather Bureau report, in Chinese, MOTC-CWB-106-M-03, http://photino.cwb.gov.tw/rdcweb/lib/cd/cd03cons/compilation/2017/106M03-final.pdf, 93 pp.

| | | |
|---|---|---|
| 23. L173. Please use better quality plots for figure S2.3. Also, be consistent in the title of subplots. | Thanks for the reviewer's opinion. The authors have redrawn the figure S2.3 (current Fig. S4.4 in the revised manuscript) and revised the caption to be consistent with the y-axis title.
The wind vectors in the new figures are much clear now. | Fig. S4.4 |
| 24. Table 1 and table 2. Please add mean model and observed values to these tables This can help better compare January and June values and values in different stations. | The authors have added mean model and observed values in the new Table 1 and Table 2 in the revised manuscript. | Table 1 and Table 2 |
| 25. L173. 2.3.2. Evaluation of CMAQ chemical modeling: : : Please add an emission map. Are any of the stations close to major emission sources? | The authors have added emission maps for four domains in supplement 2.
The locations of evaluated air quality monitoring stations are embedded in grids. They are mostly influenced directly by mobile and | supplement 2 |

| | area sources but should be far from point sources. | |
|---|---|---|
| 26. L173. 2.3.2. Evaluation of CMAQ chemical modeling: : : Please mention that PM2.5 values are very low in HC compared to other stations (Fig S2.4) | The authors have added the narrative that PM$_{2.5}$ values are very low in HC compared to other stations. | On line 215-216

The PM$_{2.5}$ at HC is lower compared to the other stations because it is located in a small town, unlike the other stations that were in large cities. This suggests HC is influenced by the local mobile and area emissions and background atmosphere. |
| 27. L173. 2.3.2. Evaluation of CMAQ chemical modeling: : : Is there a significant difference between PM2.5 values in January compared to June? Results and Discussion | The authors have added a discussion on the difference of PM$_{2.5}$ values in January and July. | On 209-220

The difference between observed PM$_{2.5}$ in January and that in July is between 1.8 µg m$^{-3}$ to 31.8 µg m$^{-3}$, the largest in southern Taiwan (CY, TN, and ZY) followed by central (ZM and ML) and northern Taiwan (BQ and PZ), and the smallest at HC. Since the LRT in the prevailing northeast wind should have more impact on upstream northern Taiwan than downstream southern Taiwan (Chuang et al., 2018), this reveals that the LP has more impact on southern Taiwan than northern Taiwan. Chuang et al. (2018) used to estimate the contribution of LRT and LP under prevailing northeast wind from 2006 to 2015. The contribution of LP to northern, central, and southern Taiwan were 40%, 60%, and 70% for ordinary events.

The PM$_{2.5}$ at HC is lower compared to the |

| | | |
|---|---|---|
| | | other stations because it is located in a small town, unlike the other stations that were in large cities. This suggests HC is influenced by the local mobile and area emissions and background atmosphere. Even if we ignore the LP and assume the background atmosphere is the only $PM_{2.5}$ source for HC, from Table 2, it is estimated that the contributions of local pollution for northern (BQ and PZ), central (ML and ZM), and southern Taiwan (CY, TN, and ZY) were 41–42%, 54–63%, and 75–78% in January, and 22–32%, 33–48%, and 36–39% in July, respectively. However, the $PM_{2.5}$ levels in January were much higher than those in July due to the impact of EAH.

On line 366

Fig. 10(a) and Fig. 10(b) reveal that the impact of BRIR on $PM_{2.5}$ in Taiwan was negligible in July compared with January. |
| 28. L185. How did you calculate 5%? Is this for the whole island or 5% is the maximum value? | The impact expressed in percentage is the ratio of difference between BASE and zero-out case to BASE case.
The maximum impact of about 5 % is for northern Taiwan. Actually the magnitude is between 4.6% to 5.3% in the metropolitan Taipei area (The largest city in north Taiwan). We think it is ok to say "approximately" 5% for northern Taiwan. | On line 226-227

The impact was higher in northern Taiwan, approximately 5% of total $PM_{2.5}$. |

| | | |
|---|---|---|
| 29. Fig 2. Please consider using a better color bar. Why negative values for the color bar? Use more colors for 0-2ug/m3 (right column) and 0-5% (left column). | The authors have redrawn Fig. 2 and Fig. 10 according to reviewer's comments. More color scales are used especially for low values. Besides, negative values in the color bar has been eliminated. | Fig. 2 and Fig. 10 |
| 30. L186. Fig 3 only shows three stations, not seven. Why did you use only these stations? How far are they from major local emission sources? | The authors drew Fig. 3 for seven stations in the original manuscript. Because reviewer#1 thought Fig. 3 was a bit busy and suggested to remain few locations out of seven to avoid redundancy; therefore, the authors chose three stations: BQ, ZM, and CY because $PM_{2.5}$ sampling were implemented at these three stations, which is discussed in section 3.6.

Basically, these three stations are all located in cities. Therefore, they are influenced by mobile and area sources but they are a bit distant from point sources. | |
| 31. L187. This is not true for PRDIB contribution which is higher in central and southern Taiwan (C-2 and C-3) compared to northern Taiwan (C-1). | The authors agree with reviewer#2's opinion and have removed that sentence. | |
| 32. L189. January 8th or 9th? 14th or 13th? In Fig 3 column a 9th, 14th, 20th, in column b 9th, 13th, 20th had the highest PM2.5 concentrations and contribution from BRIB and YRDIB. Why did you pick 9th and 13th? Throughout the text, different days were | The authors picked January 13th for two reasons. First, according to their experience, January 13th is a classical common LRT $PM_{2.5}$ event. The $PM_{2.5}$ in Taiwan is a mix of LRT and LP. The impact of LRT on northern Taiwan is obviously higher than central and southern Taiwan. YRDIR get much attention because it has a great influence on Taiwan. Second, they got $PM_{2.5}$ sampling on that day. Lee et al. (2017) executed | On line 310-312
On most days, northeast wind prevailed over East Asia. In this section, we chose January 13, 2017 to discuss the physical and chemical processes in detail because it is a classical moderate EAH episode in which $PM_{2.5}$ sampling |

| | | |
|---|---|---|
| mentioned which can be confusing for the readers. Please be consistent and clearly justify your choice of Jan 9th and 13th. | PM$_{2.5}$ sampling every six days instead every day. While Jan 9th was selected because it is indeed a strong LRT PM2.5 event. On Jan 9th, the impact of EAH on central and southern Taiwan is comparable to northern Taiwan. However, it is pity that there is no PM2.5 sampling on Jan 9th.

Lee, C. T., Wang, J. L., Chou, C. C. K., Chang, S. Y., Hsiao, T. C., and Hsu, W. C.: Fine suspended particles (PM$_{2.5}$) compositions observations and analysis project for 2016 and 2017, EPA-105-U102-03-A284, https://epq.epa.gov.tw/EPQ_resultDetail.aspx?proj_id=1051435574&recno=&document_id=19986#tab3, in Chinese, 2017. | was implemented and will be discussed in section 3.6.

On line 348-349

The severe EAH episodes always go along with the arrival of strong anticyclones (Fig. 6(b)). This study chose January 9th to discuss because of its largest impact on January 2017. |
| 33. L 195. What do you mean by almost the same? Please be more specific. | The authors have rewritten the narrative. | On line 232-234

For the daily mean influence, the impact of YRDIR was also higher than BRIR and the influencing period were almost the same for both regions because EAH originated from YRDIR and BRIR arrived in Taiwan one after another under the prevailing northeast wind (Fig. 3(a-1)-3(a-3), Fig. 3(b-1)-3(b-3)). |
| 34. L196. : : :could reach : : : In which stations? 6-8 ug/m3 and 9-12ug/m3, why giving a range? | Thanks for the reviewer2#'s opinion. The authors have modified the value to 8 and 11 µg m$^{-3}$ instead of a range. | On line 19-20

When the Asian anticyclone moved from the Asian continent to the West Pacific, e.g., on Jan 9, 2017, the contributions from BRIR and YRDIR to northern Taiwan could reach 8 and 11 µg m$^{-3}$. |

| | | On line 235-236 |
|---|---|---|
| | | In particular, the contributions from BRIR and YRDIR to northern Taiwan could reach 8 and 11 μg m$^{-3}$ on Jan 9, 2017. |
| 35. L200. Please show where Fujian and Guangdong are in Fig 2. | Thanks the reviewer#2's reminder. The authors have added *F* and *G* to indicate Fujian and Guangdong province in Fig. 1 and added them in the caption of that figure. | The caption of Figure1

**F and G indicate the location of Fujian and Guangdong province, respectively.** |
| 36. L202. Fig. 4. There are two red lines in Fig. 1. Did you use both of them? Please clearly mention this in the text. | Thanks the reviewer#2 pointing out the extra red line. The authors have removed the unneeded one. | |
| 37. L214. Locations #17-20 are missing from the updated Fig 1. | Thanks the reviewer#2 pointing out the error. The authors somehow made a mistake in the updated version of manuscript. Currently the authors have change #17-20 to #1-4 in Fig. 1 in the revised manuscript. | Fig. 1 |
| 38. L 214. Please mention that you did not evaluate model performance (transport and chemistry) in these locations. | The authors have added the narratives that those physical and chemical processes are all based on the modeling results and no evaluation of such processes were made. | On line 261-262.

It should be noted that each term resolved by the process analysis is based on modeling results and no evaluation of such processes was available. |
| 39. L224. The positive and negative : : : I don't understand this sentence | Sorry that our writing led to reviewer#2's confusion. The authors have rewritten that sentence in the revised manuscript. | On line 265-266

The physical or chemical terms in Fig 5 (a-1) and Fig. (a-2) did not always appeal synchronously, and their proportions in total were not equal. |
| 40. Fig. 5. What does column 1 represent? What do you mean by contribution of total emission? Do you mean the base case? | The authors have emphasized that the column 1 is for base case in the title of each subplots and the caption of Fig. 5. | Fig. 5 |

| | | |
|---|---|---|
| 41. Fig. 5. Please add titles to the subplots. Or at least put titles for each row and column. It is very difficult to interpret this figure. | The authors have added titles for subplots of Fig. 5, Fig. 8, Fig. 9. | Fig. 5, Fig. 8, Fig. 9 |
| 42. L226. Can you be more specific about the evaporation of ammonium nitrate in PM2.5 when moving from high latitude to low latitude regions? | When aerosol plume moves from high latitude regions to low latitude regions, the ammonia nitrate would evaporate from aerosol phase to gas phase due to increasing ambient temperature. This process has been simulated by Chuang et al. (2008b) already. | On line 268-270
The removal process is likely caused by the evaporation of ammonium nitrate in the $PM_{2.5}$ plume moving from high latitude regions to low latitude regions through increasing ambient temperature (Stelson and Seinfeld, 1982; Chuang et al., 2008b). |
| 43. L245. I cannot distinguish between ZADV and CHEM in Fig 5. Use different colors | The authors have redrawn Fig. 5, Fig. 8, Fig. 9, Fig. S4.9, and Fig. S4.11 in which the color of ZADV has been change to yellow. | Fig. 5, Fig. 8, Fig. 9, Fig. S4.9, and Fig. S4.11 |
| 44. Fig 5e-1. Any comment on why the daily concentration change is much higher in BQ (#10) than others? Does this mean a high contribution of local emissions? Please discuss this. | The production term is mainly HADV and AERO, which indicate the LRT is the contribution instead of local emissions. The reason why the daily concentration change is much higher in BQ is possibly that BQ was also influenced by other upstream sources in addition to the three industrial regions. | |
| 45. L247. The removal process of : : :. This sentence is unclear. | The authors have rewritten that sentence and made it clear. | On line 291-292
The removal process of $PM_{2.5}$ at BQ was mainly ZADV, which can be explained by BQ being located in the Taipei basin and the $PM_{2.5}$ is transported up to leave the basin. |
| 46. L250. : : : the PM2.5 of ZM: : : I don't understand this sentence. | The authors have rewritten that sentence and made it clear. | On line 292-295
Comparing Fig. 5(f-1) with Fig 5(f-2)-Fig 5(f-3), it is obvious that the $PM_{2.5}$ of ZM was |

| | | produced by local pollution, i.e., the downward diffusion of VDIF, which probably came from northern Taiwan and was removed through HADV to further southern Taiwan under the prevailing north wind. |
|---|---|---|
| 47. L259. For CY: : : Please mention that CY (#14) and ZM (#13) are closer to each other than BQ (#10). | The authors have mentioned that CY and ZM are closer to each other than BQ. | On line 257-259

Although CY and ZM are closer to each other than BQ, CY was selected due to PM$_{2.5}$ being sampled at this station and it is representative among many stations in southern Taiwan. |
| 48. 3.2. The physical : : : Please justify why you chose to only use #10, #14, and #13 in this section. Please provide a more detailed discussion on the contribution of local emissions. | Although the local pollution is not the focus of this study, the authors have added the discussion of local emissions in the revised manuscript.

They chose BQ, ZM, and CY because PM2.5 sampling were implemented at these three stations. | On line 209-220

The difference between observed PM$_{2.5}$ in January and that in July is between 1.8 µg m$^{-3}$ to 31.8 µg m$^{-3}$, the largest in southern Taiwan (CY, TN, and ZY) followed by central (ZM and ML) and northern Taiwan (BQ and PZ), and the smallest at HC. Since the LRT in the prevailing northeast wind should have more impact on upstream northern Taiwan than downstream southern Taiwan (Chuang et al., 2018), this reveals that the LP has more impact on southern Taiwan than northern Taiwan. Chuang et al. (2018) used to estimate the contribution of LRT and LP under prevailing northeast wind from 2006 to 2015. The contribution of LP to northern, central, and southern Taiwan were |

40%, 60%, and 70% for ordinary events.

The $PM_{2.5}$ at HC is lower compared to the other stations because it is located in a small town, unlike the other stations that were in large cities. This suggests HC is influenced by the local mobile and area emissions and background atmosphere. Even if we ignore the LP and assume the background atmosphere is the only $PM_{2.5}$ source for HC, from Table 2, it is estimated that the contributions of local pollution for northern (BQ and PZ), central (ML and ZM), and southern Taiwan (CY, TN, and ZY) were 41–42%, 54–63%, and 75–78% in January, and 22–32%, 33–48%, and 36–39% in July, respectively. However, the $PM_{2.5}$ levels in January were much higher than those in July due to the impact of EAH.

On line 292-295

Comparing Fig. 5(f-1) with Fig 5(f-2)-Fig 5(f-3), it is obvious that the $PM_{2.5}$ of ZM was produced by local pollution, i.e., the downward diffusion of VDIF, which probably came from northern Taiwan and was removed through HADV to further southern Taiwan under the prevailing north wind.

One line 376-378

| | | |
|---|---|---|
| | | We can consider the Asian continent has almost no impact on Taiwan in July. In other words, the origin of $PM_{2.5}$ in Taiwan in July is local pollution and the background atmosphere.

On line 385-386

This suggested the $PM_{2.5}$ was mainly from local pollution and background atmosphere in July.

On line 404-405

In addition, the proportions of nitrate in $PM_{2.5}$ at BQ, ZM, and CY were higher than those over #1 - #4. That should be caused by the local pollution.

On line 440-442

In July 2017, the influence from the three industrial regions on the $PM_{2.5}$ was ignorable in Taiwan, i.e., $PM_{2.5}$ mainly came from local or upwind adjacent sources and the background atmosphere unless there was special weather system, e.g., a thermal low nearby that may carry small amounts of pollutants from PRDIR to Taiwan. |
| 49. L266. The section number is not correct. Why Jan 13th was discussed before Jan9th? How did you classify Jan 13th as a severe episode and Jan 9th as a moderate episode? | Thanks the reviewer#2 for pointing out the error. Jan 13th is a moderate but Jan 9th is a severe episode. In our experience, a moderate episode usually has more impact on northern Taiwan and less on central and southern Taiwan. The occurrence of such moderate cases are much | |

| | more than the severe cases. However, a strong episode could transport LRT haze all the way to southern Taiwan. Moreover, a severe could bring much more haze than a moderate one. The occurrence of severe cases are usually along with the passing of cold surge. | |
|---|---|---|
| 50. L274 Fig. 8. Please add the altitude of each layer to the figure. | The authors have redrawn Fig. 8, Fig. 9, and Fig. S4.11 and added altitude for each layer in the first column of subplots. | Fig. 8, Fig. 9, and Fig. S4.11 |
| 51. L275. The arrival of LRT haze on Jan 14-15 can also be seen in Fig 3. | The authors did not chose Jan 14th or 15th but Jan 13th and 9th because they think the contrast between Jan 13th and Jan 9th is obvious. Furthermore, there was $PM_{2.5}$ sampling implemented on Jan 13th. | |
| 52. Fig 8. Again I don't understand why Jan 13th was chosen for this discussion. The contribution of LRT was small on this day compared to Jan 14th or 15th. Maybe using these days for Fig 8 would be more helpful? | The authors picked January 13th for two reasons. First, according to their experience, January 13th is a classical common LRT $PM_{2.5}$ event. The $PM_{2.5}$ in Taiwan is a mix of LRT and LP. The impact of LRT on northern Taiwan is obviously higher than central and southern Taiwan. While Jan 9th was selected because it is indeed a strong LRT PM2.5 event. On Jan 9th, the impact of EAH on central and southern Taiwan is comparable to northern Taiwan. However, it is pity that there is no PM2.5 sampling on Jan 9th. The contrast between Jan 13th and Jan 9th is quite distinct. Second, they got $PM_{2.5}$ sampling on that day. Lee et al. (2017) executed $PM_{2.5}$ sampling every six days instead every day. | On line 310-312
 On most days, northeast wind prevailed over East Asia. In this section, we chose January 13, 2017 to discuss the physical and chemical processes in detail because it is a classical moderate EAH episode in which $PM_{2.5}$ sampling was implemented and will be discussed in section 3.6. |
| 53. L296. Downstream not upstream. | Under northeast wind, BQ is located at upstream of PRDIR. | |
| 54. L266 Analysis of : : : Adding Hysplit back-trajectories released from locations discussed in this section can be very helpful. It can reveal the trajectory and the origin of the plumes arrived at each of the locations and add confidence to this discussion. | The authors have added backward trajectory figures by using HYSPLIT modeling results on Jan 13th, Jan 9th, July 18th, and July 30th in Fig. S4.7 and discussed in the revised manuscript.
 We chose the ensemble method and reanalysis archived data for the calculating the backward trajectories. | On line 313-314
 The 72-hour backward trajectory ensemble starting from BQ/ZM/CY obviously traced back to the East Asia continent where BRIR and YRDIR are located (Fig. S4.7(a-1)-(a-3)).
 On line 349-350 |

| | | The 72-hour backward trajectory ensemble starting from BQ/ZM/CY on January 9th is similar to that on January 13th (Fig. S4.7(b-1)-(b-3)). |
|---|---|---|
| | | On line 383-385 |
| | | Furthermore, the 72-hour backward trajectory ensemble starting from BQ/ZM/CY on this day traced back to the clean Southwest Pacific, which implied the airflow was controlled by the Pacific High (Fig. S4.7(c-1)-(c-3)). |
| | | On line 387-388 |
| | | The 72-hour backward trajectory ensemble starting from the end at BQ/ZM/CY went through a cyclone near Taiwan and then to the South China Sea and Philippines (Fig. S4.7(d-1)-(d-3). |
| 55. L309. What is vv? | Thanks the reviewer#2 for pointing out this error in the updated version of manuscript. After checking the original manuscript, the authors have removed it. | |
| 56. 3.5 Analysis of the moderate : : : I think it is worth discussing this event further (similar to Jan 13th) especially with the high values in BQ at lower levels. | The authors have corrected the type that Jan 9th was a severe event instead of a moderate one, which should be Jan 13th. The authors have added discussion regarding to the high values in BQ at lower levels. | On line 359-362

The higher production of HADV without AERO near the surface on Jan 9th explains the massive accumulation of EAH over the Asian continent and the rapid movement of anticyclone. The strong and fast plume passing |

| | | BQ led to insufficient time for the formation of PM$_{2.5}$ at BQ but it could transport EAH further to southern ZM and CY. |
|---|---|---|
| 57. L316 : : :for all cities. Cities or stations | The authors have rewritten that sentence. | On line 368-369
As illustrated in Fig. S4.8, the daily contribution from the three industrial regions to western Taiwan was similar for all cities. |
| 58. L325. Why July 18th? I don't see high PM2.5 concentrations for July 18th in any of the subplots in row a (Fig. S2.8). | On most days of July, the impact of three industrial regions on Taiwan was extremely small because the prevailing wind is southwest or southeast wind. The authors picked July 18th, because they got PM2.5 sampling on that day ( Lee et al., 2017). | On line 379-380
Take July 18, 2017 as an example, in which the PM$_{2.5}$ sampling was implemented, it was found that #1 was influenced more by YRDIR than BRIR among three industrial regions (Fig. S4.11(a-1)-(a-4)).
On line 394-395
Lee et al. (2017) conducted PM$_{2.5}$ sampling at BQ, ZM, and CY every six days in 2017. Only the sampling days are suitable for analysis in this study. |
| 59. L325. The positive and negative contribution : : : Does this refer to July 18th? This is not shown in any figure. | Thanks the reviewer#2 for pointing out this error in the updated version of manuscript. After checking again, the authors have recovered the figure for July 18th in the supplement, which is the Fig. S4.11 in the revised manuscript. | |
| 60. Fig 2.9 and L330. Please use a better color bar. More colors between 0-20 ug/m3. | The authors have redrawn Fig. 7 and Fig S2.9 of the updated version of manuscript. The latter is current Fig. S4.10 in the revised manuscript. In addition, more color scales are added between 0-20 ug m$^{-3}$. | Fig. 7 and Fig. S4.10 in the revised manuscript. |

| | | |
|---|---|---|
| 61. How much is the local emission contribution in July and how does this compare with January? | In this study, the authors did not simulate other cases which can be used to estimate the local contribution. But they tried to discuss the impact of local pollutions in the revised manuscript. | On line 209-220

The difference between observed $PM_{2.5}$ in January and that in July is between 1.8 μg m$^{-3}$ to 31.8 μg m$^{-3}$, the largest in southern Taiwan (CY, TN, and ZY) followed by central (ZM and ML) and northern Taiwan (BQ and PZ), and the smallest at HC. Since the LRT in the prevailing northeast wind should have more impact on upstream northern Taiwan than downstream southern Taiwan (Chuang et al., 2018), this reveals that the LP has more impact on southern Taiwan than northern Taiwan. Chuang et al. (2018) used to estimate the contribution of LRT and LP under prevailing northeast wind from 2006 to 2015. The contribution of LP to northern, central, and southern Taiwan were 40%, 60%, and 70% for ordinary events.

The $PM_{2.5}$ at HC is lower compared to the other stations because it is located in a small town, unlike the other stations that were in large cities. This suggests HC is influenced by the local mobile and area emissions and background atmosphere. Even if we ignore the LP and assume the background atmosphere is the only $PM_{2.5}$ source for HC, from Table 2, it is estimated that the contributions of local |

pollution for northern (BQ and PZ), central (ML and ZM), and southern Taiwan (CY, TN, and ZY) were 41–42%, 54–63%, and 75–78% in January, and 22–32%, 33–48%, and 36–39% in July, respectively. However, the $PM_{2.5}$ levels in January were much higher than those in July due to the impact of EAH.

On line 292-295

Comparing Fig. 5(f-1) with Fig 5(f-2)-Fig 5(f-3), it is obvious that the $PM_{2.5}$ of ZM was produced by local pollution, i.e., the downward diffusion of VDIF, which probably came from northern Taiwan and was removed through HADV to further southern Taiwan under the prevailing north wind.

One line 376-378

We can consider the Asian continent has almost no impact on Taiwan in July. In other words, the origin of $PM_{2.5}$ in Taiwan in July is local pollution and the background atmosphere.

On line 385-386

This suggested the $PM_{2.5}$ was mainly from local pollution and background atmosphere in July.

On line 404-405

In addition, the proportions of nitrate in

| | | |
|---|---|---|
| | | PM$_{2.5}$ at BQ, ZM, and CY were higher than those over #1 - #4. That should be caused by the local pollution.

On line 440-442

In July 2017, the influence from the three industrial regions on the PM$_{2.5}$ was ignorable in Taiwan, i.e., PM$_{2.5}$ mainly came from local or upwind adjacent sources and the background atmosphere unless there was special weather system, e.g., a thermal low nearby that may carry small amounts of pollutants from PRDIR to Taiwan. |
| 62. L225. Where is Fig 15? | Thanks the reviewer#2 for point out this error. It should be Fig. 11. | On line 395-396

The sampling from Jan 13th was compared with simulated PM$_{2.5}$ compositions, as indicated in Fig. 11. |
| 63. L338. According to the main content: : :.
Are you referring to Fig 8? If yes then your statement is incorrect, BRIR and YRDIR did not have a contribution to #19 (c-2 and c-3) and #20 (d-2 and d-3). Looks like Jan 13th is not the best day to pick for this discussion. Is this measurement available on Jan 9th or 20th? | Jan 13th is a moderate EAH event. The impact of BRIR and YRDIR on #19 (#3 in the revised manuscript) and # 20 (#4 in the revised manuscript) is not obvious. However, the impact of YRDIR has certain impact on the northern Taiwan, BQ site. If the LRT is severe, the impact on ZM and CY can be comparable to that on BQ. It suggests that the distance of southward transport is related to the intensity of EAH and moving air masses.

As explained, the authors chose Jan 13th because it is a moderate event which is often seen in winter period and there is PM$_{2.5}$ sampling on this day. Moreover, the contrast between Jan 13th and Jan 9th was quite | On line 398-401

As illustrated in Fig. 11, on both Jan 12th and Jan 13th, the major compositions were sulfate and OC for #1 - #4. However, the proportion of nitrate in PM$_{2.5}$ at #1 on Jan 12th was slightly higher than that at #2 but much higher than that at #3 and #4. This can be explained by the nitrate evaporating from the aerosol phase to the gas phase for the PM$_{2.5}$ plume transported from high to low latitude regions (Chuang et al., 2008b). |

| | | |
|---|---|---|
| | distinct. Lee et al. (2017) held $PM_{2.5}$ sampling every six days. Therefore, it is a pity there is no measurement available on Jan 9th or 20th.

    The authors admit that they did not explain correctly. Therefore, they have rewritten the narratives in the revised manuscript. | |
| 64. Fig 11. OC and NH4+ colors are very similar. | The authors have redrawn Fig. 11, Fig. S4.12, and Fig. S4.13 and make the colors of OC and $NH_4^+$ distinguishable. | Fig. 11, Fig. S4.12, and Fig. S4.13. |

---

## Author Response (AR2)

Dear reviewers and editor:

We are really grateful to reviewers who spent much time reviewing the original manuscript. Through the review processes, we totally understand that this manuscript could not be accepted without the reviewers' valuable comments. Please notice that the revision according to reviewers' comments are written in **red words.** In this response, we have attached three files: the manuscript of the main context, the supplement, and the one-to-one response. Also, we sincerely thank for the editor and ACP staff's effort again.

Best regards.

Ming-Tung Chuang

**Response to Reviewers**
**Manuscript *acp-2019-762**

*We greatly appreciate the insightful comments and suggestions of the reviewers. Below please find a list of the Reviewers' remarks in contrast to our responses to them:*

**Review #1**

| Major Concerns | | |
|---|---|---|
| (1) comments from Anonymous Referee #1 | (2) author's response | (3) author's changes in manuscript. |
| The manuscript has been revised thoroughly well according to the reviewer's comments and become more scientifically focused than before. However, there still are several sentences not easy to understand their meaning properly which will require further English editing. Furthermore, there are several sentences which describe a figure or table, but it is not clear which part of figure or table they are describing. I strongly recommend the authors to check all | On behave of all authors the first author sincerely thank the reviewer spent a lot of time reading the manuscript, finding numerous errors, and giving many valuable comments. The first author has to be responsible for unclear expression of narratives in the manuscript. Before the last submission, the first author has asked two co-authors and a professional editing company to revise the manuscript. However, the revised manuscript still contains some improper writing. Due to limited time, the authors have asked a senior colleague to help revise it instead for this revised manuscript. Furthermore, the first author has tried to read carefully and repetitively to ensure the clarification of writings. | **In addition to the response regarding to comments in this review, the authors have revised other narratives in this revised manuscript.** On line 19-20 When the Asian anticyclone moved from the Asian continent to the West Pacific, e.g., on Jan 9, 2017, the contributions from BRIR and YRDIR to northern Taiwan could reach daily averages of 8 and 11 $\mu g\ m^{-3}$. |

| | | |
|---|---|---|
| such descriptions and revise them if necessary. One example of such description is in the line 267 starting from "The increase of PM2.5…". I could not understand which part of the Figure 5 you want to describe. There still are such unclear descriptions which greatly deteriorate the readability of the manuscript. Another big concern is that the introduction section is still long and redundant. It's good to have a thorough review of the background, but it should cite only indispensable papers. Followings are specific points for further revision. | **In this revision, the authors have tried to reduce some cited literatures in the Introduction section already, listed below:**

[revised manuscript text omitted]

|---|---|---|
| -L121: What does "integrated emissions" mean? | The authors have recover the "integrated emissions" back to "all emissions". | On line 105-108

It applied the CTM with the BFM method to simulate four scenarios: *Base* (control case with all emissions), *BRIR* (all emissions except BRIR), *YRDIR* (all emissions except YRDIR), and *PRDIR* (all emissions except PRDIR) scenarios and thus resulted in the determining the contributions of each industrial region. |
| -L139: Why "therefore" here? | Thanks the reviewer finding this improper writing. It was an error made in the last submission. The authors have modified the narratives and make the sentence smooth. | On line 122-126

In previous studies (Zheng et al., 2018; Chuang et al., 2018), the anthropogenic emissions in China have obviously decreased since 2013; therefore, a year after 2013 was chosen. Moreover, in order to show the difference of transport between winter and summer, this study chose January and July 2017 to represent the LRT in the winter and summer period and their contrast, with more discussion on the winter transport due to greater impact of EAH. |
| -L172: MIX inventory is not only for China | Yes, the authors have removed the words "Multiresolution Emission | On line 157-158 |

| | Inventory for China" which can be short to MEIC instead of MIX and in order to avoid misleading. | The anthropogenic emissions for East Asia and Taiwan island were obtained from MIX ( Li et al., 2017) and TEDS 10.0 (Taiwan Emission Data System, TEPA, 2017), which are based on the years 2010 and 2016, respectively. |
|---|---|---|
| -L174-176: Did you do this adjustment for MIX emission in entire model domain? | The authors only adjusted the MIX emissions for Chinese mainland and have modified that sentence to avoid misleading. | On line 158-161

 The MIX emissions of $SO_2$, $NO_X$, NMHC, $NH_3$, CO, $PM_{10}$, and $PM_{2.5}$ covering Chinese mainland were adjusted with changes of -62%, -17%, 11%, 1%, -27%, -38%, and -35%, respectively, according to the change of annual emissions between 2010 and 2017 (Zheng et al., 2018). |
| -Table1: Which scenario did you use for this table? This kind of basic information should be described in the table caption or manuscript. Temporal resolution of the observation should be described somewhere in the manuscript. The same comments go to Table2. | The model performance was for the base case. The authors have added the basic information in the manuscript and bottom of Table 1 and Table 2. | On line 176-177
 The MB performance for *Base* case shows that the temperature was slightly overestimated for PJY (Table 1), which is located in the outer sea of northern Taiwan.
 On line 190-191

 For the *Base* case, the simulated $PM_{2.5}$ was overestimated at all stations except CY and HC in January 2017 (Table 2). The performance of the trend (correlation coefficient, R) is acceptable or good for all stations except HC. |

| | | At the bottom of Table 1 and Table 2 |
| --- | --- | --- |
| | | Note: 1. The standard of statistical evaluation is based on Emery (2001) and TEPA (2016); 2. The above evaluation was for base scenario; 3. The observation and simulation data for above evaluation was in hourly resolution. |
| -L192-193: Does this sentence describe only for July case? | Thank the reviewer for pointing out this mistake. The authors have modified that sentence to indicate that is only for July case | On line 177-178
The MAGE of simulated temperatures at all stations are reasonable for both months. However, the IOA indicates the simulated temperature at PJY and KH in July was less correct. |
| -L219: How did you estimate these values of the contributions of local pollution? | We have improved the narratives for clearer description. We made a very simple assumption that the $PM_{2.5}$ at HC is the $PM_{2.5}$ from background atmosphere for all sites. The difference between measured $PM_{2.5}$ at each site and the background $PM_{2.5}$ is attributed to local pollutions for each sites, respectively. | On line 201-205
Even if we ignore the LP and simply assume the measured $PM_{2.5}$ at HC represents the background air quality for all sites, from Table 2, it is estimated that the contributions of local pollution was the difference between measured $PM_{2.5}$ at each sites and the background $PM_{2.5}$, for northern (BQ and PZ), central (ML and ZM), and southern Taiwan (CY, TN, and ZY) were 41–42%, 54–63%, and 75–78% of measured $PM_{2.5}$ in January, and 22–32%, 33–48%, and 36–39% in July, respectively. |
| -Figure 2: Figure caption should state that these figures show the difference between Base | The authors totally agreed with the reviewer's suggestion and have added information both in the manuscript and the caption of Figure 2. | On line 211-213.
For the impact of the three industrial regions on |

| | | |
|---|---|---|
| case and the other sensitivity simulation case for the sake of clarity. The term "impact" is not so clearly describing what you show here. | | PM$_{2.5}$ in Taiwan in January 2017, the monthly mean impact from BRIR (difference between *Base* and *BRIR* scenario) was approximately 0.7–1.1 µg m$^{-3}$ as illustrated in Fig. 2(a).

 The caption of Figure2
 **Figure 2: The monthly average wind field and impact of PM$_{2.5}$ from BRIR (difference between Base and zero-out scenarios): (a) concentration and (b)percentage ;YRDIR: (c) concentration and (d)percentage ; PRDIR: (e) concentration and (f) percentage on Taiwan in January 2017** |
| -Figure 3: The same indications for Figure2 above. Furthermore, the Y-axis of Fig 3 should be delta(concentration). | The authors have added similar information in caption of Fig. 3. In addition, the authors have modified the label of Y-axis to "Δ concentration" for Fig. 3 and Figure S4.8 in the manuscript of this submission. | The caption of Figure3
 **Figure 3: The daily average impact of PM$_{2.5}$ from BRIR, YRDIR, PRDIR on air quality stations in Taiwan in January 2017. a,b, and c denote the impact on BQ, ZM, and CY from 1 (BRIR), 2 (YRDIR), and 3 (PRDIR). The impact was calculated with BFM method, i.e., the difference between the base and zero-out scenarios.** |
| -L226: The "relative" impact was.. | Yes, the authors have added "relative" into that sentence which is more clear than the original. | On line 213-214
 The relative impact was higher in northern Taiwan, approximately 5% of total PM$_{2.5}$. |
| -L239: 2(f)) --> 2(f) | Thanks the reviewer for finding this error which have been revised | On line 226-227 |

| | | |
|---|---|---|
| | already. | The spatial distribution of influence from PRDIR was totally different from BRIR and YRDIR, as shown in Fig. 2(e) and Fig. 2(f). |
| -L266: appeal --> appear? | Thanks the reviewer for finding this error which have been revised already. | On line 254-255
The physical or chemical terms in Fig 5 (a-1) and Fig. (a-2) did not always appear synchronously, and their proportions in total were not equal. |
| -L275-276: Fig5(c-2) and Fig5(c-3) should be switched. | Thanks the reviewer for finding this error which have been revised already. | On line 264-266
For #3, PM$_{2.5}$ was influenced mainly by YRDIR (Fig. 5(c-3)) and occasionally by BRIR (Fig. 5(c-2)), but it was also influenced by PRDIR from the 8th to 12th (Fig. 5 (c-4)) with positive contribution of CLDS, which could be attributed to high relative humidity environment over Taiwan Strait. |
| -L276-277: Is this consistent with the fact that CLD is the main production process in Fig5(d-4)? | Thanks the reviewer finding this inconsistent narrative and have modified that in the revised manuscript. Meanwhile, the authors have removed "Fig. 5(d-4)" on line 280 of the original manuscript. | On line 264-266
For #3, PM$_{2.5}$ was influenced mainly by YRDIR (Fig. 5(c-3)) and occasionally by BRIR (Fig. 5(c-2)), but it was also influenced by PRDIR from the 8th to 12th (Fig. 5 (c-4)) with positive contribution of CLDS, which could be attributed to high relative humidity environment over Taiwan Strait.
On line 269-270
Although #4 is very near PRDIR, it was |

| | | influenced more by YRDIR (Fig. 5(d-3))-) and other sources in eastern and northern China rather than three industrial regions since the prevailing wind was mainly northeast wind in January. |
|---|---|---|
| -L281: What is the "north" here? | The authors would like to express sources in east and northern China other than BRIR and YRDIR. They have modified that sentence in the revised manuscript. | On line 269-270

Although #4 is very near PRDIR, it was influenced more by YRDIR (Fig. 5(d-3)) and other sources in eastern and northern China rather than three industrial regions since the prevailing wind was mainly northeast wind in January. |
| -L299: What does "the lower 20 averaged layers" mean here? Does Figure 5 show the daily process contributions averaged in the lower 20 layers? If so, you must clearly state it in the figure caption and/or in somewhere in the manuscript. | The authors have added the information in the manuscript and caption of Fig. 5 and Figure S4.9. | On line 287-291

On Jan 8th to 10th, the negative ZADV indicated the concentration was decreasing in the lower 20 averaged layers, where the daily processes occur, but the concentration gradient was positive ($\frac{\partial PM_{2.5}}{\partial z} > 0$, the concentration of PM$_{2.5}$ from PRDIR was higher at a high altitude than that at a low altitude over Taiwan), which implies the vertical velocity had to be negative, i.e., a downward motion.

The caption of Figure 5

**Figure 5: The daily contributions of individual processes averaged over the lower 20 layers to the concentrations of PM$_{2.5}$ in January 2017,** |

| | | **a,b,c,d,e,f, and g represent #1, #2, #3, #4, BQ, ZM, and CY, respectively;1, 2, 3, and 4 represent influence of total emissions (base case), BRIR, YRDIR, and PRDIR, respectively** |
|---|---|---|
| -L310: This is not always true. Could you specify when and where this statement is true? | Thanks the reviewer for pointing out this error. The authors have modified that sentence in the revised manuscript. | On line 300
On most days in winter period, northeast wind prevailed over East Asia. |
| -L331: What does "nonuniform" mean here? What do you want to mean? | The authors would like to express some process is not consistent in continuous layers. The have used "inconsistent" to replace "nonuniform" in that sentence. | On line 261-264
From Fig. 5(b-1)-(b-4), among the three industrial regions it is apparent that #2 was influenced by both the BRIR and YRDIR, mainly produced through inconsistent HADV, VDIF, ZADV, and CLDS; and removed through AERO and occasional HADV and DDEP processes, and almost unaffected by PRDIR.

On line 320-322
Although #1 was slightly influenced by YRDIR, the contribution of different processes from YRDIR on #1 was less and inconsistent (Fig. 8(a-3)). The contribution of different processes from PRDIR to #1 was also inconsistent and even less (Fig. 8(a-4)).

On line 370-371
The positive and negative contribution processes were inconsistent below 80 m (layer 4). |

| (1) comments from Reviewers | (2) author's response | (3) author's changes in manuscript. |
|---|---|---|
| -L382: Fig S4.12 --> Fig S4.11? | Thanks the reviewer for finding this error which have been revised already. | On line 372-373
Fig. S4.11 shows that the influence of the three industrial regions on #2, #3, #4, BQ, ZM, or CY were almost ignorable. |
| -Figure 5, 8. 9: BR --> BQ | Thanks the reviewer for finding this error which have been revised for Figure 5, 8. 9, S4.9, S4.11, and S4.13. | |
| -Figure 11: #17, 18, 19, 20 should be modified. | Thanks the reviewer for finding this error. The authors have modified the Fig. 11 and Fig. S4.14 already. | |
| -L445: overestimated <--> underestimated | Thanks the reviewer for finding this error which have been revised already. | On line 434-435
The simulated proportion of nitrate and ammonium in $PM_{2.5}$ during the winter was slightly underestimated, but the simulated $K^+$, $Ca^{2+}$, $Mg^{2+}$, $Na^+$ was overestimated at BQ, ZM, and CY. |

**Review #2**

| General Description | | |
|---|---|---|
| (1) comments from Reviewers | (2) author's response | (3) author's changes in manuscript. |
| For final publication, the manuscript should be accepted as is suggestions for revision or reasons for rejection (will be published if the paper is accepted for final publication) | We sincerely thank the reviewer who provided many valuable comments in previous reviewing processes. We have to say this manuscript could not be (if) accepted without the improvements regarding to those comments. Honestly, we did know this manuscript is not an excellent work but we will continue to study hard on unresolved issues of atmospheric chemistry. | |
| | | |